# Integrative dynamic structural biology unveils conformers essential for the oligomerization of a large GTPase

Thomas-O Peulen[1†], Carola S Hengstenberg[2†], Ralf Biehl[3], Mykola Dimura[1,4], Charlotte Lorenz[3,5], Alessandro Valeri[1], Julian Folz[1], Christian A Hanke[1], Semra Ince[2], Tobias Vöpel[2], Bela Farago[6], Holger Gohlke[4,7], Johann P Klare[8]*, Andreas M Stadler[3,5]*, Claus AM Seidel[1]*, Christian Herrmann[2]*

[1]Chair for Molecular Physical Chemistry, Heinrich Heine University Düsseldorf, Düsseldorf, Germany; [2]Physical Chemistry I, Ruhr University Bochum, Bochum, Germany; [3]Jülich Centre for Neutron Science (JCNS-1) and Institute of Biological Information Processing (IBI-8), Forschungszentrum Jülich GmbH, Jülich, Germany; [4]Institut für Pharmazeutische und Medizinische Chemie, Heinrich Heine University Düsseldorf, Düsseldorf, Germany; [5]Institute of Physical Chemistry, RWTH Aachen University, Düsseldorf, Germany; [6]Institut Laue-Langevin, Grenoble, France; [7]Institute of Bio-Geosciences (IBG-4: Bioinformatics), Forschungszentrum Jülich, Jülich, Germany; [8]Macromolecular Structure Group, Department of Physics, University of Osnabrück, Osnabrück, Germany

*For correspondence:
jklare@uos.de (JPK);
a.stadler@fz-juelich.de (AMS);
cseidel@hhu.de (CAMS);
Chr.Herrmann@rub.de (CH)

[†]These authors contributed equally to this work

**Abstract** Guanylate binding proteins (GBPs) are soluble dynamin-like proteins that undergo a conformational transition for GTP-controlled oligomerization and disrupt membranes of intracellular parasites to exert their function as part of the innate immune system of mammalian cells. We apply neutron spin echo, X-ray scattering, fluorescence, and EPR spectroscopy as techniques for integrative dynamic structural biology to study the structural basis and mechanism of conformational transitions in the human GBP1 (hGBP1). We mapped hGBP1's essential dynamics from nanoseconds to milliseconds by motional spectra of sub-domains. We find a GTP-independent flexibility of the C-terminal effector domain in the µs-regime and resolve structures of two distinct conformers essential for an opening of hGBP1 like a pocket knife and for oligomerization. Our results on hGBP1's conformational heterogeneity and dynamics (intrinsic flexibility) deepen our molecular understanding relevant for its reversible oligomerization, GTP-triggered association of the GTPase-domains and assembly-dependent GTP-hydrolysis.

## Editor's evaluation

This study uses a broad range of experimental and theoretical biophysical techniques to provide fundamental insights into the conformational pathway of the human dynamin-related GTPase guanylate-binding protein 1 from its resting to an active state. The convincing integrative approach identifies hitherto hidden, dynamic conformers. The work will be of interest to the communities of experimental and theoretical protein biophysics and protein dynamics in signal transduction.

## Introduction

The biological function of proteins is directly linked to dynamic changes of their structures. Conformational flexibilities, heterogeneities, and polymorphisms are known to enable interactions among biomolecules, promote promiscuity with different binding partners, and are essential for enzymatic activity (*Tompa and Fuxreiter, 2008*; *Hensen et al., 2012*). This is most evident for motor proteins such as myosin or dynamin, where cyclic structural changes are crucial for their function. Thus, for a mechanistic molecular understanding of biological processes, the structure and the associated dynamics of the key components need to be characterized in great detail, ideally on a single-molecule level (*Lerner et al., 2018*; *Lerner et al., 2021*).

While NMR spectroscopy is an excellent tool to map conformationally excited states and intermediates (*Neudecker et al., 2012*) the determination of dynamic biomolecular structures of large systems is extremely challenging. To-date no individual technique fully maps structures and dynamics on all time scales and on a length scale necessary to understand large molecular systems. Thus, multiple experimental techniques need to be combined to probe different aspects and unveil structures of large multi-domain proteins (*Felekyan et al., 2012*; *Kilic et al., 2018*; *Lerner et al., 2021*). Here, we present and apply a framework that integrates short and long-range distances with shape information amended by time-resolved spectroscopy and molecular dynamics simulations for dynamic structures. Our framework identifies functional elements as building blocks, and balances experimental information in a meta-analysis to generate integrative dynamic structures with a small number of informative distances.

We apply our framework to study molecular mechanisms of a guanylate binding protein (GBP), a class of soluble proteins that belong to the dynamin superfamily and to the class of interferon-γ induced effector molecules (*Praefcke and McMahon, 2004*). GBPs are important for innate cell-autonomous immunity in mammals. GBPs form supramolecular complexes during infection and are recognized for their immune activity against a wide range of intracellular pathogens such as viruses (*Anderson et al., 1999*; *Itsui et al., 2009*), and bacteria (*Kim et al., 2011*; *MacMicking, 2012*; *Li et al., 2017*). Noteworthy, a GBP in mice translocates from the cytosol to endomembranes and attacks the plasma membrane of eukaryotic cellular parasites by the controlled formation of productive and supramolecular complexes (*Kravets et al., 2016*). As a prime example for a GBP, we study the human GBP1 (hGBP1). hGBP1 shows nucleotide-dependent dimerization (*Ghosh et al., 2006*), and the formation of supramolecular structures promoted by GTPase activity (*Shydlovskyi et al., 2017*). X-ray crystallography on the full-length hGBP1 revealed a folded and fully structured protein with the typical architecture of a dynamin superfamily member. hGBP1 consists of a large GTPase domain (LG domain), an alpha-helical middle domain, and an elongated, also purely alpha-helical, effector domain comprising the helices α12 and α13, with a length of 120 Å (*Prakash et al., 2000*; *Figure 1A*). X-ray crystallography (*Ghosh et al., 2006*) and biochemical experiments (*Ince et al., 2017*) identified the LG domains as interface for GTP induced homo-dimerization. Like for other membrane-associated dynamins that form tubular shaped assemblies to fuse or divide membranes in cells (*Faelber et al., 2012*; *Reubold et al., 2015*), cylindrical and tubular structures have been observed for hGBP1 (*Shydlovskyi et al., 2017*). For hGBP1 neither molecular structures of these tubules nor precursor structures in solution that could inform on the assembly pathway are known (*Cui et al., 2021*). Previous FRET and DEER experiments on hGBP1 dimers identified two conformers. In the dominant dimer, the two C-terminal α13 helices associate (*Vöpel et al., 2014*). This is in line with live-cell experiments that highlight the relevance of helix α13 for the immune response (*Tietzel et al., 2009*; *Li et al., 2017*; *Piro et al., 2017*). Previously, we identified monomeric and dimeric forms of farnesylated hGBP1 by SEC-SAXS and ultracentrifugation (*Figure 1—figure supplement 1*). These experiments lead to the hypothesis that specific intramolecular interactions stabilize the GTPase and act as a safety mechanism preventing hGBP1 dimerization (*Lorenz et al., 2020*). Here, to unravel the conformational changes necessary for the formation of a fully bridged dimer (b-hGBP1:L)$_2$ (*Figure 1B*), we study non-farnesylated hGBP1, where nucleotide ligands L (GTP) are bound and the effector domains *and* the LG domains are both associated (*Figure 1B*). The association of two α13 helices in a dimer requires large-scale structural rearrangements that cannot be explained by known X-ray structures (*Ghosh et al., 2006*). On the pathway to a bridged dimer, there are at least two intermediates - the ligand complex hGBP1:L and the flexible dimer (f-hGBP1:L)$_2$ (*Figure 1B*).

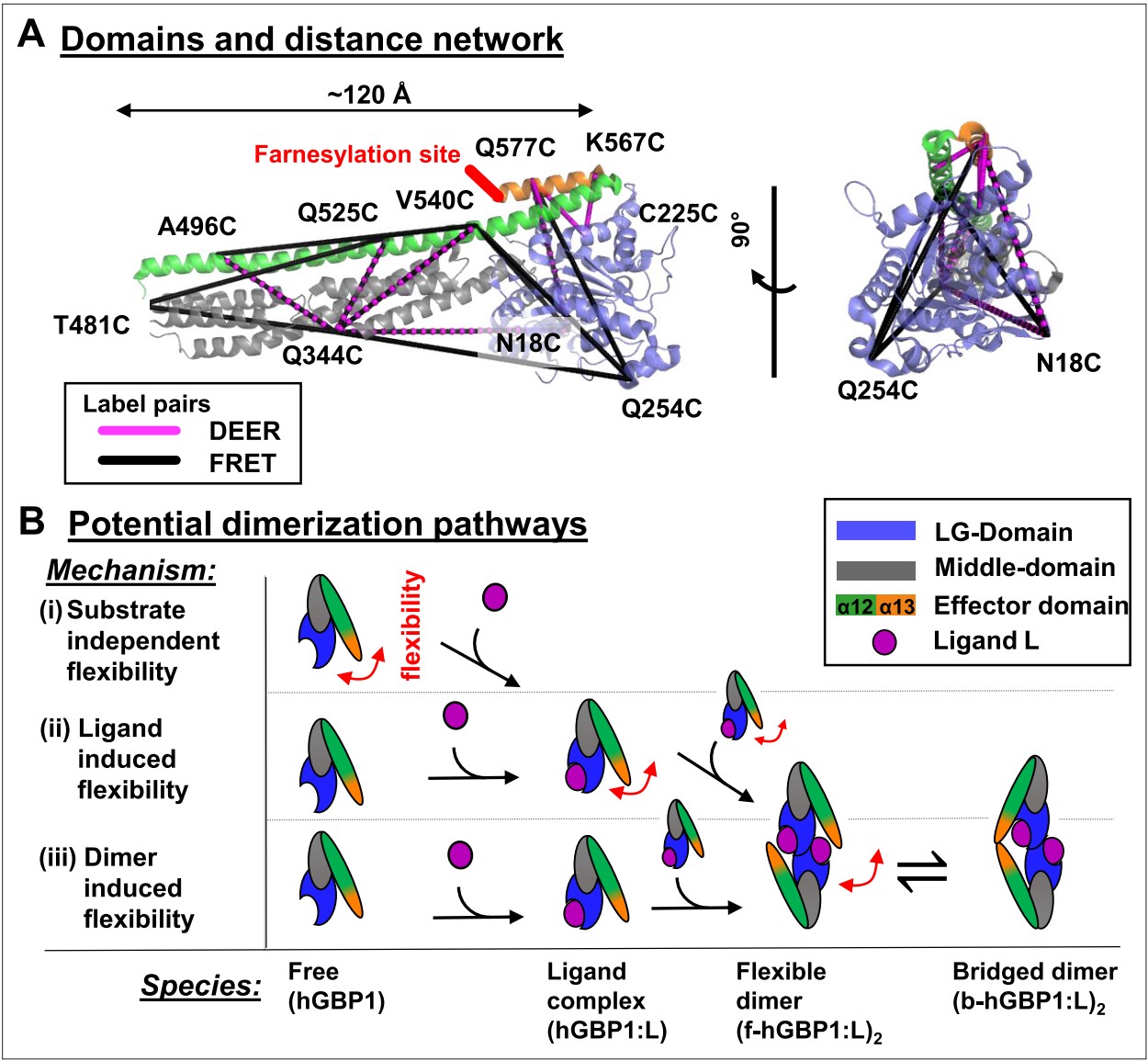

**Figure 1.** DEER and FRET distance network that probes structural arrangement of the human guanylate binding protein 1 (hGBP1) and potential dimerization pathways. (**A**) The network is shown on top of the crystal structure (hGBP1, PDB-ID: 1DG3). hGBP1 consists of three domains: the LG domain (blue), a middle domain (gray) and the helices α12/13 (green/orange). The amino acids highlighted by the labels were used to attach spin-labels and fluorophores for DEER-EPR and FRET experiments, respectively. Magenta and black lines represent the DEER- and FRET-pairs, respectively. In hGBP1 the C-terminus is post-translationally modified and farnesylated for insertion into membranes (red). (**B**) Potential different pathways for the formation of a functional hGBP1 homodimer where the substrate binding LG domains and the helix α13 associate. The association of the helix α13 requires flexibility (red arrows). This flexibility could be induced at different stages of a dimerization pathway.

The online version of this article includes the following figure supplement(s) for figure 1:

**Figure supplement 1.** Structural knowledge on hGBP1.

By probing hGBP1's flexibility (red arrows, *Figure 1B*), we discern the different dimerization paths (black arrow, *Figure 1B*). Either the flexibility is substrate independent (*i*), induced by the ligand (*ii*), or induced by the dimerization (*iii*). In the first path (*Figure 1B, i*), the flexibility is an intrinsic property already present in the absence of substrate, although the flexibility is only needed for the dimerization at a later step. In the second path (*Figure 1B, ii*), the monomer is stiff; the binding and/or the hydrolysis of the substrate in the complex hGBP1:L increases the flexibility for the dimerization. In the third path (*Figure 1B, iii*), GTP binds to hGBP1 to enable the dimerization of the LG domains and the LG domain dimerization triggers a rearrangement for effector domains. The path could be distinguished

if one studies the dynamics of monomeric hGBP1 in the presence and the absence of the substrate. Thus, we map the structure and dynamics of the free and the ligand bound hGBP1.

Experimentally, we map the motions of the monomeric non-farnesylated hGBP1 in the absence and in the presence of the non-hydrolysable ligand GDP-AlF$_x$, corroborated by GTP control experiments. By combining experimental information through integrative modeling, we also resolve hGBP1 structures that explain the molecular prerequisites for dimerization. To generate structures, we use information from small-angle X-ray scattering (SAXS), electron paramagnetic resonance (EPR) spectroscopy by site-directed spin labeling (*Klare and Steinhoff, 2009*), ensemble and single-molecule fluorescence spectroscopy (*Hellenkamp et al., 2018*). smFRET and DEER independently yield distance restraints for modeling, the former with the advantage of being a single-molecule technique that can be applied under ambient conditions, whereas the latter uses a single type of label that is smaller compared to FRET labels, simplifying treatment of the label for modeling purposes. For dynamic information, we apply neutron spin-echo spectroscopy (NSE) and filtered fluorescence correlation spectroscopy (fFCS) (*Felekyan et al., 2012*; *Lerner et al., 2021*). We resolve structures of two new conformational states by integrative modeling and mapped hGBP1's kinetics from nanoseconds to milliseconds. Interrogating conformational dynamics by a network of 12 FRET pairs (*Figure 1A*), we generate a temporal spectrum of hGBP1's internal motions. Finally, we discuss potential implications of the detected protein flexibility and conformers controlling the formation of multimers via an opening like a pocketknife. This allows us to understand the mechanisms excreting the function of this large multi-domain system, that is, the programmed and controlled oligomerization.

## Results

### Experimental equilibrium distributions

We performed DEER, FRET and SAXS experiments to probe short distances, long distances, and molecular shapes, respectively. For the DEER and FRET experiments, we used engineered non-farnesylated hGBP1 cysteine variants (*Figure 1A*) labeled with MTSSL (R1) as spin label and with Alexa488-Alexa647 as FRET pair (Förster radius $R_0$=52 Å), respectively. In SAXS measurements, we studied native non-farnesylated hGBP1 (*Figure 2A*, *Figure 2—figure supplement 1A*). Corroborating results indicate deviations from the non-farnesylated crystal structure (PDB-ID: 1DG3). A Kratky-plot of the SAXS data (*Figure 2A*, middle) visualizes that the non-farnesylated hGBP1 crystal structure 1DG3 disagrees with its structure in solution, which is clearly visible in the weighted residuals in the scattering vector range between 0.05 and 0.2 Å$^{-1}$ showing a significant deviation of the theoretical SAXS curve of 1DG3 from the experimental SAXS data as well as in the large $\chi_r^2$ value of 9.3 that highlights the mismatch between theoretical 1DG3 SAXS curve and experimental data. *Ab initio* modeling of the SAXS data recorded for native non-farnesylated hGBP1 revealed a shape with an additional kink between the LG and the middle domain (*Figure 2A*, right, *Figure 2—figure supplement 1B*), which does not agree with the straight crystal structure of non-farnesylated (PDB-ID: 1DG3) and farnesylated hGBP1 (PDB-ID: 6K1Z; *Figure 2A*).

DEER and FRET experiments on engineered non-farnesylated hGBP1 cysteine variants probed distances between specific labeling sites (*Figure 1A*) - exemplified for the dual cysteine variant Q344C/A496C (*Figure 2B and C*). The inter-spin distances recovered for Q344C/A496C by a model free DEER analysis are clearly shifted by ~2.5 Å towards shorter distances compared to the distances simulated for an X-ray structure of non-farnesylated hGBP1 (PDB-ID: 1DG3) using a rotamer library analysis (RLA) approach (*Polyhach et al., 2011*; *Figure 2B*, right). This shows that the protein exhibits conformations, where the spin-labels come closer to each other than suggested by the crystal structure. Overall, the experimental inter-spin distributions, $p(R_{SS})$, of all eight DEER measurements were unimodal (*Figure 2—figure supplement 2*) and average experimental distances, $\langle R_{SS,exp} \rangle$, differ from the RLA-predicted distances, $\langle R_{SS,sim} \rangle$, by 1.0 Å to 3.6 Å (*Appendix 1—table 1*). The RLA approach does not account for protein backbone dynamics. Thus, we expected to find narrower $p(R_{SS})$ in the simulations compared to the experiments. Yet, for the variants Q344C/Q525C and Q344C/V540C the experimental $p(R_{SS})$ is narrower than the $p(R_{SS})$ predicted by RLA for the crystal structure (*Appendix 1—table 1*). The reduced spread of inter-spin distances is indicative for a reduced conformational freedom of the spin-labeled side chains caused for instance by a denser packing of the spin label(s) with the neighboring side chains and/or the backbone elements than predicted for the crystal

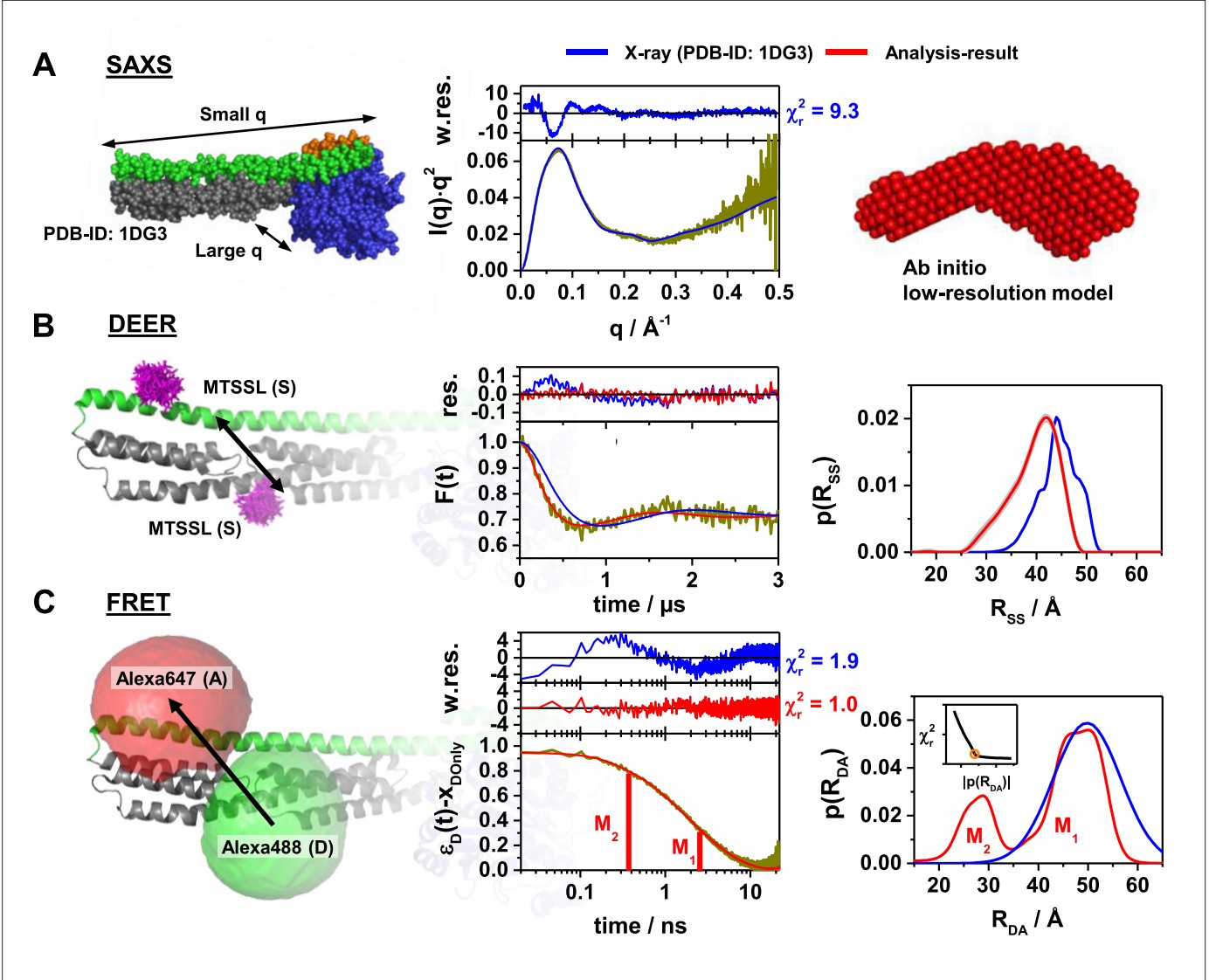

**Figure 2.** Probing the structure of hGBP1 in solution experimentally. The left panels illustrate the characteristic properties probed by the experiments: (A) small angle X-ray scattering (SAXS), (B) double electron-electron resonance spectroscopy (DEER), and (C) Förster resonance energy transfer spectroscopy (FRET). In general, all middle panels display representations of the experimental data (dark yellow curves). The right panels show model-free analysis (red). Predicted experimental data based on a full-length X-ray crystal structure (PDB-ID: 1DG3) are shown in blue. To the top of the experimental curves, either data noise weighted, w.res., or unweighted residuals, res., are shown (middle panels). DEER and FRET experiments sense distances between labels that are flexibly coupled to specific labeling sites (exemplified for the double cysteine variant Q344C/A496C). The time-dependent responses of the sample (middle) inform on the inter-label distance distributions (right panels). The recovered distance distributions are compared to structural models by simulating the spatial distribution of the labels around their attachment point (left panels). The spatial distributions of the MTSSL-labels (B, left), as well as the donor and acceptor dye (C, left), are shown in magenta, green, and red, respectively. All distances resolved by EPR and FRET are compiled in *Appendix 1—table 1* (A) Left: In SAXS the scattered intensity $I(q)$ is measured as a function of the scattering vector $q$. Middle: For better illustration, $I(q)$ is presented in a Kratky-plot. The data are deposited in SASBDB (ID: SASDDD6). Right: SAXS ab initio bead modeling determines an average shape of hGBP1 in solution. (B) Left: The DEER experiments measured the dipolar coupling between two MTSSL spin-labels (magenta). Middle: DEER-traces, $F(t)$, analyzed by Tikhonov regularization (red curve). Right: Recovered inter-spin distance distributions, $p(R_{SS})$. (C) Left: FRET experiments measure the energy transfer from a donor fluorophore (Alexa488, green) to an acceptor fluorophore (Alexa647, red). Middle: Fluorescence intensity decays of the donor analyzed by the maximum entropy method (MEM) recover donor-acceptor distance distributions, $p(R_{DA})$. The inset displays the L-curve criterion of the MEM reconstruction for the presented data set. The FRET-induced donor decay, $\varepsilon_D(t)$, represents the fluorescence decays (*Peulen et al., 2017*). $\varepsilon_D(t)$ is corrected for the fraction of FRET-inactive molecules, $x_{DOnly}$. The shape of $\varepsilon_D(t)$ reveals characteristic times (labeled $M_1$ and $M_2$) that correspond to peaks in $p(R_{DA})$. Right: Recovered inter-label distance distributions for FRET.

The online version of this article includes the following figure supplement(s) for figure 2:

**Figure supplement 1.** Small-angle X-ray scattering measurements on the nucleotide-free hGBP1.

*Figure 2 continued on next page*

*Figure 2 continued*

**Figure supplement 2.** DEER-spectroscopy on a network of MTSSL spin-labeled pairs of the hGBP1 resolves pairwise inter-label distance distributions.

**Figure supplement 3.** Quality controls for labeling based methods.

structure. This can for example be the case if contacts between the molecules in the crystal reorient parts of the structure that are in contact with the label(s) in the solution 'structure'.

FRET experiments using ensemble time-correlated single photon counting (eTCSPC) recovered inter-fluorophore distance distributions, $p(R_{DA})$. The measured data are available in a public data repository (**Data availability**). TCSPC data of donor fluorophore in the presence (DA) and in the absence (D0) of acceptor fluorophores are visualized by $\epsilon_D(t)$, the FRET-induced donor decay. $\epsilon_D(t)$ is the ratio of the fluorescence intensity decay of the donor in the presence, $f_{DD}^{(DA)}(t)$, and the absence, $f_{DD}^{(D0)}(t)$, of FRET (**Peulen et al., 2017**). For mono-exponential $f_{DD}^{(D0)}(t)$ the position (time) and the height (amplitude) of steps in $\epsilon_D(t)$ correspond to DA distances and species fractions, respectively. The variant Q344C/A496C revealed two distances. This is a hallmark for conformational heterogeneity. (**Figure 2C**, center). A model free analysis by the maximum-entropy method (MEM) resolved a bimodal distance distribution $p(R_{DA})$ (**Figure 2C**, right) with a *major* and *minor* subpopulation. The associated conformational states are referred to by $M_1$, and $M_2$ (**Figure 2C**, right). The distances of the states of all 12 data sets (**Figure 1**) were recovered by a joint/global analysis of all measured datasets. In this analysis, we consider distance uncertainty estimates, statistical uncertainties, potential systematic errors of the references, uncertainties of the orientation factor determined by the anisotropy of donor samples, and uncertainties of the AVs due to the differences of the donor and acceptor linker length (**Appendix 2**). We find at room temperature a relative population of 0.61 for $M_1$ and 0.39 for $M_2$ (**Appendix 1—table 1**). A qualitative inspection of fluorescence decay curves can be misleading. Thus, models (see Materials and methods) were selected based on $\chi^2$ and Durbin-Watson tests and posterior model parameters densities were sampled in a Bayesian software framework (ChiSurf) as previously described (**Vöpel et al., 2014**; **Peulen et al., 2017**; **Sanabria et al., 2020**).

We simulate the positional distribution of the dyes by their accessible volume (AV) (**Cai et al., 2007**; **Muschielok et al., 2008**; **Sindbert et al., 2011**) to compare structures and FRET experiments (**Sindbert et al., 2011**; **Kalinin et al., 2012**). In the comparison we considered uncertainty estimates of the experimental distances (**Appendix 2**) and accounted for interactions of the dyes with the protein by the accessible contact volume (ACV) (**Dimura et al., 2016**). The fraction of dyes in an ACV was calibrated by time-resolved anisotropy experiments (**Figure 2—figure supplement 3B**, **Appendix 1—table 1**). Moreover, the anisotropy was used to estimate uncertainties using experimental informed orientation factor distributions (**Dale et al., 1979**). The dyes are only weakly quenched to an extent that is expected for their local environment validating the used model of a mobile dye (**Appendix 1—table 2**). In this case, the $\epsilon_D(t)$ approximation showed to be accurate (**Peulen et al., 2017**). Activity assays show that the dyes and the mutations only weakly affect the protein function (Appendix 2, **Figure 2—figure supplement 3A**). This provides compelling evidence that the distances can be used for structural interpretations.

Distances of $M_1$ agree better with the full length X-ray structure than $M_2$ (**Figure 2C**, right, **Appendix 1—table 1**) - the sum of uncertainty weighted squared deviations, $\chi^2_{FRET}$, for $M_1$ is significantly smaller than for $M_2$ ($\chi^2_{FRET}(M_1, 1DG3) \approx 17$ vs. $\chi^2_{FRET}(M_2, 1DG3) \approx 1500$), confirmed in an F-test with a corresponding p-value >0.999. The variants A496C/V540C and T481C/Q525C designed to test the stability of helix $\alpha$12, revealed identical distances for $M_1$ and $M_2$ (**Appendix 1—table 1**). Thus, we corroborate that helix $\alpha$12 is extended like in previous crystal structures (e.g. PDB-ID: 1DG3). N18C/Q344C and Q254C/Q344C probe distances between the middle- and the LG domain. They revealed only relatively minor differences between $M_1$ and $M_2$. In variants that interrogate motions from the middle domain and the helices $\alpha$12/13 $M_1$ and $M_2$ were significantly different.

To sum up, EPR-DEER at cryogenic temperatures detected small deviations to the crystal structure. SAXS and FRET detected clear deviations at room temperature. To describe the FRET data at least two states are necessary, which are not detected in the DEER experiments most likely due to re-equilibration of the two conformations during sample freezing. Temperature-dependent measurements revealed that these states are also populated at higher physiological temperatures (Appendix 2, **Figure 2—figure supplement 3C**).

## Identification and quantification of molecular kinetics

The distance information of the SAXS, DEER, and FRET experiments provides evidence for a motion of the middle-domain, the LG-domain, and α12/13. We probe the global and the inter-domain motion by single-molecule (sm)FRET experiments with Multiparameter Fluorescence Detection (MFD) and Neutron Spin Echo (NSE) experiments (*Sisamakis et al., 2010*; *Biehl et al., 2011*). The NSE experiments are most sensitive up to a correlation time of 200 ns. The filtered fluorescence correlation spectroscopy (fFCS) of our MFD data is most sensitive from sub-microseconds to milliseconds. Thus, by combining NSE with MFD-fFCS, we effectively probe for conformational dynamics from nano- to milliseconds.

An analysis of the NSE data is visualized in *Figure 3A*, which displays the effective diffusion coefficient $D_{eff}$ extracted from the initial slope of the NSE spectra in dependence of the scattering vector $q$ (*Figure 3—figure supplement 1*). The measured $D_{eff}(q)$ agrees well with the theoretical calculations, accounting for rigid body diffusion alone. The same result was obtained by directly optimizing the parameters of an analytical model describing rigid protein-diffusion (Materials and methods, *Equation 9*) to the NSE spectra (*Figure 3—figure supplement 1B*) by $\Delta D_{eff} \approx u^2/\tau$ with MSD $u^2$ and relaxation time $\tau$ of internal protein dynamics. Hence, a reasonably large $u^2$-value could in principle be compensated by a long relaxation time $\tau$ of internal dynamics leading to small $\Delta D_{eff}$-values. To test this scenario and to assess potential contributions of internal protein dynamics to rigid-body diffusion, we consider a full analytical model in Materials and methods *Equation 8* and *Equation 9* and examine the intermediate scattering functions $I(q, t)$ (*Figure 3—figure supplement 1A*). The additional contribution at short times due to internal dynamics can be estimated by a Debye-Waller argument (*Biehl et al., 2008*): Internal protein dynamics with a MSD of $u^2$ leading to a change in the $I(q, t)$ within the observed errors (errors ~0.007 < 0.01) can be estimated by $\frac{-u^2 q^2}{3} = ln(1 - err.)$. If we consider the deviation at $Q = 0.5 nm^{-1}$ then we obtain a value of $u = 0.25\ nm$. Compared to the size of the hGBP1 protein internal motions with such small amplitudes are not observable by NSE. A significant additional contribution of internal protein dynamics to the measured effective diffusion coefficients cannot be identified. Hence, the overall internal protein dynamics may only result in negligible amplitude, that is, minor overall shape changes, within the observation time up to 200 ns.

To cover sub-μs to ms dynamics, we performed MFD smFRET experiments on freely diffusing molecules. We determine for every molecule the average fluorescence lifetime of the donor, $\langle\tau_{D(A)}\rangle_F$, and the FRET efficiency, $E$, to create MFD-diagrams that visualize heterogeneities among the molecules. MFD diagrams correlate calibrated intensity-based observables to the fluorescence lifetimes for revealing conformational heterogeneity. For FRET efficiencies, we calibrate our instrument by DNA reference samples as previously described (*Hellenkamp et al., 2018*) and account for sample-specific dark-states using fluorescence decay and FCS measurement of single-labeled samples (*Appendix 1—table 3*). In MFD-diagrams, 'static FRET-lines' serve as a reference to detect fast conformational dynamics (*Kalinin et al., 2010*). In case of sub-millisecond hGBP1 dynamics, we expect to observe multimodal distributions (*Barth et al., 2022*; *Opanasyuk et al., 2022*). Analogous to NMR relaxation dispersion experiments, a peak shift from the static FRET line towards longer $\langle\tau_{D(A)}\rangle_F$ is evidence for conformational dynamics faster than the observation time (~ms) (*Kalinin et al., 2010*; *Sisamakis et al., 2010*). All 12 FRET variants had single FRET peaks in the 2D-histograms (*Figure 3B*, *Figure 3—figure supplement 2*). In 8 out of 12 variants the peak was significantly shifted, a clear indication of dynamics (*Figure 3—figure supplement 2*). An analysis of the fluorescence decays of the FRET sub-ensembles (*Figure 3—figure supplement 3*) by a two-component model (see Materials and methods) revealed limiting states (*Appendix 1—table 5*) that agree with the eTCSPC data (*Appendix 1—table 1*). The MFD peak positions (*Figure 3—figure supplement 2*) are consistent with the eTCSPC data (*Figure 3B*, *Figure 3—figure supplement 3*). This is additional evidence for conformational heterogeneity and sub millisecond dynamics.

To quantify the dynamics, we performed filtered FCS (fFCS) and jointly analyze all species cross-correlation functions (sCCF) and the species autocorrelation functions (sACF) by a single model (*Figure 3—figure supplement 4*) to determine characteristic times (*Appendix 1—table 6*). For a pure two state system ($M_1 \leftrightharpoons M_2$), we expected to find a single characteristic time. However, at least two relaxation times with corresponding amplitude were required to describe individual fFCS datasets (36 relevant free parameters). Thus, there are more than two (kinetic) states. To compare the relaxation times across FRET variants, and to reduce the number of free parameters we performed a

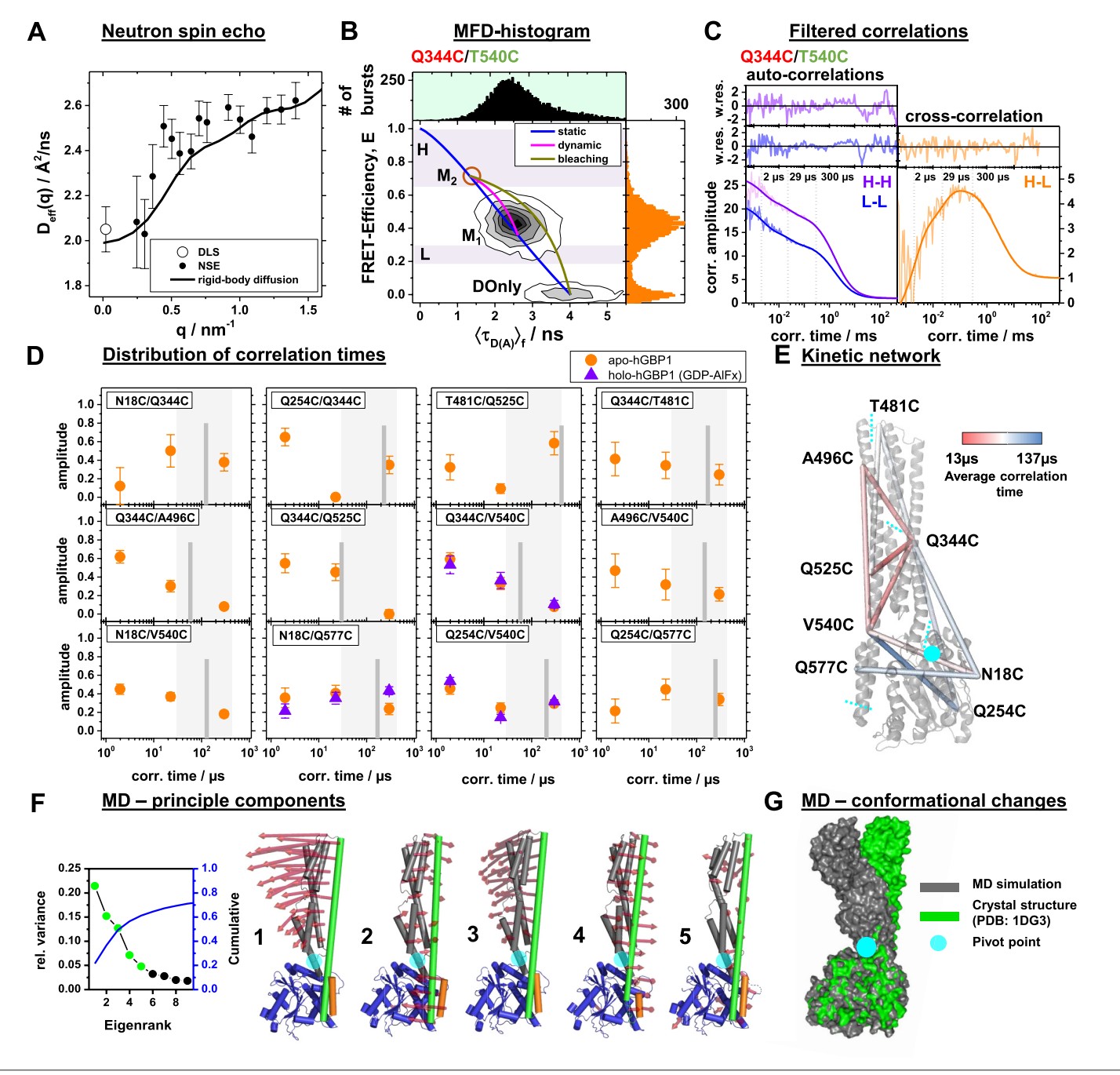

**Figure 3.** Conformational dynamics of hGBP1 studied by neutron spin echo (NSE), single molecule (sm) FRET with multi-parameter fluorescence detection (MFD), and molecular dynamics (MD) simulations. (**A**) Effective diffusion coefficients of hGBP1, $D_{eff}$, determined by NSE and dynamic light scattering (DLS) compared to a model describing only the rigid body translational and rotational diffusion as a function of the scattering vector, $q$. The agreement of the experimental and calculated diffusion coefficients demonstrates insignificant shape changes on fast time scales up to 200 ns. (**B**) Two-dimensional single-molecule histogram of the absolute FRET-efficiency, $E$, and the average fluorescence weighted lifetimes of the donor in the presence of FRET, $\langle \tau_{D(A)} \rangle_f$, of the double cysteine variant Q344C/V540C. One-dimensional histograms are projections of the 2D histogram. The color of the variant's name indicates the location of the donor (green) and acceptor (red) determined by limited proteolysis and time-resolved anisotropies. The static-FRET line (blue) relates $E$ and $\langle \tau_{D(A)} \rangle$ for static proteins. The dynamic FRET line (magenta) describes molecules that change their conformation from $M_1$ to $M_2$ (brown circle) and *vice-versa* while being observed. The $\langle \tau_{D(A)} \rangle$- $E$ diagrams of all variants are compiled in *Figure 3—figure supplement 2*. $M_1$ and $M_2$ were identified by eTCSPC (*Appendix 1—table 1*) and sub-ensemble TCSPC (*Figure 3—figure supplement 3*, *Appendix 1—table 5*). Molecules in $M_2$ with bleaching acceptors are described by a bleaching line (dark yellow) that describes the transitions from $M_2$ to the donor only population (DOnly). Photons of molecules in the H and L area (H - high FRET, L - low FRET) were used to generate filters for filtered FCS (fFCS).

*Figure 3 continued on next page*

*Figure 3 continued*

(**C**) fFCS species autocorrelation functions (s*ACF*) and species cross-correlation function (s*CCF*) of the variant Q344C/T540C (semitransparent lines) and corresponding model functions (solid lines) (Materials and Methods, **Equation 17**). The fFCS model parameters were determined by a global analysis of all 12 FRET-pairs (**Figure 3—figure supplement 4**) and revealed three correlation times (vertical dotted lines). The weighted residuals are shown to the top. The filter setting for fFCS of all samples and the fit results are compiled in **Appendix 1—table 6**. (**D**) Amplitudes of the fitted fFCS correlation times of the GTP free apo- (orange circles) and GDP-AlF$_x$ bound holo-state (violet triangles) (values see **Appendix 1—table 6**). The average correlation times for the variants are shown as gray vertical lines. The gray boxes highlight the minimum and maximum of the average correlation times. (**E**) The average correlation times of the apo-state are mapped color-coded to a crystal structure (PDB-ID: 1DG3). Sections of the five rigid elements are displayed by cyan dashed lines. (**F**) Principle components analysis (PCA) of molecular dynamics (MD) and accelerated molecular dynamics (aMD) simulations (Materials and methods). The LG domain, the middle domain, and α12, and α13 are colored in blue, gray, green, and orange, respectively. The red arrows indicate the direction of the motion (scaled by a factor of 1.5 for better visibility). The semi-transparent cyan circle corresponds to a pivot point. The first five principal components (PCs), sorted by the magnitude of the eigenvalues, contribute to 60% of the total variance of all simulations. (**G**) Superposition of a MD trajectory frame (gray) deviating the most in RMSD (~8 Å) from the crystal structure (green). Both structural models were aligned to the LG domain.

The online version of this article includes the following figure supplement(s) for figure 3:

**Figure supplement 1.** Neutron spin echo spectroscopy (NSE) on the hGBP1 resolves internal dynamics on the nanosecond timescale.

**Figure supplement 2.** Single-molecule fluorescence measurements.

**Figure supplement 3.** Sub-ensemble fluorescence decays $f^{(species)}_{Em.|Exc.}$ of single-molecule FRET measurements on different FRET-labeled (Alexa488, Alexa647) variant of hGBP1.

**Figure supplement 4.** Global analysis of filtered fluorescence correlation spectroscopy of FRET-labeled variants for the hGBP1 probing its internal dynamics from µs to ms.

**Figure supplement 5.** Single-molecule fluorescence measurements under dimer conditions and in the presence of nucleotides.

global analysis of the fFCS data (Materials and Methods, **Equations 17–19**). In global analysis, local and global parameters are simultaneously optimized (**Beechem et al., 2002**). Global parameters are varied parameters shared across datasets. Local parameters are varied parameters of a single dataset. In our analysis, relaxation times were global parameters (3 relaxation times, shared across FRET variants), corresponding amplitudes were local parameters (2 amplitudes per FRET pair) of FRET variants. Model parameters and uncertainties were determined by optimizing and sampling over local and global parameters using the sum of all weighted squared deviations computed for all 48 model and experimental fFCS curves as objective function. **Figure 3—figure supplement 4** display the experimental fFCS and model fFCS curves computed for the global analysis result (**Appendix 1—table 6**). This analysis recovered three correlation times (2, 23 and 297 µs) with significantly varying amplitudes (**Figure 3D**, **Appendix 1—table 6**) and average relaxation times varied approximately (gray bars in **Figure 3D**). In most cases, the shortest component has the highest amplitude. This is consistent with the MFD-diagrams because we detected shifted/dynamic unimodal peaks. We mapped the average correlation times color coded to the FRET network (**Figure 3E**). This highlights that the fast dynamics is associated to α12/13 and the middle domain while the slow dynamics is predominantly linked to the LG domain. Referring to the sketch in **Figure 1B**, we hypothesize that the states M$_1$ and M$_2$ and the transition among them are of functional relevance (pathway *i*). Therefore, we studied the effect on the dynamics exerted by the ligand GDP-AlF$_x$ as a non-hydrolysable substrate that mimics the holo-state hGBP1:L. The GDP-AlF$_x$ concentration was sufficiently high (100 µM) to fully induce dimerization of hGBP1 at µM concentrations (**Kravets et al., 2016**). For comparison, the affinity of hGBP1 for mant-GDP is ~3.5 µM and much higher for GDP-AlFx (**Praefcke et al., 1999**). MFD control experiments performed on hydrolysable GTP agree with the non-hydrolysable GDP-AlF$_x$ (**Figure 3—figure supplement 5B**). Hence, in the sm-measurements GDP-AlF$_x$ was bound to the LG domain while the non-farnesylated hGBP1 (20 pM) was still monomeric. We refer to this as the holo-form of the protein and selected a set of variants (N18C/Q577C, Q254C/V540C, Q344C/V540C) for which we found large GDP-AlF$_x$ and GTP induced effects at higher hGBP1 concentrations because of oligomerization (**Figure 3—figure supplement 5A**). Surprisingly, the amplitude distribution is within errors indistinguishable from the measurements of the nucleotide-free apo forms (**Figure 3D**). Moreover, the FRET observables changed neither.

We found for hGBP1 in solution a conformationally heterogeneous ensemble that can be approximated by conformers M$_1$ and M$_2$ (TCSPC), no significant shape changes of non-farnesylated hGBP1 on a timescale up to 200 ns (NSE), and complex kinetics spanning the µs-range mainly associated to

α12/13 and the middle domain (fFCS) that is unaffected by a nucleotide analog as a substrate. Based on the distance and the dynamic information, we propose a complex motion of α12/13 relative to the LG and the middle domain and additional intermediate conformational states, captured by fFCS through their kinetic fingerprint.

## Essential motions determined by molecular dynamics simulations

We performed molecular dynamics (MD) simulations without experimental restraints to (*i*) identify functional elements of non-farnesylated hGBP1 in the presence and the absence of GTP, (*ii*) to assess the structural dynamics of the full-length crystal structure at the atomistic level, and (*iii*) to capture potential motions of non-farnesylated hGBP1 (Materials and methods). The apo (PDB-ID: 1DG3) and a GTP bound holo-form of non-farnesylated hGBP1 were simulated in three replicas by conventional MD simulations for 2 μs each (*Figure 4—figure supplement 1A*). Additionally, accelerated molecular dynamics (aMD) simulations, which samples free-energy landscapes of a small protein approximately 2000-fold more efficiently (*Pierce et al., 2012*), were performed in two replicas of 200 ns each to enhance the conformational sampling. An autocorrelation analysis of the RMSD determined for the conventional MD simulations *vs.* the average structure of the MD simulations reveals fast correlation times. The average correlation time in the presence and the absence of GTP were 11 ns and 17 ns, respectively (*Figure 4—figure supplement 1B*). The amplitude of the fluctuations is, on average, below an RMSD of 3 Å, and could thus not be resolved by our NSE experiments. In the MD simulations, larger conformational changes (RMSD >7 Å) with considerable shape changes were very rare events. Consistent with previous computational observations (*Barz et al., 2019*; *Figure 4—figure supplement 1C*), a principal component analysis revealed kinking motions of the middle domain and helix α12/13 around a pivot point as most dominant motions in the MD simulations (*Figure 3F*). A visual inspection of structures deviating most from the mean reveals a kink at the connector of the LG and the middle domain (*Figure 3G*) consistent with rearrangements required for average shape as revealed by SAXS (*Figure 2*).

To sum up, the MD simulations cover timescales of a few microseconds, show potential directions of motions, and identified a pivot point between the LG and the middle domain. In agreement with NSE on the simulation timescale, the overall shape is mainly conserved, and large conformational changes are rare events. The helices α12/13 were mobile and exhibited a limited 'rolling' motion along the LG and middle domain that could connect the conformers $M_1$ and $M_2$ as suggested by our FRET studies.

## Experimentally guided structural modeling

Altogether, the SAXS, NSE, EPR, and FRET measurements give a unified and consistent view on hGBP1 conformational dynamics. As recommended in *Lerner et al., 2021*, the assessment of sample properties and function (effects of mutations and labeling on enzyme properties, temperature effects of the conformational equilibrium), the estimation of the uncertainty of the determined interlabel distances and the consistency check between distinct measurement methods in **Appendix 2** provide confidence that our data are suitable for quantitative integrative structural modeling.

Thus, we used the obtained structural experimental information, the kinetics, the MD simulations, and the prior structural information provided by existing crystal structures to create new structures for $M_1$ and $M_2$ by experimentally guided structural modeling. Previously, we developed approaches for integrative modeling using FRET data (*Sindbert et al., 2011*; *Kalinin et al., 2012*) that could successfully resolve three short-lived conformational states of proteins in two benchmark studies: (1) For a large GTPase with synthetic simulated data (*Dimura et al., 2016*) and (2) the enzyme Lysozyme (T4L) of the bacteriophage T4 with experimental ensemble and single-molecule data (*Dimura et al., 2020*; *Sanabria et al., 2020*). Here, we extended this framework to incorporate DEER and SAXS data. The framework combines the experimental data in meta-analysis via their information content. A detailed description of our integrative modeling can be found in Materials and methods and Appendix 3.

In a nutshell, we generate quantitative structures in three major steps (*Figure 4A*): (*i*) '*Data acquisition*' (steps 1–3), (*ii*) '*Model generation*' (steps 4–5), and (*iii*) '*Model discrimination*' (steps 6–7). In our previous benchmark study 29 optimal chosen FRET pairs achieved an accuracy and a precision below 2 Å for a similar large GTPase (*Dimura et al., 2016*). Here, given only 12 FRET and 8 DEER pairs we

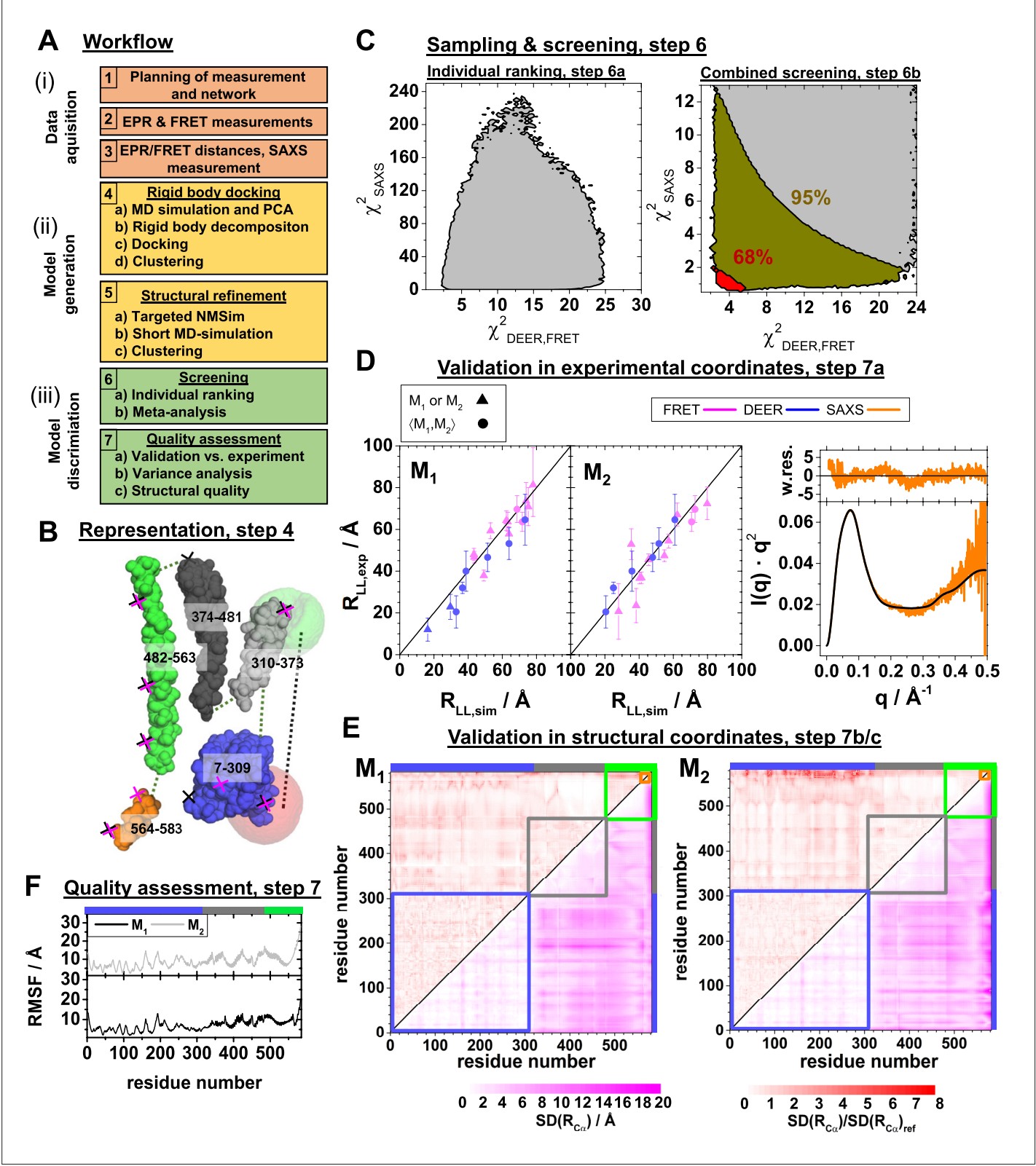

**Figure 4.** Integrative modeling workflow and structure validation. A detailed description and the used data can be found in **Appendix 3**. (**A**) The workflow combines rigid body docking (RBD), structural refinements, and molecular dynamics (MD) simulations. Rigid bodies (RBs) are identified by MD simulations and principal components analysis (PCA) (Materials and methods). (**B**) RBD representation of hGBP1: LG-domain (blue), the middle domain (gray), helix α12 (green), helix α13 (orange). The numbers correspond to the RB amino acid ranges. The crosses mark the FRET (black) and

*Figure 4 continued on next page*

*Figure 4 continued*

the EPR (magenta) labeling positions. The RBD considers the label distribution illustrated for a FRET pair by semi-transparent green (donor) and red (acceptor) surfaces. (**C**) Left: outline of $\chi^2_{SAXS}$ (Appendix 3, **Equation 27**) and $\chi^2_{DEER,FRET}$ (Appendix 3, **Equation 28**) for all ($M_1$, $M_2$) pairs of structures (left). Confidence levels of the meta-analysis (Materials and methods, **Equation 20**) that discriminates ($M_1$, $M_2$) pairs (right). Red and dark yellow regions correspond to p-values smaller than 0.68 and 0.95, respectively. (**D**) Experimental validation of the best pair of structures. Comparison of experimental $R_{LL,exp}$ (for DEER $\langle R_{SS,exp} \rangle$ and for FRET $\bar{R}_{DA,exp}$) and modeled label distances $R_{LL,sim}$ (for DEER $\langle R_{SS,sim} \rangle$ and FRET $\bar{R}_{DA,sim}$). Specific symbols display label distances $R_{LL,exp}$ for label pairs with distinct (▲) and equal (●) values for $M_1$ and $M_2$, respectively (see **Appendix 1—table 1**). For SAXS the scattering curve (black line) of the structure pair ($M_1$, $M_2$) is compared to the experimental data (orange line) by the weighted residuals to the top. (**E**) The standard deviation, SD, of the pairwise $C_\alpha$-$C_\alpha$ distance $SD(R_\alpha)$ of the experimental ensemble with a p-value <0.68 (lower triangles) highlights the structural uncertainty. $SD(R_\alpha)$ normalized by the $SD(R_\alpha)_{ref}$ computed by the experimental uncertainty validates the structures. (**F**) Root mean square fluctuations (RMSF) of the $C_\alpha$ atoms of structures with a p-value <0.68 are displayed for the globally aligned ensemble.

The online version of this article includes the following figure supplement(s) for figure 4:

**Figure supplement 1.** Analysis of molecular dynamics simulations, conformational space, identification of flexible regions, ensemble selection.

**Figure supplement 2.** Analysis of structure generation and discrimination.

expect to recover structures with an average RMSDs of 8–15 Å. We mainly aim to resolve molecular shapes, domain arrangement, and topologies.

We generate new structures (**Figure 4A**, steps 4–5) by sampling the conformational space of a coarse grained (cg) hGBP1 representation using FRET and DEER restraints. The representation (**Figure 4B**) is based on an order-parameter based rigidity analysis (**Figure 4—figure supplement 1D**), knowledge on the individual domains (**Low and Löwe, 2010**; **Chen et al., 2017**). It can reproduce the motion of the MD simulations (**Figure 3F**). For maximum parsimony, the DEER, FRET, and SAXS data were described by pairs of the structures ($M_1$, $M_2$) ranked by their agreement with SAXS, and DEER, FRET using $\chi^2_{SAXS}$ and $\chi^2_{DEER,FRET}$, respectively (**Figure 4C**; Appendix 3). The pair best agreeing with SAXS has a middle domain kinked towards the LG domain. A SAXS ensemble analysis revealed species population fractions for $M_1$ between ~0.1–0.7 (**Figure 4—figure supplement 1E**, p-value = 0.68). A meta-analysis by Fisher's method jointly scores pairs of structures considering all available data (**Figure 4A**, step 6b) and estimates for the effective degrees of freedom (dof) of the representation and the experiments (**Figure 4C**). A stability test demonstrates that varying the dofs has a minor influence on the structure (**Figure 4—figure supplement 2A**). A combined p-value of 0.68 discriminates 95% of all ($M_1$, $M_2$) pairs (**Figure 4C**, red area; **Figure 4—figure supplement 2B**) leaving models with average RMSDs of 11.2 Å and 14.5 Å for $M_1$ and $M_2$, respectively. The uncertainties are largest for α12/13 (**Figure 4E**). The pair of structures are validated for DEER and FRET comparing experimental and modeled average distances (**Figure 4D**, *left*). This comparison identified initial assignment outliers (Appendix 2, **Figure 2—figure supplement 3**). For SAXS, pairs of structures are compared by computed scattering curves (**Figure 4D**, right). This comparison demonstrates that the integrative structures capture the essential features of the experiments and that the data recorded on non-farnesylated hGBP1 cysteine variants for FRET/DEER is consistent with SAXS data recorded on native non-farnesylated hGBP1.

The standard deviation of pairwise $C_\alpha$ distances, $SD(R_{C_\alpha})$, reveals alignment free regions of low and high variability (**Figure 4E**, lower triangles). To check if the variability exceeds the expectances based on experimental precision, $SD(R_{C_\alpha})$ is normalized by computing a weighted (normalized) precision, $SD(R_{C_\alpha})/SD(R_{C_\alpha})_{ref}$ (**Figure 4E** upper triangles). The reference $SD(R_{C_\alpha})_{ref}$ is the precision of "ideal and perfect" model ensembles, determined using the experimental uncertainties under the assumption, that the best experimental determined model is the ground truth. For $M_1$, this procedure yields a distribution for the weighted precision of the recovered structural models that fluctuates around unity, the theoretical optimum (**Figure 4E**, left). The weighted precision for $M_2$ close to the C-terminus (end of helix α12 and α13) is lower than expected (**Figure 4E**, right), presumably due to granularity of the model or systematic experimental errors. The heterogeneity of the structural ensembles judged by their root-mean-squared-fluctuations (RMSF) is in the expected range of ~7 and~9 Å, for $M_1$ and $M_2$ respectively (**Figure 4F**). We deposited the conformational ensemble with all meta data at the prototype archiving system PDB-Dev with the ID: PDBDEV_00000088.

To visualize differences among the structural models, we aligned the selected conformers to the LG domain. This demonstrates that in $M_1$ and $M_2$ α12/13 binds at two distinct regions of the LG domain (**Figure 5A**, red spheres). In $M_1$, α12/13 binds to the same side of the LG domain as in the known

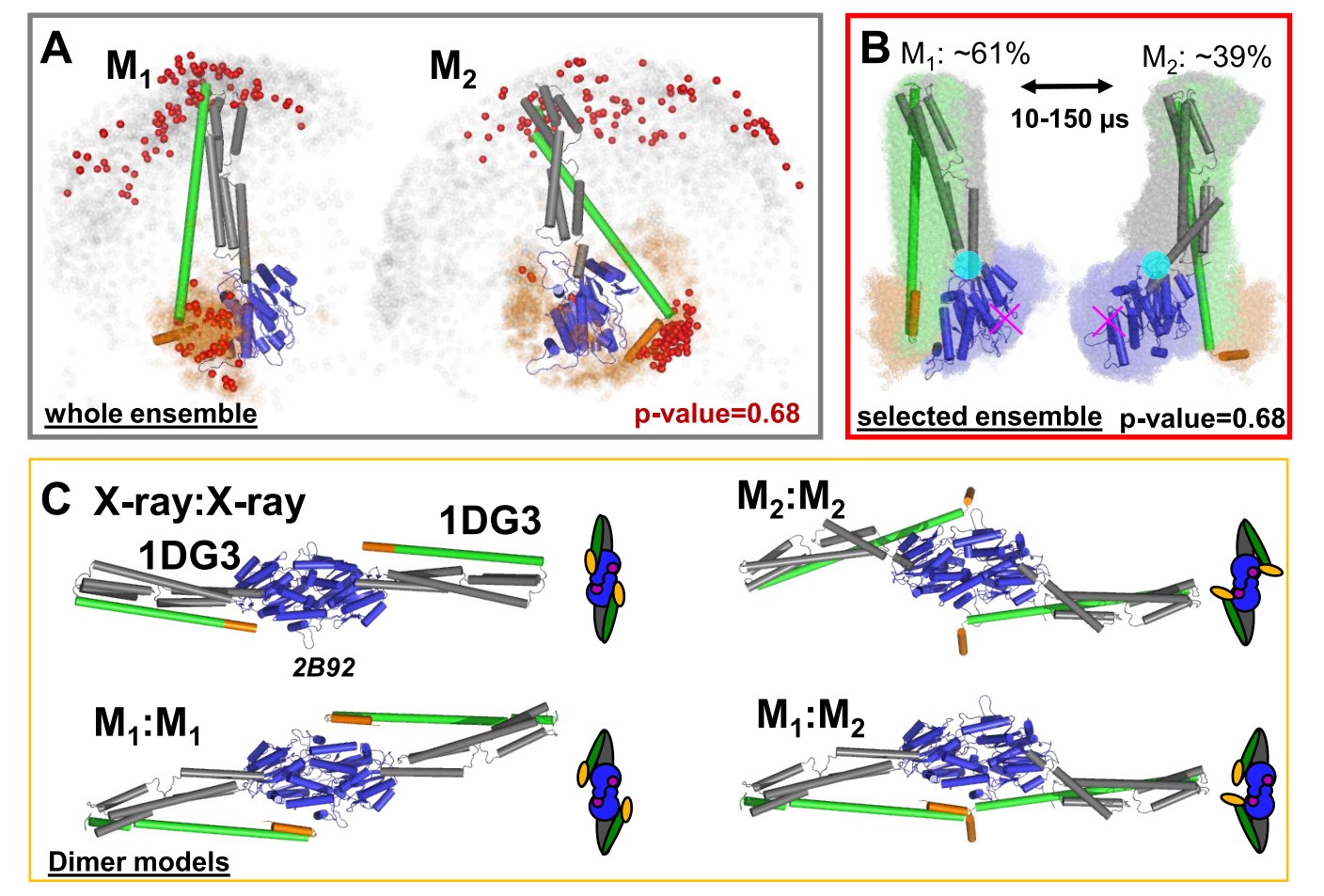

**Figure 5.** Selected conformers and corresponding dimer models based on integrative modeling structures using of DEER, FRET, and SAXS data. (**A**) All structures for M$_1$ and M$_2$ were aligned to the LG domain and are represented by orange and gray dots, indicating the C$_\alpha$ atoms of the amino acids F565 and T481, respectively. The structures best agreeing with all experiments are shown as cartoon representation (ribbon presentation see ***Figure 5— figure supplement 1***). Non-rejected structures (p-value = 0.68, ***Figure 4—figure supplement 1E***) represented by red spheres. The ensemble has been at deposited at PDB-Dev with the ID: PDBDEV_00000088. (**B**) Global alignment of all selected structures (p-value = 0.68). In the center, the structures best representing the average of the selected ensembles are shown. The transition from M$_1$ to M$_2$ (average correlation times 10–150 μs) can be described by a rotation around the region connecting the LG with the ligand binding site (magenta cross) and the middle domain (cyan circle). (**C**) Potential hGBP1:hGBP1 dimer structures constructed by superposing the head-to-head interface of the LG domain (PDB-ID: 2B92) to the full-length crystal structure (1DG3). The LG and middle domain are colored in blue and gray, respectively. Helices α12 and α13 are colored in green and orange, respectively.

The online version of this article includes the following figure supplement(s) for figure 5:

**Figure supplement 1.** Selected conformers models based on integrative modeling structures using of DEER, FRET, and SAXS data.

crystal structure (PDB-ID: 1DG3). In M$_2$, α12/13 binds to the opposing side of the LG domain. In a global alignment of the M$_1$ and M$_2$ structures, the best representatives of the ensembles visualize the transition between M$_1$ and M$_2$. A rearrangement of residues 306–312 results in a rotation of the middle domain around a pivot point (***Figure 5B***, cyan circle) and describes the experimental data. The relocation of α12/13 agrees well with global motions identified by PCA of the MD simulations (***Hamelberg et al., 2004***). In the transition from M$_1$ to M$_2$ α12/13 'rolls' along the LG domain, while the middle domain rotates and kinks towards the LG domain. M$_1$ is comparable to the crystal structure except for a kink of the middle towards the LG domain, the movement of α12/13 stops on the opposite side of the LG domain.

## Discussion

In non-farnesylated hGBP1 we found two conformations ($M_1$ and $M_2$), determined corresponding structures by integrative modeling, and mapped the $M_1$/$M_2$ exchange dynamics by NSE spectroscopy and fFCS. NSE showed no shape changes on the ns-timescale up to 200 ns. fFCS on a network of FRET-pairs revealed considerable dynamics on slower time scales (2–300 µs, *Figure 3*). The distribution of dynamics over such a wide range is indicative of a frustrated/rugged potential energy landscape with several substates and multiple kinetic barriers. Structural models for $M_1$ and $M_2$ based on SAXS, DEER, and FRET data revealed that the middle domain kinks towards the LG domain and that the helices α12/13 are bound on opposite sides of the LG domain. Notably, largest relative changes in DA distances are correlated with the fastest relaxation time (*Figure 3D*, *Appendix 1—table 6*). These findings are self-consistent, as the conformational transition from $M_1$ to $M_2$ and vice versa is complex and may cause a distribution of relaxation times, indicating a rough energy landscape with several intermediates, and the dynamics is mainly associated to α12/13. Analogous to protein folding, where (*Chung et al., 2012*) monitored the transition from the unfolded to the folded state and defined a transition path time, it would be intriguing to define an effective time for the conformational transition from $M_1$ to $M_2$. The conformational transition time would be a convolute of all observed relaxation times (*Figure 3*, *Appendix 1—table 6*) that is expected to be in the sub-millisecond time range. To sum up, the experiments can be described by two conformational states separated by a rugged energy landscape, resulting in slow transition invisible on the NSE timescale. The smFRET measurements demonstrate that this transition is an intrinsic property of non-farnesylated hGBP1 that does not depend on the presence of substrate (pathway (i) in *Figure 1B*).

To understand the functional relevance of $M_1$ and $M_2$, various observations and existing experimental information on farnesylated hGBP1 must be considered. We previously speculated that the farnesyl anchor acts as a 'safety latch' that attaches α12/13 to the LG domain. Nevertheless, we identified monomeric as well as dimeric forms of farnesylated and non-farnesylated hGBP1 by SEC-SAXS that both require large structural rearrangements (*Lorenz et al., 2020*). Thus, the dimerization, as the first step in oligomerization of hGBP1, is a feature that demands flexibility of the structure as deduced from major structural rearrangements described so far (*Vöpel et al., 2014*; *Ince et al., 2017*; *Shydlovskyi et al., 2017*). In particular, large movements of the LG, the middle domain and helices α12/13 against each other are required to establish the elongated building blocks of the polymer (*Shydlovskyi et al., 2017*). It is also most conceivable that multiple dynamically interchanging configurations of the sub-domains need to be sampled to assemble the highly ordered protein. Dynamins and farnesylated hGBP1 form highly ordered oligomers (*Shydlovskyi et al., 2017*) requiring at least two binding sites. We previously showed that non-farnesylated hGBP1 forms dimers via the LG domains (in a head-to-head manner) *and* via helix α13 (*Vöpel et al., 2014*) in the presence of a GTP analog. This finding is inconsistent with non-farnesylated nucleotide free (PDB-ID: 1DG3), nucleotide bound (PDB-ID: 1F5N) and farnesylated nucleotide free (PDB-ID: 6K1Z) full-length crystal structures. In dimers formed by two hGBP1s in a 1DG3, 1F5N, or 6K1Z conformation the helices α13 are on opposite sides and thus could not be associated (*Figure 5*). Similar findings were recently published for hGBP5 (*Cui et al., 2021*), showing that the middle domain undergoes a drastic movement after GTP binding, forming a closed dimer. However, in a dimer formed of two distinct conformers ($M_1$:$M_2$), the helices α13 are located on the same side of their LG domains. Thus, GTP binding likely leads to dimerization because of the increased affinity between the LG domain. In the formed dimer the low affinity between the α12/13 helices and the middle domain suffices to induce opening like a pocket knife which is the prerequisite for oligomerisation. In line with previous studies, which identified preferred pathways to increase the association yield of protein-protein complexes (*Kozakov et al., 2014*), we suggest dimerization path (i) as mechanism for dimerization of non-farnesylated hGBP1 (*Figure 1B*), that is, owing to the substrate independent conformational flexibility, precursors necessary for oligomerization are already formed spontaneously before binding of the oligomerization-inducing substrate GTP. Remarkably, we detected virtually no substrate induced differences in the amplitude distribution of the correlation times demonstrating that the flexibility is independent of the bound nucleotide. Overall, the findings strongly suggest that the GTP induced dimerization of the GTPase domains, and a substrate independent flexibility are needed for a dimerization of the effector domains (pathway *i* in *Figure 1B*). The substrate solely facilitates hGBP1 association by increasing the affinity of the LG domain as a hub for dimerization.

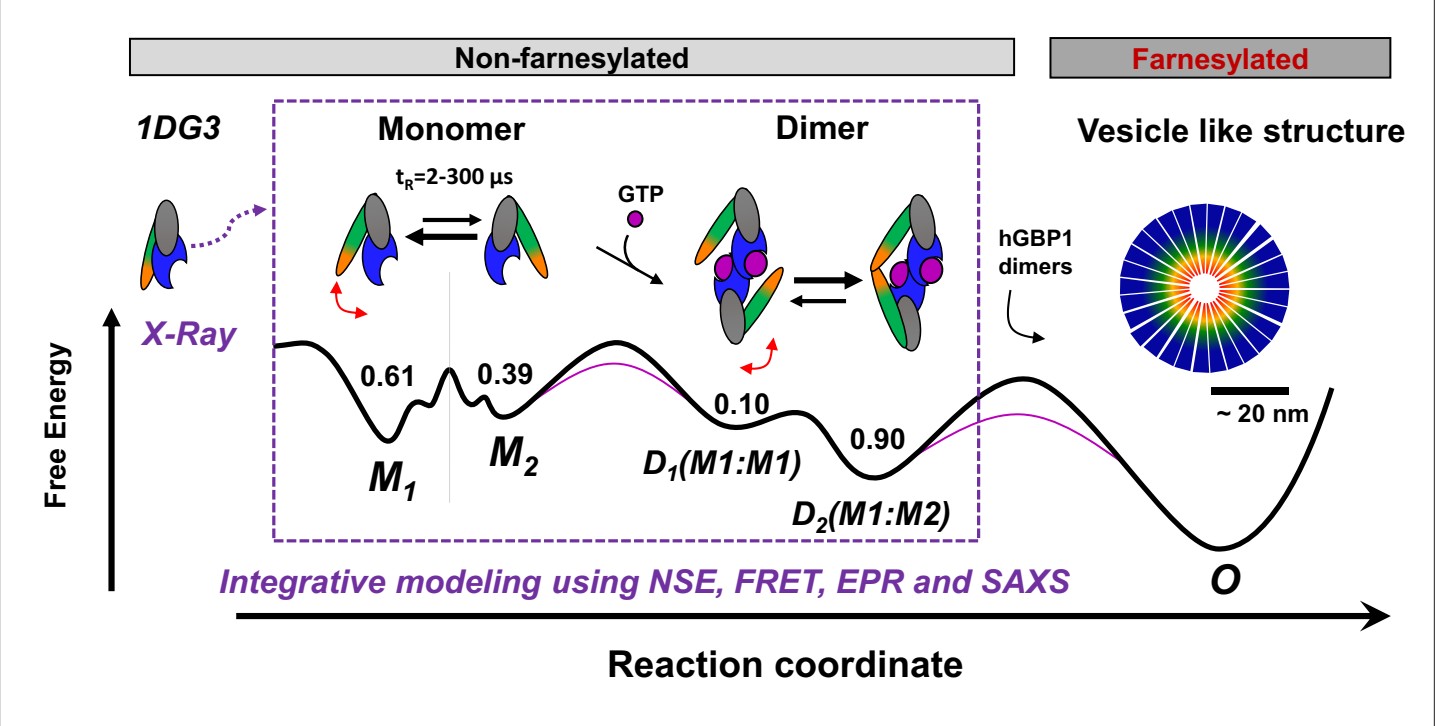

**Figure 6.** Potential oligomerization pathways of the human guanylate binding protein 1 (hGBP1) summarizing current experimental findings (*Kravets et al., 2012*, *Vöpel et al., 2014*; *Kravets et al., 2016*; *Shydlovskyi et al., 2017*). In the presence and absence of a nucleotide, hGBP1 is in a conformational exchange with a Pivot point between LG and middle domain resulting in at least two conformational states $M_1$ and $M_2$ with a correlation time of 2–300 µs. Binding of a nucleotide to the LG domain activates hGBP1 for dimerization. After hGBP1 dimerization via the LG domains conformational changes of the middle domains and the helices α12/13 lead to an association of both helices α13. The species fractions for respective populations are given as numbers on top of the wells of a schematic energy landscape (black line). The substrate GTP lowers the activation barrier (red line). Under turn-over of GTP, farnesylated hGBP1 further self-assembles to form highly ordered, micelle-like polymers.

Structure-wise, we found that the middle domain is kinked towards the LG domain as found for other dynamins (*Low and Löwe, 2010*; *Chen et al., 2017*). Moreover, our data supports two conformations with distinct binding sites of helix α12/13 that can be explained by major rearrangements of the region connecting the middle and the LG domain. *Prakash et al., 2000* described already the interconnecting region of LG and middle domain, which comprise residues 279–310 including a small β-sheet and α-helix 6. The packing of helix α6 (residues 291–306) against α1/β1 of the LG domain and against helix α7 of the middle domain was hypothesized to stabilize the relative location of LG and middle domain against each other. Most intriguingly, the Sau group reported on the importance of helix α6 for full catalytic activity of hGBP1 and for oligomer formation. They could also clearly establish the relationship between oligomer formation and defensive activity against hepatitis C virus showing that impairing catalytic activity and oligomer formation by mutations leads also to a decreased antiviral activity (*Pandita et al., 2016*). These observations support our conclusions as to the importance of the movements around the pivot point located close to α-helix 6. Similar movements have been reported for other dynamin-like proteins, where the GTPase domain rearranges with respect to the middle domain along the catalytic cycle (*Faelber et al., 2012*; *Kalia et al., 2018*; *Cui et al., 2021*).

Previous data revealed two hGBP1 dimer conformations. In the major populated $D_2$ conformation two α13 helices dimerize while in the minor $D_1$ conformation helix α13 are separated (*Kravets et al., 2012*; *Vöpel et al., 2014*; *Kravets et al., 2016*; *Shydlovskyi et al., 2017*). Our new findings in this work lead to a common model which describes the reaction pathway of hGBP1 from a monomer to the formation of mesoscale droplets in vitro and living cells (*Figure 6*). We found that $M_1$ is the prevailing conformation in solution. Thus, even though hGBP1 is flexible it likely first dimerizes via the LG domain to form a stable $D_2$ dimer. All structural requirements for this multi-step conformational rearrangement for positioning the two interaction sites and defining the molecular polarity are already predefined in the monomeric hGBP1 molecule. In the absence of substrate and other GBP molecules,

hGBP1 adopts at least two distinct conformational states. Upon addition of GTP, the LG domain can bind to another protomer, whilst the conformational dynamics appear to remain unchanged (*Vöpel et al., 2014*) which agrees with our current findings. When two GTP-bound hGBP1s associate, a head-to-head dimer is formed either in a $M_1$:$M_1$, $M_2$:$M_2$ or a $M_1$:$M_2$ configuration. As the $M_1$:$M_2$ dimer has a higher stability, the α13 helices of the two subunits associate and the equilibrium is shifted towards the $M_1$:$M_2$ dimers (*Vöpel et al., 2014*).

*Figure 6* highlights the capability of hGBP1 to form networks during phase separation. Notably, hGBP1 shares these features with other proteins that also undergo phase separation. As observed in this work, conformational flexibility, multivalent interactions and amphiphilicity were reported as important factors for phase separation (*Banani et al., 2017*; *Cui et al., 2021*). Moreover, directionality is introduced because hGBP1's interaction sites have distinct affinities that define the polarity of the formed molecular assembly. The high affinities of LG domains ensure formation of a dimeric encounter complex already at low concentrations in the first step. The conformational flexibility of hGBP1's effector domain promotes the second key step for multimerization - the association of helices α13 that makes the dimer amphiphilic.

In a more general view, our results on hGBP1 demonstrate that the exchange between distinct protein conformations is usually encoded in its design (pathway *i*, *Figure 1B*). Thus, the conformational flexibility of a protein can already be a characteristic of the apo form although this property is only relevant for a later stage of the protein's functional cycle, for example in a complex with its ligand GTP, substrates and other proteins, respectively. Considering, for example, the movement of the substrate-dependent conformational transitions in the finger subdomain of a DNA polymerase (*Rothwell et al., 2013*), these opening and closing movements are essential for catalyzing polymerization under ambient conditions. The rule that functionally relevant conformational equilibria may be predefined by protein design also applies to other steps in protein function. In future, when considering additional quantitative live-cell, single-molecule and kinetic studies on farnesylated hGBP1, such integrative approaches may provide a molecular picture of complex biological processes like intracellular immune response.

In a broader perspective, this work and further experimental studies *Hellenkamp et al., 2017*; *Borgia et al., 2018*; *Lerner et al., 2018*; *Dimura et al., 2020*; *Sanabria et al., 2020*; *Lerner et al., 2021*; *Agam et al., 2022*, *Hamilton et al., 2022* demonstrate the great capabilities of integrative label-based studies in combination with other experimental techniques to resolve the structure and dynamics of proteins under native conditions. This information on the promiscuous nature of proteins can contribute to shape a dynamic view on these macromolecules that links structural states and conformational dynamics with function. In case of hGBP1, the intrinsic flexibility is crucial for oligomerization (*Figure 1* and *Figure 6*). Moreover, the obtained knowledge paves the way toward dynamic structural biology where structural models and kinetic information can be archived and disseminated in databases such as the prototype archiving system PDB-Dev (*Berman et al., 2019*).

# Materials and methods
## Protein expression and labeling
### Expression and purification
SAXS experiments were performed on native non-farnesylated hGBP1 variants. Cysteine non-farnesylated variants for EPR and fluorescence experiments are based on cysteine-free hGBP1 (C12A/ C82A/ C225S/ C235A/ C270A/ C311S/ C396A/ C407S/ C589S) and were constructed in a pQE80L vector (Qiagen, Germany) following the instructions of the QuikChange site-directed mutagenesis kit (Stratagene, USA) according to *Vöpel et al., 2009*; *Vöpel et al., 2014*. Neither amino acid positions in direct proximity to the nucleotide binding pocket nor inside the G domain dimerization interface nor charged amino acids on the protein surface were taken into consideration for labeling (*Tsodikov et al., 2002*). All chosen positions had an accessible surface area (ASA) value higher than 60 Å$^2$. Previously, these mutations were shown to only weakly affect non-farnesylated hGBP1's function (*Vöpel et al., 2009*; *Vöpel et al., 2014*). New cysteines were introduced at various positions of interest (N18C, Q254C, Q344C, T481C, A496C, Q525C, V540C, Q577C). The GTPase activity of the labeled and unlabeled non-farnesylated hGBP1 variants was quantified by an assay as previously described (*Kunzelmann et al., 2006*) (**Appendix 2**). The mutagenesis was verified by DNA sequencing with

a 3130xl sequencer (Applied Biosystems, USA). hGBP1 was expressed in BL21-CodonPlus(DE3)-RIL (Supplier Agilent) and purified following the protocol described previously (*Praefcke et al., 1999*). A Cobalt-NTA-Superflow was used for affinity chromatography. No glycerol was added to any buffer as it did not make any detectable differences. To not interfere with the following labeling reactions, the storage buffer did not contain DTT or DTE. Protein concentrations were determined by absorption at 280 nm according Gill and Hippel using an extinction coefficient of 45,400 $M^{-1}$ $cm^{-1}$. Tests of enzyme activity and function demonstrate that the effect of mutations and labeling on non-farnesylated hGBP1's function is small (**Appendix 2**).

## Protein labeling

FRET labeling was performed in two steps. To start the first labeling reaction, a solution with a hGBP1 concentration 100–300 μM in labeling buffer containing 50 mM Tris-HCl (pH 7.4), 5 mM $MgCl_2$, 250 mM NaCl was gently mixed with a 1.5-fold molar excess of Alexa647. After 1 hour incubation on ice, the unbound dye was removed using a HiPrep 26/20 S25 desalting column (GE Healthcare, Germany) with a flow rate of 0.5 ml/min. After this first labeling step, double, single, and unlabeled proteins were separated based on the charge difference introduced by the coupled dyes using anion exchange chromatography on a ResourceQ column (GE Healthcare, Germany) and a salt gradient running from 0 to 500 mM NaCl over 120 ml at a pH of 7.4 and flow rate of 2.0 ml/min. The peaks in the elugram were analyzed for their degree of labeling (dol) by measuring their absorption by UV/Vis spectroscopy at a wavelength of 280 nm and 651 nm. The fraction with the highest, single-acceptor labeled protein amount was labeled with a fourfold molar excess of Alexa488 C5 maleimide (Alexa488). The unreacted dye was separated as described for the first labeling step. Finally, the degrees of labeling (dol) for both dyes were determined (usually 70–100% for each dye). The dol was determined by absorption using 71,000 $M^{-1}$ $cm^{-1}$ and 265,000 $M^{-1}$ $cm^{-1}$ as extinction coefficients for Alexa488 and Alexa647, respectively. The labeled proteins were aliquoted into buffer containing 50 mM Tris-HCl (pH 7.9), 5 mM $MgCl_2$, 2 mM DTT, shock-frozen in liquid nitrogen and stored at –80 °C. We determined a Förster radius $R_0$=52 Å for the FRET-pair Alexa488 - Alexa647.

The spin labeling reactions were conducted at 4 °C for 3 hr using an 8-fold excess of (1-Oxyl-2,2, 5,5-tetramethylpyrroline-3-methyl) methanethiosulfonate (MTSSL) as a spin label (Enzo Life Sciences GmbH, Germany). The reaction was performed in 50 mM Tris, 5 mM $MgCl_2$ dissolved in $D_2O$ at pH 7.4. Unbound spin labels were removed with Zeba Spin Desalting Columns (Thermo Fisher Scientific GmbH, Germany) equilibrated with 50 mM Tris, 5 mM $MgCl_2$ dissolved in $D_2O$ at pH 7.4. Concentrations were determined as described before. Labeling efficiencies were determined by double integration of CW room temperature (RT) EPR spectra by comparison of the EPR samples to samples of known concentration. In all cases, the labeling efficiencies were ~90–100%.

## Small angle X-ray scattering

SAXS experiments were performed on the beamlines X33 at the Doris III storage ring, DESY and at the BM29, ESRF (*Pernot et al., 2013*) using X-ray wavelengths of 1.5 Å and 1 Å, respectively. On BM29 a size exclusion column (Superdex 200 10/300 GL, GE Healthcare) was coupled to the SAXS beamline (SEC-SAXS). The scattering vector $q$ is defined as $q = 4\pi/\lambda \cdot sin(\theta/2)$ with the incident wavelength $\lambda$ and the scattering angle θ. The measurements cover an effective $q$ range from 0.015 to 0.40 $Å^{-1}$ for X33 data and 0.006–0.49 $Å^{-1}$ for BM29 data.

SAXS allows determining the shape and low-resolution structure of proteins in solution by the measured scattering intensity $I(q)$, which is proportional to the form factor $F(q)$ multiplied by the structure factor $S(q)$. (*Svergun et al., 1995*) $F(q)$ informs about the electron distribution in the protein, while $S(q)$ contains $q$-dependent modulations due to protein-protein interactions occurring at higher protein concentration. At sufficiently low protein concentrations (in the limit of $c \to 0$) the structure factor converges towards unity. A concentration series (non-farnesylated hGBP1 concentrations of 1.1, 2.1, 5.0, 11.5, and 29.9 mg/mL) was recorded on X33 (*Figure 2—figure supplement 1A*), whereas on BM29 two SEC-SAXS runs have been performed using non-farnesylated protein concentrations of 2 mg/mL and 16 mg/mL that were loaded on the SEC column. The same buffer was used for both SAXS and SEC-SAXS experiments: 50 mM TRIS, 5 mM $MgCl_2$, 150 mM NaCl at pH 7.9. The SEC-SAXS data were averaged over the elution peak. The obtained SEC-SAXS data of the used high and low protein solutions were overlapping validating the infinite dilution limit. Therefore, the SEC-SAXS data

recorded at the high protein concentration were used for further data analysis. An automated sample changer was used for sample loading and cleaning of the sample cell on X33. The storage temperature of the sample changer and the temperature during X-ray exposure in the sample cell were 10 °C. The buffer was measured before and after each protein sample as a check of consistency. For each sample, eight frames with an exposure time of 15 s each were recorded to avoid radiation damage. The absence of radiation damage was verified by comparing the measured individual frames. The frames without radiation damage were merged. On BM29 X-ray frames with exposure time of 1 s were continuously recorded. The scattering contribution of the buffer and the sample cell was subtracted from the measured protein solutions. Measured background corrected SAXS intensities $I(q,c)$ of the non-farnesylated hGBP1 solutions are shown in *Figure 2—figure supplement 1A*. $I(q,c)$ were scaled by the protein concentration $c$ and extrapolated ($c \to 0$) to determine the form factor $I(q,0)$ of the protein at infinite dilution. At larger $q$-values, where the structure factor equals unity, the extrapolated form factor overlapped with the SAXS data of the highest protein concentration within the error bars. Therefore, for better statistics the extrapolated form factor at small $q$-values and the data of the 29.9 mg/mL solution at larger scattering vectors were merged. The structure factor $S(q,c)$ (*Figure 2—figure supplement 1B*) was extracted by $S(q,c)=I(q,c) / (c \cdot I(q,0))$ and fitted by a Percus-Yevik structure factor including the correction of *Kotlarchyk and Chen, 1983* for asymmetric particles resulting in an effective hard sphere radius of 2.2 nm (*Wertheim, 1963*). Size exclusion chromatography SAXS (SEC-SAXS) measurements were performed at different protein concentrations (*Figure 2—figure supplement 1A*). SEC-SAXS assures the data quality by discriminating a sample purification step immediately before the SAXS data acquisition.

SAXS data was analyzed using the ATSAS software package (*Petoukhov et al., 2012*). Theoretical scattering curves of the crystallographic and the simulated structures of the monomer were calculated and fitted to the experimental SAXS curves using the computer program CRYSOL. The distance distribution function $P(r)$ was determined using the program DATGNOM. Ab initio models were generated using the program DAMMIF. In total 20 ab initio models were generated, averaged and the filtered model was used. Normalized spatial discrepancy (NSD) values of the different DAMMIF models were between 0.8 and 0.9 indicative of good agreement between generated ab initio models. The resolution of the obtained ab initio model is 29±2 Å as evaluated by the resolution assessment algorithm.

## Pulse EPR (DEER) experiments

Experiments were performed and are described by *Vöpel et al., 2014*. Briefly, experiments were carried out at X-band frequencies (~9.4 GHz) with a Bruker Elexsys 580 spectrometer equipped with a split-ring resonator (Bruker Flexline ER 4118X-MS3) in a continuous flow helium cryostat (CF935; Oxford Instruments) controlled by an Oxford Intelligent Temperature Controller ITC 503S adjusted to stabilize a sample temperature of 50 K. Sample conditions for the EPR experiments were 100 µM protein in 100 mM NaCl, 50 mM Tris-HCl, 5 mM $MgCl_2$, pH 7.4 dissolved in $D_2O$ with 12.5% (v/v) glycerol-$d_8$. DEER inter spin-distance measurements were performed using the four-pulse DEER sequence (*Martin et al., 1998*; *Pannier et al., 2000*):

$$\frac{\pi}{2} \left(\nu_{obs}\right) - \tau_1 - \pi \left(\nu_{obs}\right) - t' - \pi \left(\nu_{pump}\right) - \left(\tau_1 + \tau_2 - t'\right) - \pi \left(\nu_{obs}\right) - echo \tag{1}$$

with observer pulse ($\nu_{obs}$) lengths of 16 ns for $\pi/2$ and 32 ns for $\pi$ pulses and a pump pulse ($\nu_{pump}$) length of 12 ns. A two-step phase cycling (+ ‹x›, - ‹x›) was performed on $\pi/2(\nu_{obs})$. Time $t'$ was varied with fixed values for $\tau_1$ and $\tau_2$. The dipolar evolution time is given by $t=t' - \tau_1$. Data were analyzed only for $t>0$. The resonator was overcoupled to $Q$~100. The pump frequency $\nu_{pump}$ was set to the center of the resonator dip coinciding with the maximum of the EPR absorption spectrum. The observer frequency $\nu_{obs}$ was set ~65 MHz higher, at the low field local maximum of the EPR spectrum. Deuterium modulation was averaged by adding traces recorded with eight different $\tau_1$ values, starting at $\tau_{1,0} = 400$ ns and incrementing by $\Delta\tau_1 = 56$ ns. Data points were collected in 8 ns time steps or, if the absence of fractions in the distance distribution below an appropriate threshold was checked experimentally, in 16 ns time steps. The total measurement time for each sample was 4–24 h.

The DEER data was analyzed using the software DeerAnalysis which implements a Tikhonov regularization (*Jeschke et al., 2006*). Background correction of the DEER signal dipolar evolution function $V(t)$ (normalized to unity at the time $t=0$)

$$V(t) = F(t) \cdot V_{background}(t), \tag{2}$$

was performed assuming an isotropic distribution of the spin-labeled hGBP1 molecules in frozen solution that is described by

$$V_{background}(t) = exp(-k \cdot t). \tag{3}$$

Briefly, the resulting form factor $F(t)$ is modulated with the dipolar frequency

$$\omega_{DD}(R_{SS}, \theta) = \frac{1}{4\pi} \cdot \frac{g^2 \mu_B^2 \mu_0}{\hbar} \cdot \frac{1}{R_{SS}^3} \cdot (3\cos^2\theta - 1), \tag{4}$$

that is proportional to the cube of the inverse of the inter-spin distance $R_{SS}$ ($\mu_B$: Bohr magneton; $\mu_0$: magnetic field constant; $\theta$: angle between the external magnetic field and the vector connecting the two spins, for nitroxide spin labels the $g$ values of both spins can be approximated with the isotropic value $g \approx 2.006$). Analysis of the form factor $F(t)$ in terms of a distance distribution $p(R_{SS})$ was performed by a Tikhonov regularization. A simulated time domain signal

$$S(t) = K(t, R_{SS}) \cdot p(R_{SS}) \tag{5}$$

from a given distance distribution $p(R_{SS})$ was calculated by means of a kernel function

$$K(t, R_{SS}) = \int_0^1 cos[(3x^2 - 1) \cdot \omega_{DD} \cdot t] dx \tag{6}$$

with $\omega_{DD}(R_{SS}) = \frac{2\pi \cdot 52.04 \; MHz \; nm^{-3}}{R_{SS}^3}$ for nitroxide spin labels. The optimum $p(R_{SS})$ was found by minimizing the objective function

$$G_\alpha(P) = \|S(t) - V_{local}(t)\|^2 + \alpha \cdot \left\| \frac{d^2}{dr^2} p(R_{SS}) \right\|^2. \tag{7}$$

The regularization parameter $\alpha$ was varied to find the best compromise between smoothness, that is, the suppression of artifacts introduced by noise, and resolution of $p(R_{SS})$. The optimum regularization parameter was determined by the L-curve criterion, where the logarithm of the smoothness $\left\| \frac{d^2}{dr^2} p(R_{SS}) \right\|^2$ of $p(R_{SS})$ is plotted against the logarithm of the mean square deviation $\|S(t) - V_{local}(t)\|^2$, allowing to choose the distance distribution with maximum smoothness representing a good fit to the experimental data.

Theoretical inter spin label distance distributions for MTSSL spin labels attached to structural models have been calculated using the rotamer library analysis (RLA) implemented in the freely available software MMM (*Polyhach et al., 2011*).

## Neutron spin echo

Neutron spin echo (NSE) was measured on IN15 at the Institut Laue-Langevin, Grenoble, France. The NSE data were described by rigid body diffusion of non-farnesylated hGBP1 to detect intra-molecular dynamics. Four incident neutron wavelengths with 8, 10, and 12.2, and 17.5 Å were used. The buffer composition for NSE experiments was 50 mM TRIS, 5 mM MgCl$_2$, 150 mM NaCl at pD 7.9 in heavy water (99.9 atom % D). The protein concentration was 30 mg/mL. The measured NSE spectra are shown in *Figure 3—figure supplement 1*. Effective diffusion coefficients $D_{eff}$ were determined from the initial slope of the NSE spectra by using a cumulant analysis $\frac{I(q,t)}{I(q,0)} = exp\left(K_1 t + \frac{1}{2} K_2 t^2\right)$ with $D_{eff} = \frac{-K_1}{q^2}$.

The rigid body diffusion $D_0(q)$ of a structural model at infinite dilution was calculated according to *Biehl et al., 2011*:

$$D_0(q) = \frac{1}{q^2 F(q)} \sum_{j,k} \left\langle b_j \, exp(-i\vec{q}\vec{r}_j) \begin{pmatrix} \vec{q} \\ \vec{q} \times \vec{r}_j \end{pmatrix} \hat{D} \begin{pmatrix} \vec{q} \\ \vec{q} \times \vec{r}_k \end{pmatrix} b_k \, exp(i\vec{q}\vec{r}_k) \right\rangle \tag{8}$$

where $\hat{D}$ is the 6x6 diffusion tensor, which was calculated using the HYDROPRO program (**Ortega et al., 2011**). $D_0(q)$ was calculated for the hGBP1 crystal structure (PDB-ID: 1DG3) and the best representing $M_2$ structure. The population values have been determined from fits to the SAXS data with 69% best representing $M_2$ structure and 31% crystal structure at the temperature of 10 °C.

The full NSE spectra were described by rigid body diffusion and internal protein dynamics according to **Inoue et al., 2010**:

$$I(q,t)/I(q,0) = \left[(1-A(q))+A(q)\,exp(-\Gamma t)\right]\cdot$$
$$exp\left(-q^2 D_t \frac{H_t}{S(q)}t\right)\left(\sum_{l=0}^{15} S_l(q)\,exp(-l(l+1)D_r H_r t)\right)/\sum_{l=0}^{15} S_l(q) \tag{9}$$

with $S_l(q) = \sum_m \left|\sum_i b_i j_l(qr_i) Y_{l,m}(\Omega_i)\right|^2$

where $D_t$ and $D_r$ are the calculated scalar translational and rotational diffusion coefficients found in the trace of $\hat{D}$ of the rigid protein at infinite dilution from the structural models. Rotational diffusion of the rigid protein were expressed in spherical harmonics with spherical Bessel functions $j_l(qr_i)$, spherical harmonics $Y_{l,m}$ and scattering length densities $b_i$ of atoms at positions $r_i$. Here, the crystal structure was used as a base. $D_t$ and $D_r$ were chosen according to the mixture of crystal structure and best representing $M_2$ structure. Direct interaction and hydrodynamic interactions were accounted for by the corrections $D_{t,eff}(q) = D_t H_t/S(q)$ and $D_{r,eff} = H_r D_r$. Interparticle interactions were considered by the structure factor $S(q)$ as measured by SANS. $H_t$ and $H_r$, reduce the effective translational and rotational diffusion coefficients. $H_t$ is related to the intrinsic viscosity $[\eta]$ by $H_t = 1-c[\eta]$ and $H_r$ can be approximated by $1-H_r=(1-H_t)/3$ for spherical particles (**Degiorgio et al., 1995**), which might underestimate $H_r$ for large asymmetric particles. Internal protein dynamics was described by an exponential decay with a $q$-independent rate $\Gamma$, and a $q$-dependent contribution $A(q)$ of internal dynamics to the NSE spectra.

The parameters $H_t$, $H_r$, the relaxation time $\lambda$, and the amplitudes $A(q)$ (Materials and methods, **Equation 9**) were simultaneously optimized to all NSE spectra (**Figure 3—figure supplement 1**). The fits show a small contribution of internal dynamics with amplitudes close to the error bars and seemingly long relaxation times, but not strong enough to be determined unambiguously. Fitting the spectra without additional internal dynamics shows an excellent description of the data (**Figure 3—figure supplement 1**) with $H_t = 0.61 \pm 0.01$ and $H_r = 0.72 \pm 0.03$ as the only fitting parameters.

Dynamic light scattering was measured on a Zetasizer Nano ZS instrument (Malvern Instruments, Malvern, United Kingdom) in $D_2O$ buffer identical to that used in the NSE experiment. Autocorrelation functions were analyzed by the CONTIN like algorithm (**Provencher, 1982**) to obtain the translational diffusion coefficient $D_T$ need for analysis. The diffusion coefficient of a single protein increases from the translational diffusion $D_T$ measured at low $q$ (DLS) due to contributions from rotational diffusion $D_R(q)$ and contributions related to internal protein dynamics $D_{int}(q)$ as the observation length scale $2\pi/q$ covers the protein size. The translational and rotational diffusion coefficients $D_T$ and $D_R(q)$ were calculated and corrected for hydrodynamic interactions and interparticle effects to result in the expected $D_0(q)$ for a rigid body (**Figure 3A**, black line, Materials and methods, **Equation 9**).

## Fluorescence spectroscopy

Ensemble and single-molecule FRET experiments were performed at room temperature in 50 mM Tris-HCl buffer (pH 7.4) containing 5 mM $MgCl_2$ and 150 mM NaCl. All ensemble measurements were performed at concentrations of labeled protein of approximately 200 nM. The single-molecule (sm) measurements were performed at concentrations of labeled protein of approximately 20 pM to assure that only single-molecules were detected. All sm MFD-measurements probing the hGBP1 apo state were performed under two conditions: (*i*) without unlabeled protein, and (*ii*) with 7.5 µM unlabeled protein to minimize the loss of labeled molecules due to adsorption in the measurement chamber. Both conditions gave comparable results. Due to the higher counting statistics, all results of the apo state reported in this work have been obtained for condition *ii*. To study also the ligand-bound non-farnesylated holo state hGBP1:L (**Figure 1B**) by fFCS (**Figure 4D**), we used the ligand GDP-AlF$_x$ as a non-hydrolysable substrate. The ligand GDP -AlF$_x$ is formed in situ by diluting a stock solution with 30 mM $AlCl_3$ and 1 M NaF by 1:100 in the standard buffer containing 100 µM GDP and 20 pM labeled protein without unlabeled protein (condition *i*).

Ensemble fluorescence time-correlated single-photon-counting (TCSPC) measurements of the donor fluorescence decay histograms were either performed on an IBH-5000U (HORIBA Jobin Yvon IBH Ltd., UK) equipped with a 470 nm diode laser LDH-P-C 470 (Picoquant GmbH, Germany) operated at 8 MHz or on a EasyTau300 (PicoQuant, Germany) equipped with an R3809U-50 MCP-PMT detector (Hamamatsu) and a BDL-SMN 465 nm diode laser (Becker & Hickl, Germany) operated at 20 MHz. The donor fluorescence was detected at an emission wavelength of 520 nm using a slit-width that resulted in a spectral resolution of 16 nm in the emission path of the machines. A cut-off filter (495 nm) in the detection path additionally reduced the contribution of the scattered light. All measurements were conducted at room temperature under magic-angle conditions. Typically, $14 \cdot 10^6$–$20 \cdot 10^6$ photons were recorded at TAC channel-width of 14.1 ps (IBH-5000U) or 8 ps (EasyTau300). When needed, the analysis considers differential non-linearities of the instruments by multiplying the model function with a smoothed and normalized instrument response of uncorrelated room light. The fits cover the full instrument response function (IRF) and 99.9% of the total fluorescence. The IRFs had typically FWHM of 254 ps (IBH-5000U) or 85 ps (PicoQuant EasyTau300).

Single-molecule fluorescence spectroscopy data was acquired on a custom MFD setup with polarized excitation and detection in the 'green' and 'red' detection channels (*Sisamakis et al., 2010*). Briefly, a beam of linearly polarized pulsed argon-ion laser (Sabre, Coherent) was used to excite freely diffusing molecules through a corrected Olympus objective (UPLAPO 60X, 1.2 NA collar (0.17)). The laser was operated at 496 nm and 73.5 MHz. An excitation power of 120 µW at the objective has been used during experiments. The fluorescence light was collected through the same objective and spatially filtered by a 100 µm pinhole which defines an effective confocal detection volume of ~3 fl. A polarizing beam-splitter divided the collected fluorescence light into its parallel and perpendicular components. Next, the fluorescence light passed a dichroic beam splitter that defines a 'green' and 'red' wavelength range (below and above 595 nm, respectively). After passing through band pass filters (AHF, HQ 520/35 and HQ 720/150) single photons were detected by two 'green' (either $\tau$-SPADs, PicoQuant, Germany or MPD-SPADs, Micro Photon Devices, Italy) and two 'red' detectors (APD SPCM-AQR-14, Perkin Elmer, Germany). Two SPC 132 single photon counting boards (Becker & Hickel, Berlin) have recorded the detected photons stream. Thus, for each detected photon the arrival time after the laser pulse, the time since the last photon and detection channel number (so, polarization and color) were recorded.

## Burst-wise single molecule analysis

Briefly, as the first step in the burst-wise analysis, fluorescence bursts were discriminated from the background signal of 1–2 kHz of the single-molecule measurements by applying an intensity threshold criterion. Next, the anisotropy and the fluorescence averaged lifetime, $\langle \tau_{D(A)} \rangle_F$, were determined for each burst. Moreover, the background, the detection efficiency-ratio of the 'green' and 'red' detectors, and the spectral cross-talk were considered to determine the FRET efficiency, $E$, of every burst (*Sisamakis et al., 2010*). Absolute FRET efficiencies require calibrated instruments (*Hellenkamp et al., 2018*) and considerations of the excitation power and FRET-dependent photophysics (*Widengren et al., 2001*). The species averaged fluorescence lifetime of the donor in the absence of an acceptor $\langle \tau_{D(0)} \rangle_x$, $\langle \tau_{D(A)} \rangle_F$, and the FRET efficiency estimate the mean, $\langle \tau_{D(A)} \rangle_x = \left(1 - E\right) \cdot \langle \tau_{D(0)} \rangle_x$, and variance, $var\left(\tau_{D(A)}\right) = \langle \tau_{D(A)} \rangle_F \cdot \langle \tau_{D(A)} \rangle_x - \langle \tau_{D(A)} \rangle_x^2$, of the burst averaged fluorescence lifetimes distribution. This highlights conformational dynamics by a non-zero variance (*Figure 3—figure supplement 2*). For a detailed analysis of the sub-ensemble, the fluorescence photons of multiple bursts were integrated into joint fluorescence decay histograms (seTCSPC, *Figure 3—figure supplement 3*).

## FRET-lines

By relating fluorescence parameters, FRET lines serve as a visual guide to interpret histograms of MFD parameters determined for individual molecules. The fluorescence weighted lifetime of the donor, $\langle \tau_{D(A)} \rangle_F$, and the FRET efficiency, $E$, were related by FRET-lines by a methodology similar as previously described (*Kalinin et al., 2010*). First, FRET-rate constant distributions, $p(k_{RET})$, were calculated for a given set of model parameters. Next, $p(k_{RET})$ was converted to the averages $\langle \tau_{D(A)} \rangle_F$ and $E$. This results in a parametric relation between $\langle \tau_{D(A)} \rangle_F$ and $E$ called FRET-line. We use two types of FRET-lines: dynamic and static FRET-lines. Dynamic FRET-lines describe the mixing of typically two states. A static FRET-line

relates $\langle \tau_{D(A)} \rangle_F$ to $E$ for all molecules that are static within their observation time (the burst duration). Static molecules are identified by populations in a MFD histogram located on the static FRET-line. The FRET-lines were calculated using the scripting capability of ChiSurf assuming states with normal distributed distance and are calibrated for sample-specific fluorescence properties, that is, donor and acceptor fluorescence quantum yields, the fraction of acceptor in power dependent dark states (cis-state in Alexa647), and complex fluorescence decays of the donor in the absence of FRET.

## Fluorescence decay analysis

Fluorescence decay analysis was performed using ChiSurf, an open-source software tailored for the global analysis of multiple fluorescence experiments. Fluorescence intensity decays of the donor in the presence, $f_{D|D}^{(DA)}(t)$, and the absence of FRET, $f_{D|D}^{(D0)}(t)$, inform on DA distance distributions, $p(R_{DA})$ (*Peulen et al., 2017*). However, the local environment of the dyes may result in complex fluorescence decays of the donor $f_{D|D}^{(D0)}(t)$ and the acceptor $f_{A|A}^{(AD)}(t)$ even in the absence of FRET. Such sample-specific fluorescence properties were accounted for by donor and acceptor reference samples using single cysteine variants. $f_{D|D}^{(D0)}(t)$ and $f_{A|A}^{(A0)}(t)$ were formally described by multi-exponential model functions:

$$f_{D|D}^{(D0)}(t) = \sum_i x_D^{(i)} exp\left(\frac{-t}{\tau_D^{(i)}}\right)$$
$$f_{A|A}^{(DA)}(t) = \sum_i x_A^{(i)} exp\left(\frac{-t}{\tau_A^{(i)}}\right) \tag{10}$$

Here, $D|D$ refers to the donor fluorescence under the condition of donor excitation and $A|A$ refers to the acceptor fluorescence under acceptor excitation. Species fractions $x_D^{(i)}$ and $x_A^{(i)}$ and lifetimes of the donor $\tau_D^{(i)}$ and the acceptor $\tau_A^{(i)}$ are summarized in *Appendix 1—table 2*.

We assume that the same distribution of FRET-rate constants quenches all fluorescent states of the donor (quasi-static homogeneous model *Peulen et al., 2017*). Thus, $f_{D|D}^{(DA)}(t)$ can be expressed by:

$$f_{D|D}^{(DA)}(t) = f_{D|D}^{(D0)}(t) \cdot \sum_i x_{RET}^{(i)} exp\left(-t \cdot k_{RET}^{(i)}\right) = f_{D|D}^{(D0)}(t) \cdot \epsilon_D(t), \tag{11}$$

where $\epsilon_D(t)$ is the FRET-induced donor decay. The MFD measurements demonstrate that the major fraction of the dyes is mobile (**Appendix 2**). Therefore, we approximate $\kappa^2$ by 2/3 and relate $\epsilon_D(t)$ to $p(R_{DA})$ by:

$$\epsilon_D(t) = \int_{R_{DA}} p(R_{DA}) \cdot exp\left(-t \cdot k_0 \cdot \left(\frac{R_0}{R_{DA}}\right)^6\right) dR_{DA} + x_{DOnly}. \tag{12}$$

Here, $R_0$ is the Förster radius ($R_0$=52 Å) and $k_0$=1/$\tau_0$ is the radiative rate constant of the unquenched dye ($\tau_0$ = 4 ns). In $\varepsilon_D(t)$ incomplete labeled molecules lacking an acceptor and molecules with bleached acceptors are considered by the fraction of FRET-inactive, $x_{DOnly}$.

For rigorous uncertainty estimates $p(R_{DA})$ was modeled by a linear combination of normal distributions. Overall, a superposition of two normal distributions with a central distance $\bar{R}_{DA}^{(1,2)}$ and a width $w_{DA}$ was sufficient to describe the data:

$$p(R_{DA}) = \frac{1}{\sqrt{\frac{\pi}{2} \cdot w_{DA}}} \left[ x_1 e^{-\left(\frac{2\left(R_{DA} - \bar{R}_{DA}^{(1)}\right)}{w_{DA}}\right)^2} + (1 - x_1) e^{-\left(\frac{2\left(R_{DA} - \bar{R}_{DA}^{(2)}\right)}{w_{DA}}\right)^2} \right] \tag{13}$$

In the analysis of the seTCSPC data, the FRET-sensitized emission of the acceptor, $f_{A|D}^{(DA)}(t)$, was considered to reduce the overall photon noise and a typical width of 12 Å was consistent with the data. $f_{A|D}^{(DA)}(t)$ was described by the convolution of $f_{A|A}^{(DA)}(t)$, and $f_{D|D}^{(DA)}(t)$:

$$f_{A|D}^{(DA)}(t) = f_{D|D}^{(D0)}(t) \cdot \epsilon_D(t) \otimes f_{A|A}^{(DA)}(t) \tag{14}$$

All $f(t)$s were fitted by model functions using the iterative re-convolution approach (**Straume et al., 2002**). Here, the parameters of a model function $g(t)$ were optimized to the data by using the modified Levenberg–Marquardt algorithm. The model function $g(t)$ considers experimental nuisances as scattered light and a constant background:

$$g(t) = N_F \cdot f(t) \otimes IRF(t) + N_{BG} \cdot IRF(t) + bg \tag{15}$$

$N_F$ is the number of fluorescence photons, $N_{BG}$ is the number of background photons due to Rayleigh or Raman scattering and $bg$ is a constant offset attributed to detector dark counts and afterpulsing. In seTCSPC, the fraction of scattered light and the constant background was calculated by the experimental integration time and the buffer reference measurements. In eTCSPC, the fraction of scattered light and the constant offset were free fitting parameters. Finally, $g(t)$ was scaled to the data by the experimental number of photons and fitted to the experimental data. Statistical errors were estimated by sampling the parameter space (**Foreman-Mackey et al., 2013**) and applying an F-test at a confidence level of 95% in addition to support plane analysis of the parameters (**Straume et al., 2002**).

## Filtered species cross-correlation functions

Filtered fluorescence correlation spectroscopy of the acquired MFD data was performed as previously described (**Felekyan et al., 2012**). In a global analysis, all 48 fFCS curves (two SACF and SCCF per variant) are treated as a single dataset. Filtered FCS increases the contrast by a set of state-specific filters applied to the recorded photon stream. For every FRET pair, a specific set of filters, $w_j^{(i)}$, was generated using experimental fluorescence bursts for high (H) and low (L) FRET states as previously described and listed in **Appendix 1—table 6** (**Felekyan et al., 2012**). Using these filters species cross-correlation functions $G^{(n,m)}(t_c)$ were calculated by weighted signal intensities $S_j(t)$:

$$G^{(n,m)}(t_c) = \frac{\left\langle F^{(n)}(t) \cdot F^{(m)}(t+t_c) \right\rangle}{\left\langle F^{(n)}(t) \right\rangle \cdot \left\langle F^{(m)}(t+t_c) \right\rangle} \text{ with } F^{(n)}(t) = \left( \sum_{j=1}^{d \cdot L} w_j^{(n)} \cdot S_j(t) \right) \tag{16}$$

Herein $n$ and $m$ are the two species (either H or L), $d$ is the number of detectors, $L$ is the number of TAC channels, and $S_j(t)$ is the signal recorded in the TAC-channel $j$. The choice of $n$ and $m$ defines the type of the correlation function. If $n$ equals $m$, $G^{(n,n)}(t_c)$ is a species autocorrelation function (sACF), otherwise $G^{(n,m)}(t_c)$ is a species cross-correlation function (sCCF) (**Felekyan et al., 2012**). Overall four correlation curves were generated per sample: two species auto - $sACF^{H,H}(t_c)$, $sACF^{L,L}(t_c)$ and two species cross - $sCCF^{H,L}(t_c)$, $sCCF^{L,H}(t_c)$ correlation curves. All curves were fitted by a model which factorizes $G^{(n,m)}(t_c)$ into a diffusion-, $G_{diff}^{(n,m)}(t_c)$, and a kinetic- term $G_{kin}^{(n,m)}(t_c)$:

$$G^{(n,m)}(t_c) = 1 + \frac{1}{N_{eff}^{(n,m)}} \cdot G_{Diff}^{(n,m)}(t_c) \cdot G_{kin}^{(n,m)}(t_c) \tag{17}$$

Here, $N_{eff}^{(n,m)}$ is the effective number of molecules. The sACFs were fitted by individual effective numbers of molecules. The two sCCFs shared a single effective number of molecules.

We assume that the same diffusion term can describe all correlation curves of a sample and that the molecules diffuse in a 3D Gaussian illumination/detection profile. Under these assumptions $G_{diff}^{(n,m)}(t_c)$ is

$$G_{Diff} = \left(1 + \frac{t_c}{t_{Diff}}\right)^{-1} \left(1 + \left(\frac{\omega_0}{z_0}\right)^2 \left(\frac{t_c}{t_{diff}}\right)\right)^{-1/2} \tag{18}$$

where $t_{diff}$ the characteristic diffusion time and $\omega_0$ and $z_0$ are the radii of the focal and the axial plane, respectively, where the intensity decayed to $1/e^2$ of the maximum's intensity.

The kinetic terms of the sACF and the sCCF were formally described by:

$$G_{kin}^{L,H}(t_c) = \left(1 - A_0^{LH} \cdot (A_1 \cdot e^{-t_c/t_{c,1}} + A_2 \cdot e^{-t_c/t_{c,2}} + A_3 \cdot e^{-t_c/t_{c,3}})\right)$$

$$G_{kin}^{H,L}(t_c) = \left(1 - A_0^{HL} \cdot (A_1 \cdot e^{-t_c/t_{c,1}} + A_2 \cdot e^{-t_c/t_{c,2}} + A_3 \cdot e^{-t_c/t_{c,3}})\right) \cdot \left(1 - A_b^{HL} \cdot e^{-t_c/t_b}\right)$$

$$G_{kin}^{L,L}(t_c) = \left(1 + A_1^{LL}\left(e^{-t_c/t_{c,1}} - 1\right) + A_2^{LL}\left(e^{-t_c/t_{c,2}} - 1\right) + A_3^{LL}\left(e^{-t_c/t_{c,3}} - 1\right)\right)$$

$$G_{kin}^{H,H}(t_c) = \left(1 + A_1^{HH}\left(e^{-t_c/t_{c,1}} - 1\right) + A_2^{HH}\left(e^{-t_c/t_{c,2}} - 1\right) + A_3^{HH}\left(e^{-t_c/t_{c,3}} - 1\right)\right) \cdot \left(1 + A_b^{HH}(e^{-t_c/t_b} - 1)\right)$$

$$(19)$$

Here, $A_0$ defines the amplitude of the anti-correlation; $A_b$ accounts for acceptors bleaching in the high-FRET state; $t_b$ is the characteristic bleaching time of the acceptor (under the given conditions typically 5–10ms); $A_1$, $A_2$ and $A_3$ together with $t_{c,1}$, $t_{c,2}$ and $t_{c,3}$ define the anti-correlation time spectrum of the H to L and L to H transitions. The sum of $A_1$, $A_2$ and $A_3$ was constrained to unity. The correlation times $t_{c,1}$, $t_{c,2}$ and $t_{c,3}$ were global parameters shared among all samples. $A_1$, $A_2$ to $A_3$ were sample specific. The amplitudes $A_1^{HH}, A_2^{HH}, A_3^{HH}$ and $A_1^{LL}, A_2^{LL}, A_3^{LL}$ of the sACFs were non-global parameters optimized for every curve individually. Overall, 48 correlation curves of 12 samples were analyzed as a joint dataset. The uncertainties of the amplitudes and correlation times were determined by support plane analysis that considers the mean and the standard deviation of the individual correlation channels. Estimates for the mean and the standard deviation of the correlation channels were determined by splitting individual measurements. The global data analysis of the FCS curves was performed using ChiSurf.

## MD simulations and principal component analysis

### MD simulations

We performed molecular dynamics (MD) and accelerated MD (aMD) (*Hamelberg et al., 2004*) simulations to identify collective degrees of freedom, essential movements, and correlated domain motions of hGBP1 by Principal Component Analysis (PCA) (*Hamelberg et al., 2004*). Molecular dynamics simulations were performed using the Amber14 package (*Case et al., 2015*) and the ff14SB force field. The simulations were started from a known crystal structure of the full-length non-farnesylated protein (PDB code: 1DG3) protonated with the program PROPKA (*Bas et al., 2008*) at a pH of 7.4, neutralized by adding counter ions and solvated in an octahedral box of TIP3P water (*Jorgensen et al., 1983*) with a water shell of 12 Å around the solute. The obtained system was used to perform unbiased MD simulations and aMD simulations (*Hamelberg et al., 2004*). Five unrestrained all-atom MD simulations were performed. Three of the five simulations were conventional MD (2 µs each) and two aMD simulations (200 ns each). The 'Particle Mesh Ewald' method (*Darden et al., 1993*) was utilized to treat long-range electrostatic interactions; the SHAKE algorithm (*Ryckaert et al., 1977*) was applied to bonds involving hydrogen atoms. For all MD simulations, the mass of solute hydrogen atoms was increased to 3.024 Da and the mass of heavy atoms was decreased respectively according to the hydrogen mass repartitioning method (*Hopkins et al., 2015*). The time step in all MD simulations was 4 fs with a direct-space, non-bonded cutoff of 8 Å. For initial minimization, 17500 steps of steepest descent and conjugate gradient minimization were performed; harmonic restraints with force constants of 25 kcal·mol$^{-1}$·Å$^{-2}$, 5 kcal·mol$^{-1}$·Å$^{-2}$, and zero during 2500, 10,000, and 5000 steps, respectively, were applied to the solute atoms. Afterwards, 50 ps of NVT simulations (MD simulations with a constant number of particles, volume, and temperature) were conducted to heat up the system to 100 K, followed by 300 ps of NPT simulations (MD simulations with a constant number of particles, barostat and temperature) to adjust the density of the simulation box to a pressure of 1 atm and to heat the system to 300 K. A harmonic potential with a force constant of 10 kcal·mol$^{-1}$·Å$^{-2}$ was applied to the solute atoms at this initial stage. In the following 100 ps NVT simulations the restraints on the solute atoms were gradually reduced from 10 kcal·mol$^{-1}$·Å$^{-2}$ to zero. As final equilibration step, 200 ps of unrestrained NVT simulations were performed. Boost parameters for aMD were chosen by the method as previously suggested (*Pierce et al., 2012*).

### Principal components analysis (PCA)

In the MD simulations we found fluctuations of RMSD around the average structure of at most 8 Å RMSD for GTP bound and GTP free hGBP1 (*Figure 4—figure supplement 1A*). A correlation analysis of these RMSD trajectories reveals that the dynamics is complex (non-exponential) and predominantly in the 10–100 ns regime (*Figure 4—figure supplement 1B*). Structures deviating the most from the

X-ray structure kink at the connector of the LG and the middle domain (*Figure 3G*). A PCA reveals that the first five principal components describe overall more than 60% of the variance of the MD and aMD simulations (*Figure 3F*). For PCA the GTPase domain (the least mobile domain) was superposed. The mode vectors of the principal components mapped to a crystal structure of hGBP1 (PDB-ID: 1DG3) illustrate the amplitude and the directionality of the principal components (*Figure 3F*). The first component (1) describes a motion of the middle domain towards the LG domain. In the second component (2) the middle domain and α13 move in opposite directions. The third component (3) is like the first component with a two times smaller eigenvalue. Component (4) is like the second component, except that the middle domain and α12/13 move in the same direction. Component (5) captures a similar directionality of motion for the middle domain and α12/13 as the second component. In component (5) however, the movement of α12/13 describes a breathing motion of the catalytic LG domain. The major motions of the PCA can be described by a rotation of the middle domain relative to the GTPase domain (*Figure 3F*, cyan sphere).

## Integrative modeling

A detailed description of our integrative modeling with all steps can be found in **Appendix 3**. In short, DEER, FRET (eTCSPC), and the SAXS data were used to generate integrative structure for the states $M_1$ and $M_2$. Based on the experimental data and the MD simulations, the protein was decomposed into a set of rigid bodies. The assembly of the rigid bodies was sampled using DEER and FRET restraints and refined by NMSim (*Ahmed and Gohlke, 2006*) and MD simulations. All pairs of structures for $M_1$ and $M_2$ were enumerated and scored against the DEER, FRET and SAXS data. The probability for a pair of structures for the DEER and FRET, $p_{DEER,FRET}$, and SAXS, $p_{SAXS}$, are combined in a meta-analysis using Fishers's method.

$$\chi_{2k}^2 \sim -2 \cdot \sum_{i=1}^{k} ln\left(p_i\right) = -2 \cdot ln\left(\prod_{i=1}^{k} p_i\right) = -2 \cdot ln\left(p_{DEER,FRET} \cdot p_{SAXS}\right) \tag{20}$$

The probability $p_{DEER,FRET}$ and $p_{SAXS}$ take the data information content into account. For SAXS the number of Shannon channels was used. For DEER and FRET, the information content was estimated using a greedy backward elimination feature selection algorithm to assess the effective number of informative measurements (*Dimura et al., 2016*). The estimates for the information content were varied to assess the impact on the resulting structures. Finally, an F-test on $\chi_{2k}^2$ is used to discriminate pairs.

The structure generation follows the workflow (*Figure 4A*). Starting from the crystal structure (*Figure 4A*, steps 1–3), we generate new structures (*Figure 4A*, steps 4–5). A set of rigid bodies (RBs) (*Figure 4B*, Appendix 3) was defined based on the motions observed in the MD simulations (*Figure 3F*) taking the following information into account: (1) An order-parameter based rigidity analysis (*Figure 4—figure supplement 1D*); (2) Knowledge on the individual domains within the dynamin family (*Low and Löwe, 2010*; *Chen et al., 2017*); (3) Position dependent FRET and DEER properties (*Appendix 1—table 1*); and (4) The SAXS experiments that suggest a kink in hGBP1's middle domain (Appendix 3, Structure representation). To this RB assembly, we applied DEER and ensemble FRET restrains for guided RB docking (RBD) (Appendix 3) (*Kalinin et al., 2012*). In RBD, the DEER and FRET restraints were treated by AV and ACV simulations, respectively (Appendix 3, Simulation of experimental parameters). AVs for DEER restraints were calibrated (*Figure 2—figure supplement 2*) against established simulation approaches (*Polyhach et al., 2011*; *Hagelueken et al., 2012*).The RBD structures were corrected for their stereochemistry (Appendix 3, Generation of structures) and were clustered into 343 and 414 groups for the states $M_1$ and $M_2$, respectively. Group representatives were used as seeds for short (1–2 ns) MD-simulations. The MD trajectories were clustered into 3395 and 3357 groups for $M_1$ and $M_2$, respectively, before being ranked by the DEER, FRET, and SAXS individually (Appendix 3, Individual ranking of structures). For well-balanced structures and equalized experimental contributions Fisher's method fused the experimental data in a meta-analysis (*Figure 4A*, step 6b) considering estimates for the degrees of freedom (dof) of the protein representation and the data (*Moore, 1980*; *Mertens and Svergun, 2017*) (Appendix 3, Model discrimination and quality assessment; *Figure 4C*, Combined screening). In the meta-analysis a p-value of 0.68 discriminates 95% of all structural models (*Figure 4C*, red area; *Figure 4—figure supplement 2B*). The quality of the selected structures is judged by comparison to the experiments and making use of the data uncertainty. The

local quality of the structures was assessed by checking if their variabilities is above the statistically expectation. Reference structure ensembles are computed to normalize the experimental model precision to a reference precision (Appendix 3, Assessment of model precision and quality).

## Data availability

The following material is available at Zenodo in two locations: Experimental data (https://doi.org/10.5281/zenodo.6534557): (i) fluorescence decays recorded by eTCSPC used to compute the distance restraints in *Appendix 1—table 1* and *Appendix 3—table 1*, (ii) single-molecule multiparameter fluorescence data: all raw data, burst selection and calibration measurements, fFCS (filters and generated correlation curves) (iii) double electron-electron resonance (DEER) EPR data used for structural modeling, (iv) neutron spin-echo data and SAXS structure factor of non-farnesylated hGBP1. Scripts for structural modeling of conformational ensembles through integrative/hybrid methods using FRET, DEER and SAXS together with the initial and selected structural ensembles (https://doi.org/10.5281/zenodo.6565895). The experimental SAXS data and the ab initio analysis thereof are available in the SASBDB (ID SASDDD6). Structure models of hGBP1 based on experimental restraints were deposited to PDB-Dev (PDB-Dev ID: PDBDEV_00000088) using the FLR-dictionary extension (developed by PDB and the Seidel group) available on the IHM working group GitHub site (https://github.com/ihmwg/flrCIF; *IHM Working Group, 2022*). Further data sets generated during and/or analyzed during the current study are available from the corresponding author on request.

Detailed description of the experimental files available on Zenodo (doi 10.5281/zenodo.6534557):

| Folder | Content |
| --- | --- |
| EPR_data.zip | double electron-electron resonance (DEER) EPR data |
| FRET_data.zip | eTCSPC FRET data including fit results. Sample name includes labeled aa and used dyes |
| NSE.zip | neutron spin echo data |
| SMD_hGBP1-[sample].tgz | raw single molecule FRET data used for filter FCS and MFD analysis and calibration measurements. Sample name includes labeled aa |

Subfolders for an eTCSPC FRET measurement contain the following files:

| File | Content |
| --- | --- |
| **decay.dat** | **FRET or donor only decay** |
| prompt.txt | IRF |
| whitelight.txt | Used to create linearization table for tac gates to do full correlation |

Subfolder for eTCSPC FRET measurement 'Fit_results' contains the following files:

| File | Content |
| --- | --- |
| **fit_data.txt** | **Fitted data** |
| fit_fit.txt | Fit curve |
| fit_info.txt | Fit results |
| fit_wr.txt | Residuals of the fit |

Subfolders for a single molecule FRET measurement are structured the following:

| Folder Name | Content |
| --- | --- |
| 'Al488_Al647' | Describes the used dyes, contains the measurement of one sample under various conditions including all files |

*Continued on next page*

*Continued*

| Folder Name | Content |
| --- | --- |
| 'Sample' | Raw data and burst analysis, including info file on burst selection under info-folder |
| 'BID' | Subfolder in burst analysis describing selected bursts used to create filters for fFCS |
| 'fFCS' | Subfolder in burst analysis, contains used lifetime filters, correlation curves and fits |
| 'LP' | Labeled protein, including all measurement files |
| 'LP_nucleotide_UP' | Labeled protein with a nucleotide and additional unlabeled protein, including all measurement files |
| 'LP_UP' | Labeled protein and additional unlabeled protein, including all measurement files |
| 'buffer' | Buffer measurement for background |
| 'H2O' | Water measurement for IRF |
| 'Rh110' | Free dye measurements for g-factor calibration |
| Rh101 | Free dye measurements for g-factor calibration |
| DNA | Calibration measurement for detection efficiency |

## Code availability

Most general custom-made software is directly available from http://www.mpc.hhu.de/en/software. The software ChiSurf is available at https://github.com/Fluorescence-Tools/ChiSurf (*Peulen et al., 2021*). General algorithms and source code are published under https://github.com/Fluorescence-Tools. Additional computer code custom-made for this publication is available upon request from the corresponding authors.

## Acknowledgements

TOP and CL wish to acknowledge the support of the International Helmholtz Research School of Biophysics and Soft Matter (BioSoft). This work is based upon experiments performed on the instruments BM29 at the European Synchrotron Radiation Facility (ESRF), X33 at the Doris III storage ring, DESY, and IN15 at the Institut Laue-Langevin (ILL). We acknowledge the ESRF, the EMBL and the ILL for provision of synchrotron and neutron radiation facilities and we would like to thank Drs. Martha Brennich and Clement Blanchet for assistance in using BM29 and X33.

## Additional information

### Funding

| Funder | Grant reference number | Author |
| --- | --- | --- |
| Deutsche Forschungsgemeinschaft | EXC 2033 - 390677874 - RESOLV | Christian Herrmann |
| Deutsche Forschungsgemeinschaft | HE 2679/6-1 | Christian Herrmann |
| Deutsche Forschungsgemeinschaft | SE 1195/17-1 | Claus AM Seidel |
| Deutsche Forschungsgemeinschaft | STA 1325/2-1 | Andreas M Stadler |

| Funder | Grant reference number | Author |
|---|---|---|
| Deutsche Forschungsgemeinschaft | KL2077/1-2 | Johann P Klare |
| European Research Council | Advanced Grant 2014 hybridFRET (671208) | Claus AM Seidel |
| Deutsche Forschungsgemeinschaft | project no. 267205415 / CRC 1208 | Holger Gohlke |
| Heinrich-Heine-Universität Düsseldorf | Zentrum für Informations- und Medientechnologie" (ZIM) | Holger Gohlke |
| Jülich Supercomputing Centre, Forschungszentrum Jülich | user ID: HKF7 | Holger Gohlke |
| Jülich Supercomputing Centre, Forschungszentrum Jülich | user ID: VSK33 | Holger Gohlke |

The funders had no role in study design, data collection and interpretation, or the decision to submit the work for publication.

## Author contributions

Thomas-O Peulen, Conceptualization, Resources, Data curation, Software, Formal analysis, Validation, Investigation, Visualization, Methodology, Writing – original draft, Writing – review and editing; Carola S Hengstenberg, Tobias Vöpel, Resources, Investigation, Writing – original draft; Ralf Biehl, Software, Formal analysis, Investigation, Visualization, Writing – original draft; Mykola Dimura, Software, Formal analysis, Investigation, Methodology, Writing – original draft; Charlotte Lorenz, Bela Farago, Resources, Formal analysis, Investigation, Writing – original draft; Alessandro Valeri, Formal analysis, Investigation, Writing – original draft; Julian Folz, Resources, Validation, Writing – original draft; Christian A Hanke, Johann P Klare, Andreas M Stadler, Conceptualization, Resources, Data curation, Software, Formal analysis, Supervision, Funding acquisition, Validation, Investigation, Visualization, Methodology, Writing – original draft, Project administration, Writing – review and editing; Semra Ince, Investigation, Writing – original draft; Holger Gohlke, Conceptualization, Resources, Software, Formal analysis, Supervision, Investigation, Writing – original draft; Claus AM Seidel, Conceptualization, Supervision, Funding acquisition, Investigation, Visualization, Methodology, Writing – original draft, Project administration, Writing – review and editing; Christian Herrmann, Conceptualization, Resources, Data curation, Software, Supervision, Funding acquisition, Validation, Investigation, Visualization, Methodology, Writing – original draft, Project administration, Writing – review and editing

## Author ORCIDs

Thomas-O Peulen (iD) http://orcid.org/0000-0001-8478-9755
Ralf Biehl (iD) http://orcid.org/0000-0002-1999-547X
Mykola Dimura (iD) http://orcid.org/0000-0002-9462-0264
Charlotte Lorenz (iD) http://orcid.org/0000-0003-3614-341X
Christian A Hanke (iD) http://orcid.org/0000-0002-4826-4908
Holger Gohlke (iD) http://orcid.org/0000-0001-8613-1447
Johann P Klare (iD) http://orcid.org/0000-0002-5761-5968
Andreas M Stadler (iD) http://orcid.org/0000-0003-2272-5232
Claus AM Seidel (iD) http://orcid.org/0000-0002-5171-149X

## Decision letter and Author response

Decision letter https://doi.org/10.7554/eLife.79565.sa1
Author response https://doi.org/10.7554/eLife.79565.sa2

# Additional files

## Supplementary files
• MDAR checklist

## Data availability

The following material is available at Zenodo in two locations: Experimental data (https://doi.org/10.5281/zenodo.6534557): (i) fluorescence decays recorded by eTCSPC used to compute the distance restraints in *Appendix 1—table 1* and *Appendix 3—table 1*, (ii) single-molecule multiparameter fluorescence data: all raw data, burst selection and calibration measurements, fFCS (filters and generated correlation curves) (iii) double electron-electron resonance (DEER) EPR data used for structural modeling, (iv) neutron spin-echo data and SAXS structure factor of hGBP1. Scripts for structural modeling of conformational ensembles through integrative/hybrid methods using FRET, DEER and SAXS together with the initial and selected structural ensembles (https://doi.org/10.5281/zenodo.6565895). The experimental SAXS data and the ab initio analysis thereof are available in the SASBDB (ID SASDDD6). Structure models of hGBP1 based on experimental restraints were deposited to PDB-Dev (PDB-Dev ID: PDBDEV_00000088) using the FLR-dictionary extension (developed by PDB and the Seidel group) available on the IHM working group GitHub site (https://github.com/ihmwg/flrCIF). Further data sets generated during and/or analyzed during the current study are available from the corresponding author on request.

The following datasets were generated:

| Author(s) | Year | Dataset title | Dataset URL | Database and Identifier |
|---|---|---|---|---|
| Peulen TO, Hengstenberg CS, Biehl R, Dimura M, Lorenz C, Valeri A, Folz J, Hanke CA, Ince S, Vöpel T, Farago B, Gohlke H, Klare JP, Stadler AM, Seidel CAM, Herrmann C | 2023 | SASDDD6 – Human Guanylate-binding protein (hGBP1) | https://www.sasbdb.org/data/SASDDD6/ | Small Angle Scattering Biological Data Bank, SASDDD6 |
| Peulen TO, Hengstenberg CS, Biehl R, Dimura M, Lorenz C, Valeri A, Folz J, Ince S, Vöpel T, Farago B, Gohlke H, Klare JP, Stadler AM, Seidel CAM, Herrmann C | 2023 | Neutron spin echo and intramolecular FRET and DEER-EPR measurements on hGBP1 (human guanylate binding protein 1) | https://doi.org/10.5281/zenodo.6534557 | Zenodo, 10.5281/zenodo.6534557 |
| Peulen TO, Hengstenberg C, Biehl R, Dimura M, Lorenz C, Valeri A, Folz J, Hanke C, Ince S, Vöpel T, Farago B, Gohlke H, Klare JP, Stadler AM, Seidel CAM, Herrmann C | 2023 | Screening routine for integrative dynamic structural biology using SAXS and intramolecular FRET and DEER-EPR on hGBP1 (human guanalyte binding protein 1) | https://doi.org/10.5281/zenodo.6565895 | Zenodo, 10.5281/zenodo.6565895 |

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

# Appendix 1

## Experimental results

**Appendix 1—table 1.** Inter-label distance analysis of DEER measurements, ensemble fluorescence decays of the donor (eTCSPC), and residual donor fluorescence anisotropies.

Average distances between the spin-labels are referred to as $\langle R_{LL,exp} \rangle$. The width of the inter-spin distance distribution is $w$. The center values of the donor-acceptor distance distribution correspond to $\bar{R}_{DA,exp}(M_1)$ and $\bar{R}_{DA,exp}(M_2)$ for the states, $M_1$ and $M_2$, respectively. The average donor-acceptor distance and the inter-spin distance simulated for the full-length crystal structure of hGBP1 (PDB-ID: 1DG3) are $\bar{R}_{DA,sim}$ and $\langle R_{LL,sim} \rangle$, respectively, with corresponding distribution widths $w$. The uncertainty estimates of central distance of a state determined by FRET is $\Delta\left(M_{\{1,2\}}\right)$.

| Category | DEER[a] | | DEER[a] | | ensemble FRET[b] | | ensemble FRET[b] | | Fluorescence Anisotropy[e] |
|---|---|---|---|---|---|---|---|---|---|
| Type | Experiment | | Simulation[c] | | Experiment | | Simulation[d] | | Experiment |
| State | Average over all states | | Crystal (PDB-ID: 1DG3) | | $M_1$ | $M_2$ | Crystal (PDB-ID: 1DG3) | | Donor Alexa488 |
| Joint species fractions $x_1$, $x_2$ | | | | | 0.61 | 0.39 | | | |
| Variant | $\langle R_{LL,exp} \rangle$ / Å | $(w)$ / Å | $\langle R_{LL,sim} \rangle$ / Å | $(w)$ / Å | $\bar{R}_{DA,exp}(M_1)$ $\pm\Delta(M_1)$ / Å | $\bar{R}_{DA,exp}(M_1)$ $\pm\Delta(M_2)$ / Å | $\bar{R}_{DA,sim}$ / Å | $(w)$ / Å | $r_\infty$ |
| N18C/Q344C | 64.6 | (12.2) | 66.5 | (9.5) | 73.6±8.6 | 67.0±5.5 | 72.4 | (12.1) | 0.15 |
| N18C/V540C | | | | | 57.8±3.6 | 36.6±2.7 | 63.6 | (8.8) | 0.11 |
| N18C/Q577C | 53.2 | (7.6) | 50.5 | (7.5) | 64.2±4.7 | 47.4±2.8 | 60.1 | (10.5) | 0.15 |
| C225/K567C | 12.0 | (5.5) | 13.0 | (6.5) | | | | | |
| C225/Q577C | 22.9 | (6.0) | 19.5 | (10.0) | | | | | |
| Q254C/Q344C | | | | | 81.3±16.5 | 72.3±7.8 | 73.9 | (12.2) | 0.13 |
| Q254C/V540C | | | | | 63.6±4.6 | 36.8±2.7 | 60.9 | (12.4) | 0.30 |
| Q254C/V577C | | | | | 70.8±9.1 | 52.9±7.4 | 73.1 | (8.9) | 0.17 |
| Q344C/T481C | | | | | 37.9±2.6 | 54.5±3.3 | 57.6 | (11.2) | 0.11 |
| Q344C/A496C | 40.0 | (9.6) | 42.5 | (10.0) | 48.0±2.9 | 23.5±8.2 | 48.4 | (10.6) | 0.19 |
| Q344C/Q525C | 32.0 | (5.4) | 29.5 | (10.2) | 46.7±2.8 | 20.7±13.3 | 41.5 | (10.7) | 0.30 |
| Q344C/V540C | 46.6 | (6.8) | 43.0 | (14.5) | 59.3±3.8 | 45.5±2.8 | 48.7 | (11.9) | 0.10 |
| T481C/Q525C | | | | | 69.6±6.4 | 69.5±6.4 | 71.2 | (11.0) | 0.30 |
| A496C/V540C | | | | | 63.6±4.6 | 63.6±4.6 | 66.8 | (11.6) | 0.23 |
| A551C/Q577C | 20.5 | (7.7) | 22.5 | (5.0) | | | | | |

[a] DEER distance distributions for calculation of average inter-spin distances and width were determined by Tikhonov regularization of the experimental DEER-traces (Materials and methods, **Equation 6**). All data are shown in **Figure 2—figure supplement 2**.

[b] The ensemble fluorescence decays of the donors were jointly analyzed by a quasi-static homogeneous model (**Peulen et al., 2017**) with two FRET species with the species fractions $x_1$ and $x_2$ as well as a D-only species (Materials and methods, **Equations 12; 13**) using the donor properties in **Appendix 1—table 2** and a Förster radius $R_0$=52 Å. Moreover, the model accounted for the distance distribution with a typical width of 12 Å caused by the flexible dye-linkers (Materials and methods, **Equation 13**). The reported uncertainty estimates, indicated by ±, include statistical uncertainties, potential systematic errors of the references, uncertainties of the orientation factor determined by the anisotropy of donor samples, and uncertainties of the AVs due to the differences of the donor and acceptor linker length (Appendix 2). The individual uncertainty components are listed in **Appendix 1—table 4**. Reference measurements of single D and A labeled variants are summarized in **Appendix 1—table 2**, respectively. The distances recovered by eTCSPC and seTCSPC (**Appendix 1—table 5**), respectively, agree nicely within the distinct precision of each data set.

[c] For EPR-DEER the inter-spin distance distribution was calculated by a rotamer library analysis.

[d] The inter- fluorophores distance distribution and the corresponding average distance and width were calculated by accessible volume simulations (**Kalinin et al., 2012**).

[e] Residual anisotropies of Alexa488 in FRET labeled (Alexa488, Alexa647) variants of hGBP1 determined by an analysis of the MFD histograms using a Perrin equation for a bi-exponential anisotropy decay (Appendix 2, **Equation 21**). All data are shown in **Figure 3—figure supplement 2**.

**Appendix 1—table 2.** Fluorescence lifetimes of Alexa647 and Alexa488 coupled to single cysteine variants of hGBP1 determined from ensemble TCSPC. Fluorescence lifetimes, τ, with corresponding species fractions, x, of Alexa647- and Alexa488-maleimide coupled to different single cysteine variants of the hGBP1 determined by an analysis of the fluorescence intensity decays measured by ensemble TCSPC.

| Variant | Alexa647[a] | | | | | | | | Alexa488[a] | | | | | |
| --- | --- | --- | --- | --- | --- | --- | --- | --- | --- | --- | --- | --- | --- | --- |
| | $\tau_1$/ ns | $(x_1)$ | $\tau_2$/ ns | $(x_2)$ | $\tau_3$/ ns | $(x_3)$ | $\langle\tau\rangle_x$/ ns[b] | $\Phi_{F,A}$[c] | $\tau_l$/ ns | $(x_1)$[a] | $\tau_2$/ ns | $(x_2)$ | $\langle\tau\rangle_x$/ ns[b] | $\Phi_{F,D}$[c] |
| N18C | 1.85 | (0.39) | 1.22 | (0.49) | 0.10 | (0.12) | 1.33 | 0.40 | 4.15 | (0.82) | 1.35 | (0.18) | 3.65 | 0.82 |
| Q254C | 2.23 | (0.58) | 1.42 | (0.42) | | | 1.89 | 0.57 | 3.60 | (0.69) | 0.53 | (0.31) | 2.65 | 0.59 |
| Q344C | 2.06 | (0.58) | 1.09 | (0.42) | | | 1.75 | 0.52 | 3.80 | (0.94) | 1.00 | (0.06) | 3.63 | 0.81 |
| T481C | 1.89 | (0.43) | 1.32 | (0.57) | | | 1.57 | 0.43 | 3.78 | (0.93) | 1.07 | (0.07) | 3.59 | 0.81 |
| A496C | 1.21 | (0.43) | 1.88 | (0.57) | | | 1.59 | 0.48 | 3.89 | (0.84) | 1.14 | (0.16) | 3.44 | 0.77 |
| Q525C | 1.93 | (0.65) | 1.08 | (0.35) | | | 1.63 | 0.49 | 3.51 | (0.80) | 0.66 | (0.20) | 2.94 | 0.66 |
| V540C | 2.33 | (0.65) | 1.43 | (0.35) | | | 2.02 | 0.60 | 4.00 | (0.85) | 1.86 | (0.15) | 3.67 | 0.82 |
| Q577C | 2.06 | (0.49) | 1.42 | (0.51) | | | 1.73 | 0.52 | 4.15 | (0.91) | 1.49 | (0.09) | 3.91 | 0.88 |

[a] The number of fluorescence lifetime components corresponds to the minimum number required to sufficiently describe the experimental data, as judged by a $\chi_r^2$ criterion (Materials and methods, **Equation 10**).

[b] $\langle\tau\rangle_x$ is the species weighted average fluorescence lifetime $\langle\tau\rangle_x = \sum_i (x_i \cdot \tau_i)$.

[c] $\Phi_F$ is the fluorescence quantum yield of the fluorescent dye species estimated by the species averaged fluorescence lifetime $\langle\tau\rangle_x$ using $\langle\tau\rangle_x$ and $\Phi_F$ of the free dyes as a reference; (Alexa647, $\langle\tau\rangle_x$=1.0 ns, $\Phi_{F,A}$ = 0.32) (Alexa488, $\langle\tau\rangle_x$=4.1 ns, $\Phi_{F,D}$ = 0.92).

**Appendix 1—table 3.** Fluorescence quantum yields of donor and acceptor dye in FRET samples under single-molecule measurement conditions to compute the FRET-lines in the E-diagrams (*Figure 3—figure supplement 2*).

| FRET sample | Donor[a] | Acceptor | $\Phi_{F,D}$ [b] | $\Phi_{F,A}$ [c] | $a_{trans}$ [d] |
|---|---|---|---|---|---|
| N18C/V344C | V344C | N18C | 0.83 | 0.38 | 0.75 |
| N18C/V540C | V540C | N18C | 0.76 | 0.38 | 0.69 |
| N18C/Q577C | N18C | Q577C | 0.84 | 0.47 | 0.77 |
| Q254C/Q344C | Q344C | Q254C | 0.82 | 0.50 | 0.70 |
| Q254C/W540C | Q540C | Q254C | 0.81 | 0.49 | 0.69 |
| Q344C/T481C | T481C | Q344C | 0.81 | 0.40 | 0.73 |
| Q344C/A496C | A496C | Q344C | 0.85 | 0.37 | 0.73 |
| Q344C/Q525C | Q344C | Q525C | 0.81 | 0.47 | 0.76 |
| Q344C/V540C | V540C | Q344C | 0.82 | 0.43 | 0.65 |
| T481C/Q525C | T481C | Q525C | 0.81 | 0.41 | 0.67 |
| A496C/V540C | V540C | A496C | 0.87 | 0.38 | 0.65 |
| Q254C/Q577C | Q577C | Q254C | 0.90 | 0.47 | 0.67 |

[a] Donor/acceptor position as determined by limited trypsin proteolysis of single labeled reference samples and double labeled FRET sample described by *Rothwell et al., 2013*.

[b] $\Phi_{F,D}$=Fluorescence lifetime of the reference (4.1 ns, $\Phi_{F,D}$=0.9) and correction for triplet amplitude (donor dye FCS measurement at same power).

[c] $\Phi_{F,A}$=Fluorescence lifetime reference dye and correction for triplet,

[d] $a_{trans}$ fraction of Alexa647 in trans state was estimated as previously described (*Widengren et al., 2001*; compare *Appendix 1—table 2*).

**Appendix 1—table 4.** Combined and individual uncertainty contributions of the average inter-dye distances determined by eTCSPC measurements.

| Variant | State | Combined uncertainty[a] | | Statistical uncertainty[b] | Reference uncertainty[c] | | Dye simulation[d] | Orientation factor[e] |
|---|---|---|---|---|---|---|---|---|
| | | $\delta_+$ | $\delta_-$ | $\delta_{stat}$ | $\delta_{+,ref}$ | $\delta_{-,ref}$ | $\delta_{AV}$ | $\delta_{,\kappa2}$ |
| N18C/Q344C | (1) | 12.1% | 11.3% | 9.1% | 6.0% | 4.2% | 1.1% | 4.9% |
| N18C/V540C | (1) | 6.3% | 6.3% | 3.1% | 1.2% | 1.1% | 1.4% | 4.5% |
| N18C/Q577C | (1) | 7.4% | 7.3% | 4.6% | 2.3% | 2.0% | 1.2% | 4.9% |
| Q254C/Q344C | (1) | 22.0% | 18.6% | 16.5% | 13.5% | 6.9% | 1.0% | 4.7% |
| Q254C/V540C | (1) | 7.2% | 7.1% | 4.4% | 2.2% | 1.9% | 1.3% | 5.8% |
| Q344C/T481C | (1) | 6.9% | 6.9% | 4.1% | 0.1% | 0.1% | 2.1% | 4.5% |
| Q344C/A496C | (1) | 6.0% | 6.0% | 2.5% | 0.4% | 0.4% | 1.7% | 5.4% |
| Q344C/Q525C | (1) | 6.0% | 6.0% | 2.5% | 0.3% | 0.3% | 1.7% | 6.1% |
| Q344C/V540C | (1) | 6.5% | 6.4% | 3.4% | 1.4% | 1.3% | 1.3% | 4.3% |
| T481C/Q525C | (1) | 9.4% | 9.1% | 6.7% | 4.0% | 3.1% | 1.2% | 6.1% |
| A496C/V540C | (1) | 7.2% | 7.2% | 4.4% | 2.2% | 1.9% | 1.3% | 5.7% |
| Q254C/Q577C | (1) | 13.0% | 12.7% | 11.0% | 4.5% | 3.4% | 1.1% | 5.1% |
| N18C/Q344C | (2) | 8.1% | 7.9% | 5.6% | 3.1% | 2.5% | 1.2% | 4.9% |
| N18C/V540C | (2) | 7.3% | 7.3% | 4.6% | 0.1% | 0.1% | 2.2% | 4.5% |
| N18C/Q577C | (2) | 5.8% | 5.8% | 2.5% | 0.4% | 0.3% | 1.7% | 4.9% |
| Q254C/Q344C | (2) | 10.9% | 10.3% | 8.3% | 5.3% | 3.8% | 1.1% | 4.7% |

*Appendix 1—table 4 Continued on next page*

*Appendix 1—table 4 Continued*

| Variant | State | Combined uncertainty[a] | | Statistical uncertainty[b] | Reference uncertainty[c] | | Dye simulation[d] | Orientation factor[e] |
|---|---|---|---|---|---|---|---|---|
| | | $\delta_+$ | $\delta_-$ | $\delta_{stat}$ | $\delta_{+,ref}$ | $\delta_{-,ref}$ | $\delta_{AV}$ | $\delta_{,\kappa 2}$ |
| Q254C/V540C | (2) | 7.7% | 7.7% | 4.5% | 0.1% | 0.1% | 2.2% | 5.8% |
| Q344C/T481C | (2) | 5.5% | 5.5% | 2.7% | 0.8% | 0.8% | 1.5% | 4.5% |
| Q344C/A496C | (2) | 34.8% | 34.8% | 34.2% | 0.0% | 0.0% | 3.4% | 5.4% |
| Q344C/Q525C | (2) | 64.6% | 64.6% | 64.2% | 0.0% | 0.0% | 3.9% | 6.1% |
| Q344C/V540C | (2) | 5.4% | 5.4% | 2.6% | 0.3% | 0.3% | 1.8% | 4.3% |
| T481C/Q525C | (2) | 10.0% | 9.7% | 6.7% | 4.0% | 3.1% | 1.2% | 6.1% |
| A496C/V540C | (2) | 7.6% | 7.5% | 4.4% | 2.2% | 1.9% | 1.3% | 5.7% |
| Q254C/Q577C | (2) | 14.0% | 14.0% | 12.9% | 0.7% | 0.7% | 1.5% | 5.1% |

[a] The combined uncertainty used to calculate the relative ($\delta_\pm$) and the absolute ($\Delta_\pm$) uncertainties of the inter-dye distances considers the uncertainty of the dye simulations, $\delta_{AV}$, the uncertainty of the orientation factor, $\delta_{,\kappa2}$, the uncertainty of the reference sample, $\delta_{\pm,ref}$, and the statistical uncertainty determined by the noise of the measurements, $\delta_{stat}$ (Appendix 2, **Equation 23**).

[b] The statistical uncertainties were estimated by sampling the distances that agree with the experimental ensemble TCSPC data (**Appendix 1—table 1**).

[c] The reference uncertainties were calculated assuming an uncertainty of the reference donor fluorescence lifetime of $\Delta\tau_{D(0)} = 0.15ns$.

[d] The dye simulation error considers the labeling uncertainty of the dye. In accessible volume simulations (AV) for the dye pair Alexa488/Alexa647 this labeling uncertainty results in an expected error of $\Delta AV = 0.8$ Å (**Vöpel et al., 2014**).

[e] The uncertainty of the distance due to the orientation factor was estimated using a wobbling in a cone model of the dyes using the experimental anisotropies (**Appendix 1—table 1**).

**Appendix 1—table 5.** Complementary inter-dye distance analysis of donor and sensitized acceptor fluorescence decays of the FRET-active DA sub-ensemble from single-molecule FRET experiments. Complementary inter-dye distance analysis of donor and sensitized acceptor fluorescence decays of the FRET-active DA sub-ensemble (seTCSPC) (**Figure 3—figure supplement 3**) obtained from of single-molecule FRET experiments (**Figure 3—figure supplement 2**) for different FRET labeled (Alexa488, Alexa647) hGBP1 variants.

The donor and acceptor fluorescence decays were described by a combination of two normal distributed distances with the central distances of $\bar{R}_{DA}$ of a state. The fractions $x_1$ and $x_2$ correspond to the fraction of the distance $\bar{R}_{DA}(M_1)$ and $\bar{R}_{DA}(M_2)$, respectively. $x_{DOnly}$ is the fraction of molecules with no energy transfer to an acceptor. The distances recovered by eTCSPC (**Appendix 1—table 1**) and seTCSPC, respectively, agree nicely within the distinct precision of each data set.

| hGBP1 variant | $\bar{R}_{DA}(M_1)$ / Å | $x_1$ | $\bar{R}_{DA}(M_2)$ / Å[a] | $x_2$[a] | $x_{DOnly}$ |
|---|---|---|---|---|---|
| N18C/Q344C | 69.3 | 1.00 | - | - | 0.18 |
| N18C/V540C | 60.0 | 0.50 | 34.1 | 0.50 | 0.37 |
| N18C/Q577C | 63.3 | 0.65 | 45.1 | 0.35 | 0.17 |
| Q254C/Q344C | 76.1 | 1.00 | - | - | 0.21 |
| Q254C/V540C | 63.4 | 0.74 | 39.3 | 0.26 | 0.31 |
| Q344C/T481C | 45.0 | 0.44 | 59.3 | 0.56 | 0.17 |
| Q344C/A496C | 47.0 | 0.77 | 36.0 | 0.23 | 0.45 |
| Q344C/Q525C | 51.5 | 0.63 | 36.1 | 0.37 | 0.40 |
| Q344C/V540C | 57.8 | 0.65 | 43.8 | 0.35 | 0.24 |
| T481C/Q525C | 70.6 | 1.00 | - | - | 0.20 |
| A496C/V540C | 63.9 | 1.00 | - | - | 0.37 |

*Appendix 1—table 5 Continued on next page*

*Appendix 1—table 5 Continued*

| hGBP1 variant | $\bar{R}_{DA}\,(M_1)$ / Å | $x_1$ | $\bar{R}_{DA}\,(M_2)$ / Å [a] | $x_2$ [a] | $x_{DOnly}$ |
|---|---|---|---|---|---|
| Q254C/Q577C | 70.8 | 0.84 | 52.9 | | 0.16 | 0.34 |

[a] For cases where a single normal distribution was enough to describe the data no second distance and fraction is reported.

**Appendix 1—table 6.** Filtered fluorescence correlation spectroscopy (fFCS) of all FRET-labeled variants of the human guanylate binding protein 1 analyzed by a global model with joint relaxation times and individual amplitudes $A_i$.

Selection criteria for the definition of the filters for the high FRET (HF) and low FRET (LF) species.

| hGBP1 variant | Amplitude [a] | | | Burst selection criteria for fFCS filter generation [b] | | | |
|---|---|---|---|---|---|---|---|
| | $A_1$ | $A_2$ | $A_3$ | HF range, $S_g/S_r$ | | LF range, $S_g/S_r$ | |
| N18C/Q344C | 0.38±0.09 | 0.50±0.18 | 0.12±0.20 | 0.09 | 2.03 | 5.82 | 13.19 |
| N18C/V540C | 0.18±0.03 | 0.37±0.04 | 0.45±0.06 | 0.10 | 1.00 | 4.00 | 10.80 |
| N18C/Q577C | 0.24±0.06 | 0.41±0.09 | 0.36±0.11 | 0.40 | 1.50 | 3.00 | 26.00 |
| Q254C/Q344C | 0.35±0.09 | 0.00±0.00 | 0.65±0.09 | 0.70 | 1.70 | 2.80 | 5.90 |
| Q254C/V540C | 0.34±0.03 | 0.21±0.05 | 0.45±0.06 | 0.18 | 0.56 | 3.50 | 8.90 |
| Q344C/T481C | 0.24±0.11 | 0.34±0.14 | 0.41±0.18 | 0.10 | 0.60 | 1.60 | 6.30 |
| Q344C/A496C | 0.08±0.03 | 0.30±0.06 | 0.62±0.07 | 0.03 | 0.58 | 1.70 | 22.36 |
| Q344C/Q525C | 0.00±0.05 | 0.45±0.09 | 0.55±0.10 | 0.07 | 0.28 | 2.94 | 8.07 |
| Q344C/V540C | 0.08±0.02 | 0.33±0.07 | 0.59±0.07 | 0.08 | 0.70 | 1.92 | 8.74 |
| T481C/Q525C | 0.58±0.13 | 0.09±0.05 | 0.32±0.14 | 0.60 | 1.52 | 16.27 | 25.40 |
| A496C/V540C | 0.21±0.07 | 0.32±0.17 | 0.47±0.18 | 0.07 | 1.77 | 4.61 | 11.40 |
| Q254C/Q577C | 0.34±0.07 | 0.45±0.11 | 0.21±0.13 | 0.51 | 1.39 | 5.18 | 11.07 |
| Correlation time / µs | 297.6 | 22.6 | 2.0 | | | | |

[a] The correlation times were determined by a joint/global analysis of fFCS curves (Materials and methods, **Equation 19**, **Figure 3—figure supplement 4**). The amplitudes $A_1$, $A_2$, and $A_3$ are variant specific. The uncertainties were determined by a support plane analysis, which considers the mean and the standard deviation of the individual correlation channels determined by splitting the measurements into smaller sets.

[b] Filters defining the high FRET (HF) and the low FRET (LF) species were generated by selecting bursts that are within the given ranges. To select high FRET and low FRET bursts the ratio of the green and red signal intensity ratio, $S_g/S_r$, was used. Sub-ensemble fluorescence decay histograms of the molecules in these ranges were generated and used to calculated filters for fFCS as previously described (**Felekyan et al., 2012**).

## Appendix 2

### Sample quality, uncertainty estimation, and consistency analysis

Quality assessment of labeled samples for fluorescence spectroscopy

Mutations and labels introduced to different sites of a protein may influence the conformations the protein adopts. Thus, any kind of modification is a putative cause of alterations in protein structure, function, and activity and may, in the worst-case, invalidate conclusions of following experiments. Moreover, for labels that specifically interact with the studied biomolecule, modelling the positional distribution of a labels by their sterically allowed accessible volume (AV) and/or accessible contact volume (ACV) may lead to inaccurate structural models. The ACV explicitly the fraction of fluorescent dyes bound to the molecular surface. The fraction of bound dyes was estimated by the residual anisotropy.

To address these general concerns, we: (**1**) select potential labeling sites based on biochemical pre-knowledge, e.g., we avoid active/catalytic sites, (**2**) characterize the effect of the mutations on hGBP1's activity, (**3**) measure the rotational mobility of the fluorescence dyes by their anisotropy, (**4**) use the fluorescence quenching of the donor dyes by their environment in combination with coarse-grained simulations as an indicator for their translational mobility. By (**2**) and (**3**) we probe the effect of the mutations and the labels on the protein. By (**4**) we assure correct references for accurate analysis results of the fluorescence decay. By (**3**) and (**4**) we test the applicability of the coarse-grained AV and ACV model to describe the spatial distribution by which we recover for a given structural model the theoretical spectroscopic properties.

### (1) Selection of labeling sites

To avoid alteration of protein function (nucleotide binding and hydrolysis, oligomerization), neither amino acid positions in direct proximity to the nucleotide binding pocket nor inside the G domain dimerization interface nor charged amino acids on the protein surface were taken into consideration for labeling (*Tsodikov et al., 2002*). All chosen positions had an accessible surface area (ASA) value higher than 60 Å$^2$. In the end, eight amino acids distributed over the entire protein were chosen.

### (2.1) hGBP1's function: Effect of mutations

The used cysteine mutants are based on a cysteine-free hGBP1 construct where all nine native cysteines were mutated to alanine or serine namely: C12A, C82A, C225S, C235A, C270A, C311A, C396A, C407S and C589S. Previously, these mutations were shown to only weakly affect hGBP1's function (*Vöpel et al., 2009*; *Vöpel et al., 2014*). Before introduction of new cysteines for site-specific labelling the *GTPase activity* and *nucleotide binding behavior* were characterized. The GTPase activity of the labeled and unlabeled hGBP1 variants was quantified by an assay as previously described (*Kunzelmann et al., 2006*). Briefly, hGBP1's hydrolytic activity of was controlled by high-performance liquid chromatography using a Chromolith Performance RP-18 end-capped column (Merck, Darmstadt) as described earlier (*Kunzelmann et al., 2006*). 1 µM of protein were incubated with 350 µM GTP at 25 °C. The samples were analyzed at different time points. The time dependence of the substrate concentration was used to calculate the specific activities of the different protein mutants (*Figure 2—figure supplement 3A*). The assay for measuring the protein activity has an error smaller than 10%. However, besides the relative activity the absolute uncertainty in determining the (active) protein concentration needs to be considered. Hence, the overall uncertainty in determining the absolute protein activities is ~30%. Except of A496C and Q344C/A496C, all mutants produced more GMP than GDP, as known for the wildtype hGBP1 (*Hengstenberg, 2013*).

### (2.2) hGBP1's function: Effect of labeling

To check if the fluorophores bound to cysteines in hGBP1 have an impact on the oligomerization behavior an unlabeled and a labeled construct were analyzed by analytical gel filtration in the presence and the absence of a nucleotide, which induces oligomerization (*Figure 2—figure supplement 3A*). We display the analysis for the variant N18C/Q577C, because N18C and Q577C are localized in proximity to dimerization interfaces of the LG and helix α13, respectively. The fluorophores are attached to the sulfhydryl group of the cysteines via a linker of ~20 Å in length. Thus, they potentially interfere with the self-oligomerization of hGBP1. However, the elugrams of the labeled and unlabeled N18C/Q577C did not show any differences (*Figure 2—figure supplement 3A*). This indicates that, at least for this mutant, the labels do not influence for hGBP1 assembly.

As shown for hGBP1 Cys9, no dimer formation was observed in the presence of 200 µM GppNHp independent of being labelled or not.

In addition to the biochemical activity assays that report on the hydrolytic activity of the GTPase domain, we performed single-molecule FRET measurements of the labeled protein (LP) in the presence of excess unlabeled protein (UP) and GTP or GDP-AlF$_x$ as a substrate. Under these conditions, hGBP1 forms a dimer and undergoes significant conformational changes as seen by the significant changes of the FRET efficiency in *Figure 3—figure supplement 5*. We found minor differences among three comparable hGBP1 variants, which are affected in the hydrolysis activity to a different degree by the presence of a fluorescent dye. Hence, we conclude that a fluorescent dye, which affects the hydrolysis activity due to its proximity to hGBP1's GTP binding site has only minor influence on the global domain arrangement that is of interest in this study.

## (2.3) hGBP1's function: Influence of temperature

Using a steady state fluorometer, we measured the variants T481C/Q525C, N18C/V540C, and N18C/Q577C. As anticipated, we found a larger change in the FRET efficiency in dependency of the temperature for the variants N18C/V540C and N18C/Q577C as compared to the variant T481C/Q525C (*Figure 2—figure supplement 3C(i)*). For T481C/Q525C M1 and M2 are merely indistinguishable (see distances). For these measurements, we found that the largest relative change of the populations happens between 10 °C and 25 °C. From these measurements, no absolute populations can be determined. Hence, we performed after we acquired a temperature-controlled time-resolved fluorescence spectrometer a temperature series. One measurement out of this set of measurements is shown below in *Figure 2—figure supplement 3C(ii)*. For this variant, we only found minor changes of the relative population of the states M$_1$ and M$_2$ (*Figure 2—figure supplement 3C(iii)*). We compared the different measured variants by normalizing the observed changes (*Figure 2—figure supplement 3C(iv)*). We found an average midpoint for all the variant of ~15 °C. Hence, the relative population of the states at higher temperatures as found in a living cell resembles the measurements at room temperature.

## (3) Rotational mobility of the fluorescence dyes

The rotational mobility of the dyes was probed by measuring their time-resolved anisotropy, $r(t)$, using multiparameter fluorescence detection in single-molecule experiments. A formal analysis of $r(t)$ by a multiexponential relaxation model reveals typically 'fast' and slow rotational correlation times $\rho_{fast} < 1$ ns and $\rho_{slow} > 20$ ns (*Figure 2—figure supplement 3*, **upper panel**). The fast component we attribute to the rotation of the dye tethered to the protein. The slow component we attribute to the dye which sticks to the protein surface and thus senses the global rotation of the protein. Hence, the anisotropy decay $r(t)$ reflects local motions of the dye and global rotations of the macromolecule

$$r\left(t\right) = \left[\left(r_0 - r_\infty\right) e^{\frac{-t}{\rho_{local}}} + r_\infty\right] e^{\frac{-t}{\rho_{global}}} \cong \left(r_0 - r_\infty\right) e^{\frac{-t}{\rho_{local}}} + r_\infty e^{\frac{-t}{\rho_{global}}} \qquad (21)$$

Above $r_0$ is the fundamental anisotropy (fixed to $r_0$=0.38), $\rho_{global}$ is the global rotation time, $\rho_{local}$ is the local rotation time, and $r_\infty$ is the residual anisotropy. The anisotropy difference $\left(r_0 - r_\infty\right)$ relates to the fraction of freely rotating dyes.

To determine $\left(r_0 - r_\infty\right)$ for the donor dyes, the two-dimensional single-molecule histograms of the steady-state anisotropy, $r_S$, and the fluorescence lifetime, $\tau$, were analyzed with a Perrin equation derived for dyes with a bi-exponential anisotropy decay (*Figure 2—figure supplement 3B*). In this analysis, $r_\infty$ was treated as an unknown parameter, which was determined by optimizing the Perrin equation to the experimental histogram (*Figure 3—figure supplement 2*, blue lines). The Perrin equation for two components is:

$$r_S\left(\tau\right) = \frac{r_0 - r_\infty}{1 + \frac{\tau}{\rho_{local}}} + \frac{r_\infty}{1 + \frac{\tau}{\rho_{global}}} \qquad (22)$$

Using the formalism described in *Dale et al., 1979*, we obtain $\kappa^2$ uncertainties $(\Delta R_{DA}(\kappa^2))$ corresponding to each FRET distance for $r_\infty$. Moreover, $r_\infty$ was used as estimate for the fraction of the dyes bound to the surface of the protein, to calibrate the dye's accessible surface volume

(ACV) as previously described (*Dimura et al., 2016*). The labeling-site specific $r_\infty$ are compiled in *Appendix 1—table 1*.

## (4) Translational mobility of the fluorescence dyes

For all possible labeling sites, we simulated expected fluorescence quantum yields of dynamically quenched donor dyes Alexa488 diffusing within its accessible volume (AV) and accessible surface volume (ACV) using Brownian dynamics simulations with previously published parameters (*Peulen et al., 2017*). Finding no significant differences to other reference sample, we corroborate that within the model errors AV/ACVs describe the dye behavior (*Figure 2—figure supplement 3B*, **bottom panel**).

To conclude, the introduced mutations and the labeling of the dyes has no major influences on the protein function, i.e., the GTP hydrolysis and the GTP induced self-oligomerization. The time-resolved anisotropy measurements and the dynamic quenching simulations agree with a donor dye freely rotating and diffusing within its AV/ACV.

## Uncertainty estimation

For comparison of an experimentally derived distance to the distances of a structural model different sources of uncertainties of an inter-dye distance need to be combined. Here, the reported estimates of the distance uncertainties consider relative uncertainties, $\delta$, of the accessible volume model (AV), $\delta_{AV}$, the orientation factor, $\delta_{\kappa^2}$, the reference, $\delta_{Reference,\pm}$, and the statistical noise of the data, $\delta_{stat,\pm}$. These uncertainties were combined to $\delta_{R_{DA},\pm}$, a relative uncertainty of the distance:

$$\delta_{R_{DA},\pm} = \sqrt{\delta_{AV}^2 + \delta_{\kappa^2}^2 + \delta_{Reference,\pm}^2 + \delta_{stat,\pm}^2} \tag{23}$$

$\delta_{AV}$ considers the fact that both dyes were conjugated to the protein by cysteines. Therefore, two FRET species, where the donor is either attached to the first amino acid (DA) or the second (AD), are present in the measured samples. As the donor and acceptor dyes have different geometries, the DA and AD species have distinct distributions of FRET rate constants. We previously demonstrated for the used dyes Alexa488 and Alexa647 well described by AVs, that differences in the FRET rate constant distribution between DA and AD species results in an uncertainty in the distance of $\Delta R_{DA,AV} \approx 1$. This uncertainty was considered by $\delta_{AV}^2 = R_{DA}/\Delta R_{DA,AV}$ (*Peulen et al., 2017*). The uncertainty $\delta_{\kappa^2}^2$ of orientation factor $\kappa^2$ was determined as previously described using a wobbling in a cone model considering the residual anisotropies of the dyes (*Kalinin et al., 2012*). The asymmetric uncertainty $\delta_{Reference,\pm}$ considered potential reference errors, propagating to systematic errors of an experimentally determined distance $R_{DA,exp}$.

The fluorescence rate constant of the donor in the absence of FRET, $k_D$, serves as reference to recover experimental distances, $R_{DA,exp}$, in the analysis of fluorescence decays. An inaccurate reference for $k_D$ propagates to systematic errors of $R_{DA,exp}$. We estimate the contribution of an inaccurate reference to $\delta_{R_{DA},\pm}$ by $\delta_{Reference,\pm}$

$$\delta_{Reference,\pm} = \left| 1 - \left( 1 \pm \left( \frac{R_{DA,exp}}{R_0} \right)^6 \cdot \delta_{k_D} \right)^{-\frac{1}{6}} \right| \tag{24}$$

Here, $R_0$ is the Förster radius and $\delta_{k_D}$ is the relative deviation of the experimentally determine $k_D$ from the correct (true) $k_D$. To estimate $\delta_{k_D}$ we use the sample-to-sample variation of the donor fluorescence lifetimes (*Appendix 1—table 2*). The contribution of the statistical error $\delta_{stat,\pm}$ was estimated by support plane analysis and a Monte-Carlo sampling algorithm determining distributions of parameters in agreement with the experimental data (*Straume et al., 2002*). Using the relative uncertainty estimates the absolute uncertainties of the distances were calculated and compiled in *Appendix 1—table 4*.

## Consistency analysis

### Consistency analysis identifies misassigned distances

The fluorescence decays were analyzed by a model function, which assigns distances to the states by their amplitude. The model free analysis of the DEER data recovered inter-spin distance

distributions, $p(R_{LL})$, which reflect all conformational heterogeneities with unclear assignment to the corroborated states. The DEER analysis assigns no states to the recovered distributions. Therefore, initially all DEER constraints were assigned to $M_1$ *and* $M_2$ using the width of the distributions as uncertainty. This assignment resulted in structural models inconsistent with the data (*Figure 2— figure supplement 3D*). The DEER measurements on C225C/K567C and C225C/Q577C revealed short distances, highlighted by the fast-initial drop of the form factors (*Figure 2—figure supplement 2A*, gray traces). Models consistent with $M_2$ predicted long distances (>5 nm) beyond the DEER detection limit at this measurement settings for these variants, *Figure 2—figure supplement 2*, green traces (for ~6–7 nm). Hence, C225C/K567C and C225C/Q577C were considered only to model $M_1$ for highly valuable information on the position of the short helix α13 relative to helix α12. This assignment resulted in a consistent combined set of distances for FRET and DEER used for RBD (*Appendix 3—table 1*). Concluding, by analysis of the self-consistency of the data with the models, we unambiguously assigned recovered distances or average distances to biomolecular states.

## Consistency analysis finds very good agreement between eTCSPC and seTCSPC measurements

In this study, we actually performed two kinds of FRET experiments that yielded three data information altogether: (1) On the ensemble level (*Figure 2C*): eTCSPC to analyze the time-resolved fluorescence decay of the donor (*Appendix 1—table 1*, *Appendix 1—table 2*). (2) On the single-molecule level (*Figure 3—figure supplement 2*): sm MFD measurements allowed us to determine two data information for the FRET efficiencies and interdye distances of all samples by intensity-based (*Appendix 1—table 3*) and time-resolved methods (*Figure 3—figure supplement 3*, *Appendix 1— table 5*). The inter-dye distance analysis of donor and sensitized acceptor fluorescence decays of the FRET-active DA sub-ensemble (seTCSPC) yields the interdye distances compiled in *Appendix 1— table 5* The FRET-lines in the ⟨$\tau_{D(A)}$⟩ - E diagrams in *Figure 3—figure supplement 2* confirm the internal consistency of both intensity and time-resolved FRET indicators. Moreover, they display the dynamics of hGBP1 by the dynamic shift from the static FRET line (*Barth et al., 2022*; *Opanasyuk et al., 2022*).

Altogether, both methods lead to highly consistent results within the precision of the measurements.

# Appendix 3

## Integrative modeling

### Simulation of experimental parameters

Briefly, theoretical SAXS scattering curves for the structural models were calculated using CRYSOL (*Svergun et al., 1995*). DEER and the FRET inter-label distance distributions $p_{sim}(R_{LL}, M)$ were simulated by accessible volume (AV) simulations. The experimental inter-label distances were compared to the average simulated distances (*Figure 4D*). For a given protein conformation M the average simulated distance for all label linker conformations is

$$\langle R_{LL,sim} \rangle (M) = \int R_{LL} \cdot p_{sim}(R_{LL}, M) dR_{LL} \tag{25}$$

Because of the different meaning of the experimental DEER and FRET inter-label distances, the modeled average inter-spin distances $\langle R_{LL,sim} \rangle$ and the center-to-center inter-dye distances $\bar{R}_{DA,sim}$ are denoted in *Figure 4D* with the general symbol $R_{LL,sim}$ . The DEER AV simulations were calibrated against rotamer library approaches (*Figure 2—figure supplement 2B*). The AV simulations of the fluorophores were refined using experimental anisotropies to account for dyes bound to the molecular surface in accessible contact volume (ACV) simulations. The model for the fluorescent dyes were validated by reference measurements and protein activity measurements (Appendix 2).

In detail, the distribution of the labels was modeled by accessible volume (AV) simulations weighted by the fraction of dyes in contact with the protein - accessible contact volume (ACV) (*Cai et al., 2007*; *Muschielok et al., 2008*; *Sindbert et al., 2011*). The used ACV approach determines all sterically allowed positions of a label and weights the fraction of dyes in contact with the protein by experimental anisotropies (Appendix 2) (*Dimura et al., 2016*). The AV and ACV labels are approximated by ellipsoids connected by a linker to the $C_\beta$-atoms of the reactive amino-acid. The linker extends from the reactive group to the center of the dipole of the labels. The dyes were simulated with previously published parameters (*Sindbert et al., 2011*; *Kalinin et al., 2012*). The donor (Alexa Fluor 488 C5 maleimide, Alexa488) and the acceptor fluorophore (Alexa Fluor 647 C2 maleimide, Alexa647) were modeled using a linker width $L_{width}$ of 4.5 Å and linker-length $L_{link}$ of 20.5 Å and 22 Å for Alexa488 and Alexa647, respectively. The radii of the ellipsoids ($R_{dye1}$, $R_{dye2}$ and $R_{dye3}$) for Alexa488 were 5.0 Å, 4.5 Å and 1.5 Å and for Alexa647 11.0 Å, 4.7 Å and 1.5 Å, respectively. For ACV the residual anisotropy was used to estimate the fraction of dyes bound to the protein surface (*Dimura et al., 2016*). Methanethiosulfonate (MTSSL) AV parameters were determined by comparing AV simulated $p(R_{SS})$ to established simulation approaches (*Polyhach et al., 2011*; *Hagelueken et al., 2012*) resulting in linker-length of 8.5 Å, a linker-width of 4.5 Å, and an ellipsoid radius of 4.0 Å (*Figure 2—figure supplement 2B*).

The fluorescence data provides two central distances $\bar{R}$ per variant assigned and conformation $M_1$ and $M_2$. The DEER data provided distance distributions that were considered by their average distance $\langle R_{LL,exp} \rangle$ . Contrary to the simulated distances, the experimental distance is a linear combination of the distances of the two co-existing conformations.

### Structure representation

The used RBD-framework represents proteins as an assembly of flexible linked rigid bodies interacting via a very soft repulsion (clash) potential which tolerates atomic overlaps to a certain degree (*Kalinin et al., 2012*). hGBP1 was decomposed into its individual domains: the LG domain (aa 1–309), the middle domain (aa 310–481) and the helices α12 (aa 482–563) and α13 (aa 564–583) for RBD (*Figure 4—figure supplement 1*, *Figure 4B*). To allow for internal reorganization the middle domain is represented by two rigid bodies (aa 310–373, aa 374–481). The N- to the C-terminal parts of the rigid bodies were connected via bonds with a weak quadratic potential. Such reduced model does not allow for bending of the individual domains. Therefore, we used a very soft clash-potential.

### Generation of structures

As first step to generate structures we use RBD with DEER and FRET restrains. Average distances between the labels were determined by modeling their spatial distribution of the labels around their

attachment point (*Kalinin et al., 2012*). Deviations between the modeled and the experimental FRET and DEER distances were minimized by Monte-Carlo sampling driving initial random configurations the rigid-body assembly towards an optimal conformation. The restraints are compiled in *Appendix 3—table 1*. This docking procedure was repeated 20,000 times for $M_1$ and $M_2$ to generate structural models refined by subsequent NMSim and MD simulations (*Figure 4A*). Next, the RBD structures were refined by NMSim. NMSim generates representations with stereochemical accurate conformations by a three-step protocol and incorporates information about preferred directions of protein motions into a geometric simulation algorithm (*Ahmed and Gohlke, 2006*). In targeted NMSim the conformational change vector is formulated as a linear combination of the modes calculated for the starting structure (the crystal structure) weighted by the proximity to the target structure (the RBD structure). This way, the normal modes that overlap best with the direction of conformational change contribute more to the direction of motion in NMSim. The NMSim refined structures were clustered into 343 and 414 groups by their $C_\alpha$ RMSD for the states $M_1$ and $M_2$, respectively, using hierarchical agglomerative clustering with complete linkage and distance threshold of 5 Å. As final step, conventional MD simulations on the group representatives were performed for 2 ns. The MD trajectories were clustered using hierarchical agglomerative clustering with complete linkage and distance threshold of 2 Å into 3395 and 3357 groups for $M_1$ and $M_2$, respectively.

## Individual ranking of structures

For a consistent description of the FRET, DEER, and the SAXS experiments, the experimental scattering curve was described by a mixture of the conformations. For maximum parsimony the DEER, FRET and SAXS measurements were described by two states $M_1$ and $M_2$. Here, the disagreement of the simulated and experimental data was measured by weighted sums of squared deviations, $\chi^2$. The structural models were compared to the SAXS and to the combined DEER and FRET dataset by $\chi^2_{SAXS}$ and $\chi^2_{DEER,FRET}$, respectively.

For SAXS $\chi^2_{SAXS}$ was computed by first computing scattering curves for all proposed structures using the program CRYSOL. Next, model functions $I_{model}(q, M_1, M_2)$ for all possible combinations of structural models for the states $M_1$ and $M_2$ were calculated. The model functions were linear combinations of $F_{M1}(q)$ and $F_{M2}(q)$, the theoretical scattering curves for $M_1$ and $M_2$, respectively.

$$I_{model}(q, M_1, M_2) = x_{M1} \cdot F_{M1}(q) + (1 - x_{M1}) \cdot F_{M2}(q) \tag{26}$$

To determine the initially unknown fraction of molecules in the $M_1$ state, $x_{M1}$, the sum of weighted squared deviations between the experiment and the data $\chi^2_{SAXS}$ to the measured data, $I_{exp}(q)$ was minimized.

$$\chi^2_{SAXS}(M_1, M_2) = \frac{1}{N} \sum_{i=1}^{N} \left( \frac{I_{exp(q_i)} - I_{model}(q_i, M_1, M_2)}{\sigma(q_i)} \right)^2 \tag{27}$$

Above, $\sigma(q_i)$ is the noise of the experimental scattering curve and $N$ is the number of detection channels.

For DEER and FRET combined $\chi^2_{DEER,FRET}$ was computed using simulated distances and experimental distances considering the asymmetric (deviation dependent) uncertainty of the distances. For a pair of structural models ($M_1$, $M_2$) we approximate $\chi^2_{DEER,FRET}$ by:

$$\chi^2_{DEER}(M_1, M_2)$$
$$\approx \sum_i \left( \frac{\left\langle R_{LL,exp}^{(i)} \right\rangle (M_1) - \left\langle R_{LL,sim}^{(i)} \right\rangle (M_1)}{w^{(i)}(M_1)/2} \right)^2$$
$$+ \sum_i \left( \frac{\left\langle R_{LL,exp}^{(i)} \right\rangle (M_2) - \left\langle R_{LL,sim}^{(i)} \right\rangle (M_2)}{w^{(i)}(M_2)/2} \right)^2$$

$$\chi^2_{FRET}\left(M_1, M_2\right)$$

$$\approx \sum_i \left(\frac{\left\langle R^{(i)}_{DA,exp}\right\rangle (M_1) - \left\langle R^{(i)}_{DA,sim}\right\rangle (M_1)}{\Delta^{(i)}(M_1)}\right)^2$$

$$+ \sum_i \left(\frac{\left\langle R^{(i)}_{DA,exp}\right\rangle (M_2) - \left\langle R^{(i)}_{DA,sim}\right\rangle (M_2)}{\Delta^{(i)}(M_2)}\right)^2$$

$$\chi^2_{DEER,FRET}\left(M_1, M_2\right) = \chi^2_{DEER}\left(M_1, M_2\right) + \chi^2_{FRET}\left(M_1, M_2\right) \tag{28}$$

$\bar{R}^{(i)}_{DA,exp}(M_1)$ and $\bar{R}^{(i)}_{DA,exp}(M_2)$ are the central experimental donor-acceptor FRET distances assigned to M$_1$ and M$_2$. $\left\langle R^{(i)}_{LL,exp}\right\rangle \left(M_{\{1,2\}}\right)$ is the average label-label distance in a DEER experiment. Modeled average inter-label distances $\left\langle R^{(i)}_{LL,sim}\right\rangle \left(M_{\{1,2\}}\right)$ correspond to the average simulated label-label distance $\langle R_{LL,sim}\rangle$ for DEER and average simulated donor-acceptor distance and $\left\langle R^{(i)}_{DA,sim}\right\rangle \left(M_{\{1,2\}}\right)$ for FRET, which is a good approximation for the central donor-acceptor distance $\bar{R}_{DA,exp}$ of a symmetric distance distribution being used for analysis. Uncertainties that depend on the sign of the deviation between the model and the data were considered by the half width of the distance distribution $w^{(i)}\left(M_{\{1,2\}}\right)/2$ for DEER and estimate for the uncertainty of the central distance $\Delta^{(i)}\left(M_{\{1,2\}}\right)$ for FRET.

## Model discrimination and quality assessment

The SAXS, DEER, and FRET provide different information (inter-label distances in DEER, FRET *vs.* average shapes in SAXS). The relative weight of the techniques affects the final structures. In well-balanced structures reflect the information content. To FRET, DEER measurements with SAXS we estimate the information content of the measurement with respect to the structure and combine the experiments in a meta-analysis.

The values for $\chi^2_{DEER,FRET}$ and $\chi^2_{SAXS}$ assess the quality in a pair of structural models with respect to the experiment. We use these $\chi^2$ values to identify/filter models that are significantly worse than the best possible pair of structures for the respective methods. For that, we compare pairs of $\chi^2$ values for structural models by an F-test (The ratio $x := \chi^2_1/\chi^2_2$ is F-distributed). For two $\chi^2$-values with corresponding degrees of freedom $d_1$ and $d_2$ the cumulative F distribution is:

$$F\left(x, d_1, d_2\right) = \mathbf{I}_{\frac{d_1 x}{d_1 x + d_2}}\left(\frac{d_1}{2}, \frac{d_2}{2}\right) \tag{29}$$

Here, $\mathbf{I}$ is the regularized incomplete beta function. To relate the F-value $x$ to a probability $\alpha$, for given $\chi^2_1$ and $\chi^2_2$ and significantly different d$_1$, and d$_2$, we must compute the inverse of the cumulative F distribution.

We compare the $\chi^2$ value of all possible combinations of structural models (M$_1$, M$_2$) and experimental techniques DEER/FRET and SAXS F-values to the $\chi^2$ value of best pair of structures $(x = \chi^2/min(\chi^2))$. These models have the same dofs. Hence, we first identify the best model and compute $x$ for all pairs of models. Next, we determine the degrees of freedom, dof, that are calculated by the degrees of freedom of the data, dof$_d$, and the degrees of freedom of the model, dof$_m$, i.e. dof = dof$_m$ - dof$_d$. With $x$ and dof we compute *the probability $\alpha$ that a model is significantly worse than the best model*.

dof$_m$ was estimated by a PCA applied to all structural models. PCA revealed that 10 principal components explain more than 90% of the total variance. Hence, we conclude that dof$_m$ ~10. For DEER/FRET, dof$_{d,DEER/FRET}$ was estimated by correcting the total number of inter-label distances (*Appendix 3—table 1*: 22 FRET, 8 DEER) for duplicates and for redundant mutual information content. This was accomplished by determining the number of informative distances via a greedy backward elimination feature selection algorithm for our total ensemble (*Dimura et al., 2016*,

*Figure 5*) so that the precision of the obtained corresponded to our experimental one. In this way, we obtained a $dof_{d,DEER/FRET}$ = 22 (*Figure 4—figure supplement 2A*) - a value that is close to the number of independent label-pair positions of 23. For SAXS measurements the number of Shannon channels is typically in the range of 10–23. For our measurements, the number of Shannon channels approximately 18–22 (*Moore, 1980*; *Mertens and Svergun, 2017*). We used the number of Shannon channels as an initial estimate for the dof of the SAXS measurements, $dof_{d,SAXS}$, and we varied $dof_{d,SAXS}$ in the range of 10–24. We found only minor effects of $dof_{d,SAXS}$ on $\alpha_{SAXS}$ , the SAXS discimination power of the models, and used $dof_{d,SXAS}$ = 17 to discriminate structural models (*Figure 4—figure supplement 2A*). Using these estimates of $dof_m$ and $dof_d$, $\alpha$ for DEER/FRET,$\alpha_{DEER,FRET}$, and SAXS, $\alpha_{SAXS}$, were calculated for all pairs ($M_1$, $M_2$). Next, $\alpha_{DEER,FRET}$ and $\alpha_{SAXS}$ were combined in a meta-analysis to a joint probability of discriminating a pair.

$\alpha_{DEER,FRET}$ and $\alpha_{SAXS}$ measure how likely a pair ($M_1$, $M_2$) is dissimilar from the best pair for DEER/FRET and SAXS, respectively. To combine DEER/FRET and SAXS we used the probability $p$ that a pair is similar, $p=1-\alpha$. Note, $p$ for DEER/FRET, $p_{DEER,FRET}$, and for SAXS, $p_{SAXS}$, considers the degrees of freedom for the system and data. Moreover, $p_{DEER,FRET}$ and $p_{SAXS}$ are independent. Thus, Fisher's method was applied to fuse datasets in a meta-analysis. Fisher's method combines probabilities of $k$ independent tests (here $k=2$) into a combined $\chi^2_{2k}$ with 2 k degrees of freedom. For $p_{DEER,FRET}$ and $p_{SAXS}$, the combined probability is

$$\chi^2_{2k} \sim -2 \cdot \sum_{i=1}^{k} ln\,(p_i) = -2 \cdot ln\left(\prod_{i=1}^{k} p_i\right)$$

$$= -2 \cdot ln(p_{DEER,FRET} \cdot p_{SAXS})$$

(30)

Thus, $\chi^2_{2k}$ is chi-squared distributed with 4 combined degrees of freedom. In this way a $\chi^2_{2k}$ value was determined for every ($M_1$, $M_2$), and pairs ($M_1$, $M_2$) were discriminated by a chi-squared test with 4 degrees of freedom.

## Assessment of model precision and quality

To assess the local quality of the models, the inter-residue distances between all $C_\alpha$ atoms, $R_{C_\alpha}$ , and the standard deviation, $SD\left(R_{C_\alpha}\right)$, of the distribution of $R_{C_\alpha}$ were calculated for all models as a measure for the experimental model precision (*Figure 4E*, lower triangles). Next, we checked if these variabilities are larger than statistically expected. For this, we compared the experimental precision $SD\left(R_{C_\alpha}\right)$ to the precision $SD\left(R_{C_\alpha}\right)_{ref}$ of a ground truth model ensemble as an 'ideal and perfect' reference by computing the weighted (normalized) precision, $SD\left(R_{C_\alpha}\right)/SD\left(R_{C_\alpha}\right)_{ref}$ . Due to (*i*) the incomplete experimental information on the model, (*ii*) the uncertainties of the experiments, and (*iii*) imprecisions of the model, we anticipate a limited resolution of the model even for ideal experiments. We calculated the reference precision of the ground truth ensembles in two steps. At first, we use the models for $M_1$ and $M_2$ of the experimental ensemble that describe our FRET and EPR data best. Next, we use the distances corresponding to the best models and our experimental errors in *Appendix 3—table 1* to generate the ideal reference ensemble by our structural modeling pipeline (*Figure 4A*) so that we could compute the theoretical inter-residue distance distributions and precisions. The finally computed distributions of the weighted precisions, $SD\left(R_{C_\alpha}\right)/SD\left(R_{C_\alpha}\right)_{ref}$ allow us to test whether the modeled conformational ensemble approaches the theoretical optimum ratios around unity or whether systematic deviations indicate problems in the modelling.

Please note that the above procedure provides only an estimate for the reference model precision and corresponding variability of $R_{C_\alpha}$ . For a correctly estimated model precision with the corresponding $SD\left(R_{C_\alpha}\right)_{ref}$ , the weighted precision $SD\left(R_{C_\alpha}\right)/SD\left(R_{C_\alpha}\right)_{ref}$ theoretically has the meaning of an F-value. Such an F-value for pair-wise estimates of the model precision as $SD\left(R_{C_\alpha}\right)$ could be used for estimating the probability that the model insufficiently describes the data within their experimental noise. This procedure could yield such estimates for residue pairs that facilitate the detection of the model defects and limitations.

## Rigid body docking (RBD)

### Identification of rigid domain and rigid body decomposition

The first step for RBD is the segmentation of hGBP1 into rigid domains that can represent the essential motions of the protein (*Figure 3F*). Moreover, the segmentation should introduce sufficient degrees of freedom to fulfil the experimental constraints. The substrate free (PDB-ID: 2B92) and a substrate bound form of the LG domain (PDB-ID: 2B92) differ by only 1.1 Å RMSD, and the distance network (*Figure 1A*) for the label-based measurements were designed to probe distance changes between the LG, the middle domain, and α12/13. Hence, the LG domain (residue 1–309) was modeled as a single RB. In the MD simulations α12/13 moved relative to the middle and the LG domain while the middle domain changes its orientation relative to the LG domain. The DEER measurement on the variant A551C/Q577 informs on the position of α13 relative to helix α12. Consequently, α12 (residue 482–563) and α13 (residue 564–584) were treated as separate RBs. We note that we treated helix α12 in this work as rigid, because we had no experimental evidence for unfolding or kinking of this helix. Without a rearrangement of the middle domain the motion of α12/13 is highly restricted in a RBD framework. To allow for more flexibility the middle domain was represented by two bodies (residue 310–373 and residue 374–481). Overall, hGBP1 was decomposed into the LG domain, two RBs for the middle domain, helix α12, and helix α13. Therefore, to capture hGBP1's motions and fulfil the experimentally probed degrees of freedom hGBP1 was described by five RBs. Experimental evidence for such a decomposition is the FRET measurements on the variants A496C/V540C, T481C/Q525C, and Q344C/T481C which probe the conformation of α12 and the middle domain. An analysis of the protein mechanics by the $S^2$ order parameters of the NH bond determined by analysis of the MD simulations and the B-factors of the full-length protein revealed sets of characteristic spikes (*Figure 4—figure supplement 1D*). These spikes rationalize the rigid body decomposition and identify flexible regions of the protein, which mainly correspond to flexible loops connecting individual helices. They indicate that the middle domain, helix α12, and helix α13 are flexibly linked.

### Definition of restraints

As next step of RBD, a set of restraints needs to be defined. The RBD procedure uses for model generation experimental restraints, a repulsion potential as a penalty function for atomic overlaps, and bonds between the RBs. The experimental restraints, bonds connecting the bodies, and the repulsive clash are considered by the terms $\chi^2_{DEER,FRET}$ (Appendix 3, *Equation 28*), $\chi^2_{bonds}$, and, $\chi^2_{clash}$, respectively. The overall RBD potential $\chi^2_{RBD}$ is

$$\chi^2_{RBD} = \chi^2_{DEER/FRET} + \chi^2_{bonds} + \chi^2_{clash}. \tag{31}$$

The bond term was a combination of quadratic potentials

$$\chi^2_{bonds} = \sum_{i,j} \left( \frac{R_{ij} - R_{eq,ij}}{k_{ij,\pm}} \right)^2 \tag{32}$$

where $R_{ij} = \left| \vec{r}_j - \vec{r}_i \right|$ is the distance between the vectors $\vec{r}_j$ and $\vec{r}_i$, defined by the arrangement of the RBs, $R_{eq,ij}$ is the equilibrium distance of the distance pair, and $k_{ij,\pm}$ is a constant which depends on the sign of $R_{ij} - R_{eq,ij}$.

Overlaps of rigid bodies we penalized by the atomic overlaps in the repulsion potential $\chi^2_{clash}$:

$$\chi^2_{clash} = \sum_{i,j} \begin{cases} 0 & , R_{ij} \geq R_{wi} + R_{wj} \\ \left( \frac{R_{wi} + R_{wj} - R_{ij}}{R_{ctol}} \right)^2 & , R_{ij} < R_{wi} + R_{wj} \end{cases} \tag{33}$$

Here $R_{ij}$ is the distance between the atoms with the index $i$ and $j$ belonging to different subunits, $R_{wi}$ and $R_{wj}$ are their van der Waals radii, and $R_{ctol}$ is a constant defining the "clash tolerance".

The restraints defining $\chi^2_{DEER,FRET}$ for $M_1$ and $M_2$ are listed in the *Appendix 3—table 1*. Parameters defining contributions to $\chi^2_{bonds}$, namely the connection of the N- and C-termini and a set of weak bonds based on a crystal structure (PDB-ID: 1DG3) to stabilize the middle domain, are listed in

**Appendix 3—table 2.** The rigid body model does not allow for bending. Therefore, a very weak repulsion potential was used for $\chi^2_{clash}$ ($R_{ctol}$ = 6 Å).

## Rigid body docking procedure

The final docking step generates structural models fulfilling the constraints summarized by the energy function (Appendix 3, **Equation 31**). Starting from a random initial arrangement of the RBs, forces drive the RB assembly towards a configuration with a minimum energy (**Kalinin et al., 2012**). For fast calculations, the forces are applied between the average label position and optimizes the distance between the average label positions $R_{mp}$ .

$$R_{mp} = \left| \left\langle \vec{r}_{L1}^{(i)} \right\rangle - \left\langle \vec{r}_{L2}^{(j)} \right\rangle \right| = \left| \frac{1}{n}\sum_{i=1}^{n} \vec{r}_{L1}^{(i)} - \frac{1}{m}\sum_{j=1}^{m} \vec{r}_{L2}^{(j)} \right|$$

(34)

Here $\vec{r}_{L1}^{(i)}$ and $\vec{r}_{L2}^{(j)}$ are the coordinates of the two labels in the conformation ($i$) and ($j$). Using the mean position of the dyes instead of the full spatial distribution of the dyes reduces the complexity of the RBD and increases its speed (**Kalinin et al., 2012**). DEER and fluorescence decays measurements recover inter-label distance distributions, $p(R_{LL})$, and not the distance between the label positions. For a uniformly populated AV

$$\langle R_{LL} \rangle = \left| \left\langle \vec{r}_{L1}^{(i)} \right\rangle - \left\langle \vec{r}_{L2}^{(j)} \right\rangle \right| = \frac{1}{nm}\sum_{i=1}^{n}\sum_{j=1}^{m} \left| \vec{r}_{L1}^{(i)} - \vec{r}_{L2}^{(j)} \right|$$

(35)

To use average distances $\langle R_{LL} \rangle$ during RBD a transfer function converted the experimental average inter label distance $\langle R_{LL} \rangle$ to $R_{mp}$ (**Kalinin et al., 2012**). After docking, the spatial distribution of the labels were simulated for the generated structural models to calculate average distances $\langle R_{LL} \rangle$ as a set of model distances, {$R_{model}$}.

**Appendix 3—table 1.** Experimental restraints for rigid body docking.
Analysis results of DEER-EPR and FRET eTCSPC. The labels Alexa488, Alexa647, and MTSSL are referred to by D, A, and R1, respectively. The names of the labeling sites report on the location of the dyes and the introduced mutation. For DEER, average inter-label distances $\langle R_{LL,exp} \rangle$ and the widths of the inter-spin distance distribution $w_+$ and $w_-$ are reported (**Appendix 1—table 1**). For FRET, mean distances $\bar{R}_{DA,exp}$ (**Appendix 1—table 1**) and uncertainties of the mean are reported ($\Delta_+$ and $\Delta_-$) (**Appendix 1—table 4**). The measurements are grouped into three classes. Class (I) informs on $M_1$ and $M_2$ by two distinct distances. Class (II) informed on $M_1$. In class (III) measurements $M_1$ and $M_2$ were not resolved into separate states. The simulated average label-to-label distances correspond to the distances of the pair of structures (M1, M2) best agreeing with SAXS, DEER, and FRET combined (**Figure 5—figure supplement 1**).

| | | Experiment | | | | Experiment | | Simulation | | |
| | | Labelling site[a] | | $M_1$ | | $M_2$ | | $M_1$ | $M_2$ | |
| # | Technique | 1 | 2 | $\bar{R}_{DA,exp}$ / Å | ($\Delta_+$; $\Delta_-$) / Å | $\bar{R}_{DA,exp}$ / Å | ($\Delta_+$; $\Delta_-$) / Å | $\bar{R}_{DA,sim}$ / Å | | Class |
|---|---|---|---|---|---|---|---|---|---|---|
| 1 | FRET | Q344C[D] | N18C[A] | 73.6 | (8.9; 8.3) | 67.0 | (5.6; 5.4) | 73.9 | 62.3 | (I) |
| 2 | FRET | V540C[D] | N18C[A] | 57.8 | (3.6; 3.6) | 36.6 | (2.7; 2.7) | 63.9 | 40.3 | (I) |
| 3 | FRET | N18C[D] | Q577C[A] | 64.2 | (4.7; 4.7) | 47.4 | (2.8; 2.8) | 62.3 | 54.8 | (I) |
| 4 | FRET | Q344C[D] | Q254C[A] | 81.3 | (17.9; 15.1) | 72.3 | (8.0; 7.6) | 78.0 | 79.8 | (I) |
| 5 | FRET | V540C[D] | Q254C[A] | 63.6 | (4.6; 4.5) | 36.8 | (2.7; 2.7) | 63.2 | 41.3 | (I) |
| 6 | FRET | Q344C[D] | T481C[A] | 37.9 | (2.6; 2.6) | 54.4 | (3.3; 3.3) | 49.2 | 57.3 | (I) |
| 7 | FRET | A496C[D] | Q344C[A] | 48.0 | (2.9; 2.9) | 23.5 | (8.2; 8.2) | 43.4 | 38.3 | (I) |
| 8 | FRET | Q344C[D] | Q525C[A] | 46.7 | (2.8; 2.8) | 20.7 | (13.3; 13.3) | 43.6 | 28.0 | (I) |

*Appendix 3—table 1 Continued on next page*

*Appendix 3—table 1 Continued*

| | | Labelling site[a] | | Experiment M$_1$ | | Experiment M$_2$ | | Simulation M$_1$ | M$_2$ | |
|---|---|---|---|---|---|---|---|---|---|---|
| # | Technique | 1 | 2 | $\bar{R}_{DA,exp}$ / Å | $(\Delta_+; \Delta_-)$ / Å | $\bar{R}_{DA,exp}$ / Å | $(\Delta_+; \Delta_-)$ / Å | $\bar{R}_{DA,sim}$ / Å | | Class |
| 9 | FRET | Q344C$^D$ | V540C$^A$ | 59.3 | (3.8; 3.8) | 45.5 | (2.8; 2.8) | 53.1 | 45.5 | (I) |
| 10 | FRET | Q577C$^D$ | Q254C$^A$ | 70.8 | (9.2; 9.0) | 52.9 | (7.4; 7.4) | 75.2 | 35.4 | (I) |

| # | Technique | 1 | 2 | $\langle R_{ss,exp} \rangle$ / Å | $(w_+; w_-)$ / Å | | | $\langle R_{ss,sim} \rangle$ / Å | | |
|---|---|---|---|---|---|---|---|---|---|---|
| 11 | DEER | C225C$^{R1}$ | K567C$^{R1}$ | 12.0 | (5.5; 5.5) | –[b] | - | 16.4 | 58.1 | (II) |
| 12 | DEER | C225C$^{R1}$ | Q577C$^{R1}$ | 22.9 | (6.0; 6.0) | –[b] | - | 29.6 | 50.1 | (II) |

| | | | | M$_1$ & M$_2$ | M$_1$ & M$_2$ | | | M$_1$ | M$_2$ | |
|---|---|---|---|---|---|---|---|---|---|---|
| | | | | $\bar{R}_{DA,exp}$ / Å | $(\Delta_+; \Delta_-)$ / Å | | | $\bar{R}_{DA,sim}$ / Å | | |
| 13 | FRET | T481C$^D$ | Q525C$^A$ | 69.6 | (6.6; 6.3) (4.6; 4.6) | | | 68.6 | 72.7 | (III) |
| 14 | FRET | A496C$^D$ | V540C$^A$ | 63.6 | | | | 71.7 | 70.7 | (III) |

| # | Technique | 1 | 2 | $\langle R_{SS,exp} \rangle$ / Å | $(w_+; w_-)$ / Å | | | $\langle R_{SS,sim} \rangle$ / Å | | |
|---|---|---|---|---|---|---|---|---|---|---|
| 15 | DEER | N18C$^{R1}$ | Q344C$^{R1}$ | 64.6 | (12.2; 12.2) | | | 73.3 | 60.9 | (III) |
| 16 | DEER | N18C$^{R1}$ | Q577C$^{R1}$ | 53.2 | (7.6; 7.6) | | | 63.9 | 51.6 | (III) |
| 17 | DEER | A551C$^{R1}$ | Q577C$^{R1}$ | 20.5 | (7.7; 7.7) | | | 33.0 | 20.5 | (III) |
| 18 | DEER | Q344C$^{R1}$ | A496C$^{R1}$ | 40.0 | (9.6; 9.6) | | | 38.8 | 35.8 | (III) |
| 19 | DEER | Q344C$^{R1}$ | Q525C$^{R1}$ | 32.0 | (2.7; 2.7) | | | 36.7 | 25.0 | (III) |
| 20 | DEER | Q344C$^{R1}$ | V540C$^{R1}$ | 46.6 | (6.8; 6.8) | | | 51.4 | 47.9 | (III) |
| | | | | $x_r^2$ | | | | 2.07 | 1.89 | |

[a] The names of the labelling sites report on the most likely position of the donor and the acceptor dyes. The distribution among the labelling sites was determined by an analysis of the time-resolved anisotropy decay, anisotropy PDA, and limited proteolysis of the labelled protein.

[b] A consistency analysis (**Figure 2—figure supplement 3D**) identifies that M$_2$ must have long distances ($\langle R_{SS,exp} \rangle > 5$ nm) beyond the DEER detection limit for this measurement setting (see Appendix 2 **consistency analysis**).

**Appendix 3—table 2.** Additional restraints for rigid body docking.

| Atom 1 Residue | Atom 1 Atom name | Atom 2 Residue | Atom 2 Atom name | $R_{eq}$ / Å | $k_{ij+}$ / Å | $k_{ij-}$ / Å | Restrain origin |
|---|---|---|---|---|---|---|---|
| 309 | C | 310 | N | 1.5 | 0.5 | 0.5 | Primary sequence |
| 373 | C | 374 | N | 1.5 | 0.5 | 0.5 | Primary sequence |
| 481 | C | 482 | N | 1.5 | 0.5 | 0.5 | Primary sequence |
| 563 | C | 564 | N | 1.5 | 0.5 | 0.5 | Primary sequence |
| 445 | Cα | 348 | Cα | 5.1 | 2 | 2 | X-ray PDB-ID: 1DG3 |
| 391 | Cα | 336 | Cα | 8 | 4 | 4 | X-ray PDB-ID: 1DG3 |
| 381 | Cα | 527 | Cα | 8.2 | 4 | 4 | X-ray PDB-ID: 1DG3 |
| 323 | Cα | 292 | Cα | 9.3 | 4 | 4 | X-ray PDB-ID: 1DG3 |

