## [Editor Report]

This study uses a broad range of experimental and theoretical biophysical techniques to provide fundamental insights into the conformational pathway of the human dynamin-related GTPase guanylate-binding protein 1 from its resting to an active state. The convincing integrative approach identifies hitherto hidden, dynamic conformers. The work will be of interest to the communities of experimental and theoretical protein biophysics and protein dynamics in signal transduction.

---

## [Decision Letter]

**Decision letter after peer review:**

Thank you for submitting your article "Integrative dynamic structural biology unveils conformers essential for the oligomerization of a large GTPase" for consideration by *eLife*. Your article has been reviewed by 4 peer reviewers, including Hannes Neuweiler as the Reviewing Editor and Reviewer #1, and the evaluation has been overseen by José Faraldo-Gómez as the Senior Editor.

The reviewers have discussed their reviews with one another, and the Reviewing Editor has drafted this letter to help you prepare a revised submission.

All four reviewers are in support of publishing your manuscript in a revised form. The reviewers were impressed by the breadth and depth of biophysical methods and analyses applied to the complex problem of deciphering conformational heterogeneity and dynamics of hGBP1. However, as you will see from the reviews below, there are certain concerns that need to be addressed before publication. Required revisions include additional experiments on a farnesylated variant of hGBP1, which is the physiologically relevant form, and the application of GTP as a nucleotide in ligand-binding experiments (see comments from reviewers #3 and #4). It is also suggested that more mechanistic information is contained in the MD simulation data.

We ask you to address all comments of the reviewers with a point-by-point response letter highlighting changes.

*Reviewer #1 (Recommendations for the authors):*

The manuscript "Integrative dynamic structural biology unveils conformers essential for the oligomerization of a large GTPase" by Peulen et al. reports investigations on the solution structural ensemble and dynamics of human guanylate binding protein 1 (hGBP1) using a combination of complementary biophysical techniques. The authors apply single-molecule fluorescence resonance energy transfer (smFRET) spectroscopy on a number of donor/acceptor fluorophore pairs labeled to engineered cysteine (Cys) residues covering the entire structure of hGBP1. Time-resolved smFRET spectroscopy is used to determine distance distributions and correlation spectroscopy to study dynamics. A subset of double Cys mutants designed for FRET experiments was channeled into double electron-electron resonance (DEER) measurements, complementing the determination of distance distributions. The shape of hGBP1 in solution and its dynamics were further characterized using small-angle x-ray scattering (SAXS) and neutron spin echo (NSE) spectroscopy. In-silico studies were carried out using MD simulations and integrative modelling. The authors arrive at the conclusion of having resolved a novel native-state conformer of hGBP1 together with its time scales of interconversion, which is part of a mechanism of structural remodeling of hGPB1 dimers that form competent precursors in the human host defense system.

Uncovering protein functional mechanisms requires knowledge of both structure and dynamics. Dynamic structural biology is an emerging and important research field. Methods and their applications to observe changes in protein conformation in solution lag behind the ever-increasing number of high-resolution structures solved. The GTPase hGPB1 is a central player in cell-autonomous immunity, which acts by cycling between dimeric, polymeric, and membrane-bound states on infectious pathogens driven by the binding and hydrolysis of GTP. This manuscript shows, within one piece of work, the impressive application of a combination of five different biophysical methods that complement each other in the exploration of spatial and temporal scales of protein conformation. The conformational heterogeneity and dynamics of hGBP1 detected in this work report on a hitherto unknown intermediate state in the conversion of hGBP1 from the resting to an active state. Some uncertainties remain in some of the experimental designs applying extensive mutagenesis and modification.

I have the following comments.

The authors use smFRET to measure distance distributions and apply a subset of the same Cys mutants to also measure distances using DEER. Can the authors comment on the difference in spatial scales probed by both techniques? To which degree are they complementary or does one technique benchmark/test the other?

In Figure 2A, the fit of the crystal structure (blue line) overlays well with the Kratky plot of SAXS data. Still, the authors argue that the crystal structure would disagree with the SAXS data, which appears unreasonable from looking at the fit to the data.

On page 7 the authors argue that observed deviations of distances from the crystal structure and reduced spread, measured by DEER, could originate from a denser packing of the spin labels. How can a packing of spin labels in a solution structure be denser than in that of a crystal, which is optimally packed? The DEER/SAXS result appears to be in conflict with the conformational spread detected by smFRET. Won't a conformationally heterogeneous solution-structural ensemble always represent a more loose structure than that of a crystal? The discrepancies are also mentioned in the paragraph summary, in lines 211-212 on page 10, but appear not to be reconciled.

Wild type hGBP1 contains nine native Cys residues. The authors designed their Cys mutants for FRET and DEER spectroscopy on the background of a construct where all nine native Cys residues were replaced through mutagenesis to prevent their interference with site-specific modification. Replacement of such a high number of native Cys can potentially cause structural/energetic perturbations. Although the authors state in supporting note 1 that these mutations would only weakly affect hGBP1's function, some of the single and double Cys mutants (labeled/unlabeled), shown in Supporting Figure 4A, exhibit significant deviations/reductions of activity. What is the activity of the wild-type protein that contains all native Cys? It would be nice to see the wild-type activity also in the plots in Figure 4A for comparison. Can the authors exclude that their modifications induce conformational heterogeneity detected in smFRET experiments (the M1/M2 conformational ensemble)?

As a result of smFRET correlation spectroscopy, the authors find that dynamics in the middle and effector domain are an order of magnitude faster (~14 µs) than in the LG GTPase domain (~140 µs), highlighted in Figure 3E. This is an interesting finding. Since the secondary structure is the same throughout all hGBP1 domains one may have expected similarity in kinetics. The detected differences may report on functional motions. smFRET correlation spectroscopy covers the ns-µs time scale, which is indeed the relevant scale of motion of protein secondary and tertiary structure. NSE spectroscopy nicely complements the faster time scale of up to 200 ns. This faster time scale is also covered by MD simulations. Can we learn more details about the onset of the µs conformational changes seen in experiments from the computed atomistic MD data?

Collective motions of protein domains are often slower than ms, in general. There may thus be functionally relevant motions of hGBP1 that were missed out. Can the authors comment on such potential motions and how they could be detected?

The authors applied GDP-AlFx to generate a nucleotide-bound hGBP1 construct that remained monomeric under solution conditions of smFRET spectroscopy. It would be interesting to see how dimerization influences the measured hGBP1 dynamics. This could be studied by smFRET through the application of an excess of unlabeled hGBP1 to fluorescently modified, GDP-AlFx-bound hGBP1 construct. Have the authors considered such experiments?

Binding of GMP is known to stabilize the compact resting state of hGBP1 (reviewed in Kutsch & Coers, FEBS J. 2021, 288, 5826-5849). Have the authors considered experiments using GMP as a ligand? Such experiments would be interesting because one would expect that binding of GMP stalls the dynamics of hGBP1 and thus the M1/M2 motions. Experiments involving the binding of GMP could provide insights into the functional relevance of the detected M1/M2 motions.

Model (iii), "dimer induced flexibility", in Figure 1B seems to be ruled out by the fact that the binding of GTP triggers the opening of GBP1 (reviewed in Kutsch & Coers, FEBS J. 2021, 288, 5826-5849).

The last sentence of the abstract aims to highlight the conclusions of the work, which is deciphering intrinsic flexibility and conformational heterogeneity of hGBP1, but also mentions the GTP-triggered association of GTPase domains, assembly-dependent GTP hydrolysis, and functional design principles that control hGBP1's reversible oligomerization reported by others.

*Reviewer #2 (Recommendations for the authors):*

In their work, Peulen et al. investigate the conformational flexibility of the GTPase hGBP1, a dynamin-like protein responsible for the membrane disruption of intracellular parasites. The work was motivated by the apparent disagreement of the X-ray structure of the monomer with the previous FRET- and DEER experiments that suggested at least two conformations of the protein in its GTP-induced dimeric state. This disagreement indicates substantial flexibility, either in the monomeric state, which might or might not be induced by ligand binding or in the dimer. To understand the origin of this flexibility, the authors combined an impressive number of techniques, SAXS, NSE (neutron-spin echo spectroscopy), EPR, and FRET, in an integrative approach to unravel the structures and interconversion timescales of conformers in the apo- and holo-form of the protein. The authors found that at least two conformers (M1 and M2) interconvert at timescales from a few up to hundreds of microseconds. Integrative modeling combining the experimental results with rigid-body docking and structural refinement of a coarse-grained model of hGB1, following the previously published protocol of the authors (Dimura et al. 2020 Nature Comms, 11, 5394), was used to determine low-resolution models of M1 and M2. These models give rise to two potential dimers M1:M2 and M1:M1. The structural differences in M1 and M2 suggested by this integrative approach are substantial and indicate two different binding sites of the α12/13 helices on the ligand-binding (LG) domain.

In some parts, the article is written in a rather technical style, which might complicate an understanding by readers unfamiliar with the techniques and approaches (structural biologists for instance). In addition, I have a number of technical comments that the authors should address.

1. Figure 1B of the manuscript gives the impression that the authors investigate flexibility in the apo- and holo-form of the monomer but also in the dimer. I was therefore wondering why the authors excluded experiments in the presence of an excess of unlabeled monomer to investigate flexibility in the dimer. Does the formation of higher oligomers in the presence of GTP and high protein concentrations prevent such experiments?

2. Why are M1 and M2 not seen in the experimental DEER distance distributions, e.g., in the Q344C/A496C variant? Do the conformations re-equilibrate during the fast cooling period required in EPR? In fact, all EPR distributions are unimodal (p. 7, and Supplementary Figure 2), which is difficult to understand given the evidence for the existence of M1 and M2 from TCSPC and MFD histograms.

3. I was a bit puzzled by the eTCSPC analysis. The authors find bi-exponential lifetime decays for the donor in the absence of an acceptor, which is explained by the environment of the dyes after labeling. This indicates the presence of conformers with different degrees of dynamic donor quenching and hence different radiative rates of the donor dye. Yet, the combination of eq. 9 and 10 in the Supporting Information does not account for the propagation of these differences in the transfer rate. In fact, eq. 10 indicates that the same radiative rate (k0) was used to determine distance distributions. Although this approach is convenient as it allows εD to be obtained by the ratio of donor decay in the presence of the acceptor and in the absence of the acceptor, I am not yet convinced the approach is justified. If

fD0 = ∑ xD(i) exp[-kD(i) t]

is the donor decay in the absence of the acceptor (sum goes over i components), I'd expected that the donor decay in the presence of acceptor is

fDA = ∑{xD exp[-kD(i) t] ∫exp[-kT(i,r) t]dr} (A)

with kT(i,r)=kD(i) (R0/r)^6 being the component-dependent transfer rate and r is the donor-acceptor distance. Instead, the authors used eq. 9 and 10, which give

fDA = {∑xD exp[-kD(i) t] } ∫exp[-kT(r) t]dr (B)

with kT(r)=kD (R0/r)^6 with kD = k0 = 1⁄τ0.

Clearly, A and B are two very different scenarios. In the latter, εD can simply be determined via fDA / fD0, but not in the former. Hence, if expression A is more adequate, a MEM-analysis of fDA / fD0 might provide false results.

4. Related to point 3, the sub-population-specific TCSPC decays in the single-molecule FRET experiments seem to indicate single-exponential decays for donor-only molecules, at least by visual inspection (see Supporting Figure 3B). How is this reconciled with the double-exponential decays found for singly donor labeled hGBP1 in the ensemble TCSPC experiments?

5. Have the authors checked whether static quenching of the acceptor (or donor) affects the ratiometric FRET efficiencies? Direct excitation of the acceptor in a (anti)-bunching experiment can reveal the significance of static quenching.

6. Supporting Figure 4: Panels (C) and (D) are switched in the caption.

7. It is known that D2O (used in the EPR experiment) can affect protein stabilities and flexibilities (Makhadatze et al. 1995 Nat Struct Biol, 2, 852 / Cioni & Strambini (2002) Biophys J, 82, 3246 / Pica & Graziano (2017) Biopolymers, 109 and many more). Did the authors find differences in the populations and or dynamics of the state between the two solvents? Controls with MFD should be straightforward.

*Reviewer #3 (Recommendations for the authors):*

In their work, Peulen et al. applied an integrative modeling approach to scrutinize the conformational dynamics of the human guanylate binding protein (hGBP1), which is an essential part of the cell's intrinsic immune response to intracellular parasites. The assortment of biophysical methods used is striking – the fact that the authors have been able to plan and assign experiments/simulations, collect and process data, and integrate the diverse data into a meta-analysis workflow to derive molecular candidates for hGBP1 conformers is exemplary and not self-evident. This has allowed deriving important conclusions about the mechanism of hGBP1 self-assembly. In particular, the authors propose that hGBP1 undergoes GTP-independent conformational transitions on the µs-timescale between at least two conformational states, which is essential for the slower, GTP-dependent hGBP1 oligomerization. It is highly unlikely that these conclusions could ever have been made without the cross-disciplinary scheme presented by the authors. On the other hand, it is also crucial to note that the study has to incorporate several caveats that will be important when placing the proposed oligomerization pathway (Figure 6 of the paper) in a broader biological context and that several clarifications will need to be provided to delineate the conclusions based on measurements/calculations actually done by the authors from speculations based on physics intuition and/or other people's work.

As mentioned in the public review, I generally consider the work done by the authors timely and solid and would highly recommend it for publication in *eLife*, provided that a number of questions are addressed to improve the scientific rigor.

- Page 12: "The measure Deff(q) agrees well with the theoretical calculations, accounting for rigid body diffusion alone".

Can the authors provide a more objective way to assess the data-theory agreement in Figure 3A? The agreement doesn't really look good around q = 0.5/nm. It is not clear how large "a significant additional contribution of internal protein dynamics" should be to be visible as a clear deviation from the rigid-body diffusion.

- Page 13: fFCS experiments.

What does "treated as a single dataset" means for the 48 fFCS curves? The average fFCS curve? A multivariate fit to 48-dimensional data? Please clarify in the main text.

It is not clear why a 3-state model with varying amplitudes was used to recover the three correlation times shown in Figure 3C, D. The authors provide no justification for why 3 states are actually sufficient or insufficient. One way to address this could be to set up a Bayesian inference optimization and use the Bayesian-Schwarz criterion to discriminate between models with different parameter numbers.

- Why was the api-holo comparison done only for the kinetic measurements? Can the authors rule out any effects of GTP binding on the equilibrium quantities measured in Figure 2? The statement that the M1-M2 transition is GTP-independent is only complete and valid if both kinetic and equilibrium parameters of these systems are demonstrated to remain unchanged upon GTP binding.

- Why did the authors use a GTP-analog for their measurements? At picomolar concentrations of hGBP1 were monomeric (page 13, lines 297-298). Therefore, it wouldn't dimerize even in the presence of GTP at that concentration.

- The authors don't mention which version of aMD was used: the one that lowers the kinetic barriers or the one that raises the energy valleys? The use of aMD requires an explanation of how it might affect conclusions about kinetics (the authors do perform autocorrelation analysis).

- Potential GTP-dependent effects on the simulated hGBP1 are not sufficiently discussed. The RMSD distributions look different for apo vs. holo (Supp. Figure 6A). The average correlation times in the presence and absence of GTP differ as well, but the author seems to not take this result into consideration, nor do they comment on why this may or may not be statistically insignificant. Overall, the MD data contains much more valuable information than what the authors present. For example, one could assess the inter-domain correlations within hGBP1 as a function of the nucleotide state or compare the simulated data to smFRET measurements (of course, for the main conformer M1 only). Just to throw in a few ideas.

- Page 18: "indicative for a frustrated potential energy landscape". Do the authors mean a ragged landscape with several substrates and multiple kinetic barriers?

- As far as I can tell, the dimerization kinetics/thermodynamics were not assessed in the study because of the low hGBP1 concentrations necessary to keep it monomeric. It is not clear where the data comes from that supports the middle part of Figure 6 (D1 vs D2 states).

- Along the same lines, the claim that "ligand binding leads to dimerization because of the increased affinity between the LG domain" (page 20) doesn't seem to be supported by data at all.

- Also, what do the authors actually mean by "intrinsic flexibility"? When the widths of the M1/M2 potential wells are changed? When the transition rate between M1 and M2 is increased? When the equilibrium populations of M1 and M2 are changed? Or all of these at the same time?

- Page 21: "Upon addition of GTP the LG domain can bind to another promoter whilst the conformational dynamics appear to remain unchanged" and "As the M1:M2 dimer has a higher stability, …". Again, not clear whether the authors are referring to their own data, which I, unfortunately, wasn't able to find.

Overall, there are many places in the conclusions section where the actual findings by the authors should be clearly delineated from other people's work and/or physics intuition. This is very puzzling.

*Reviewer #4 (Recommendations for the authors):*

Peulen et al. provide a well-drafted manuscript describing dynamic conformers of the dynamin-related GBP1 GTPase. Using integrative modelling with DEER, FRET, and SAXS data, the authors identify at least two conformational states M1 and M2 in nucleotide-free GBP1 distinguished by a rotation around a pivot point between the GTPase and middle domain. The structural data are used to interpret the GTP-induced dimerization pathway, suggesting that intrinsic conformational flexibility between the GTPase and GTPase effector domain (GED) is a prerequisite for forming a bridged GBP1 dimer.

The manuscript provides and integrates a vast amount of structural and biophysical data that appear overall convincing. The chosen approach is of general interest since there is only a limited example combining biophysical approaches in such a systematic fashion to identify dynamic conformers of multi-domain molecular assemblies. Unfortunately, the authors appear to have used only non-farnesylated GBP1 for their studies while native GBP1 is C-terminally farnesylated. Consequently, it is not yet clear whether the observed conformational flexibility in the C-terminal GED is an intrinsic feature of GBP1 or a consequence of the missing post-translational modification. This and a few other minor issues should still be addressed to support the claim of the biological relevance of this study.

Figure 2, Figure 6, supporting Figure 2-4, farnesylation

In the cell, GBP1 is constitutively C-terminally farnesylated and the farnesyl moiety is known to bind into a pocket of the helical domains. If I understand Figure 6 correctly, the authors have used only non-farnesylated protein to characterize conformational flexibility in GBP1 (please mention the used construct explicitly in the Methods). Thus, it cannot be excluded that the observed flexibility in the C-terminal GED is related to the missing farnesyl moiety and may therefore not be an intrinsic property of the farnesylated protein. To exclude this scenario, farnesylated protein should be used to repeat at least a few of the DEER or FRET measurements in which two conformational states can clearly be discerned for the non-farnesylated protein (for example, Q344C/A496C and A496C/V540C). The two boxes 'non-farnesylated' and 'farnesylated' in Figure 6 are confusing since the relevant protein species in the cell is obviously farnesylated throughout the assembly.

Figure 3D, GDP-AlFx

AlFx is known to bind to GTPases when a stable transition state is achieved. For small GTPases, GDP-AlFx binding was demonstrated in the presence of GTPase activating proteins (GAP), while I am not aware of any study demonstrating GDP-AlFx-binding in the absence of a GAP. Accordingly, one would assume that GBP1 binds GDP-AlFx only in the context of a catalytic dimer, e.g. likely not at a protein concentration of 20 pM, as shown in Figure 3D. In this case, the measurements would represent the GDP-bound form, which would also explain the missing effect of nucleotide on protein dynamics. The authors should repeat the measurements for the selected three mutants in the presence of the relevant substrate GTP-Mg^2+^. At 20 pM protein concentration, GTP hydrolysis should be slow enough to allow FRET assays in the constant presence of GTP.

---

## [Author Response]

Reviewer #1 (Recommendations for the authors):The manuscript "Integrative dynamic structural biology unveils conformers essential for the oligomerization of a large GTPase" by Peulen et al. reports investigations on the solution structural ensemble and dynamics of human guanylate binding protein 1 (hGBP1) using a combination of complementary biophysical techniques. The authors apply single-molecule fluorescence resonance energy transfer (smFRET) spectroscopy on a number of donor/acceptor fluorophore pairs labeled to engineered cysteine (Cys) residues covering the entire structure of hGBP1. Time-resolved smFRET spectroscopy is used to determine distance distributions and correlation spectroscopy to study dynamics. A subset of double Cys mutants designed for FRET experiments was channeled into double electron-electron resonance (DEER) measurements, complementing the determination of distance distributions. The shape of hGBP1 in solution and its dynamics were further characterized using small-angle x-ray scattering (SAXS) and neutron spin echo (NSE) spectroscopy. In-silico studies were carried out using MD simulations and integrative modelling. The authors arrive at the conclusion of having resolved a novel native-state conformer of hGBP1 together with its time scales of interconversion, which is part of a mechanism of structural remodeling of hGPB1 dimers that form competent precursors in the human host defense system.Uncovering protein functional mechanisms requires knowledge of both structure and dynamics. Dynamic structural biology is an emerging and important research field. Methods and their applications to observe changes in protein conformation in solution lag behind the ever-increasing number of high-resolution structures solved. The GTPase hGPB1 is a central player in cell-autonomous immunity, which acts by cycling between dimeric, polymeric, and membrane-bound states on infectious pathogens driven by the binding and hydrolysis of GTP. This manuscript shows, within one piece of work, the impressive application of a combination of five different biophysical methods that complement each other in the exploration of spatial and temporal scales of protein conformation. The conformational heterogeneity and dynamics of hGBP1 detected in this work report on a hitherto unknown intermediate state in the conversion of hGBP1 from the resting to an active state. Some uncertainties remain in some of the experimental designs applying extensive mutagenesis and modification.I have the following comments.1. The authors use smFRET to measure distance distributions and apply a subset of the same Cys mutants to also measure distances using DEER. Can the authors comment on the difference in spatial scales probed by both techniques? To which degree are they complementary or does one technique benchmark/test the other?

In general the spatial scales are comparable between FRET and DEER, although spin labels do not exhibit a “Förster radius” around which the “working range” of the method is. The two techniques are complementary in a manner that they independently yield distance constraints for modeling, using different types of labels. For DEER only one type of label with a comparably short linker is necessary, whereas in FRET two different labels with different properties are used. (Theoretical) DEER distance distributions can therefore be relatively easily calculated for structural models. On the other hand, smFRET works at ambient temperature, whereas DEER measurements are done on frozen samples (50K). This might cause that conformational equilibria cannot be detected in all cases (like in this study). Thus, using FRET and DEER in parallel on the same set of label positions largely increases the reliability of the distance constraints and makes it quite easy to identify misleading data due to e.g. unexpected interactions between the labels and the proteins.

Changes to the manuscript, page 6:

We included an additional sentence in the last paragraph of the introduction to explain why the techniques are complementary:

“[…] smFRET and DEER independently yield distance restraints for modeling, the former with the advantage of being a single molecule technique that can be applied under ambient conditions, whereas the latter uses a single type of label that is smaller compared to FRET labels, simplifying treatment of the label for modeling purposes.”

2. In Figure 2A, the fit of the crystal structure (blue line) overlays well with the Kratky plot of SAXS data. Still, the authors argue that the crystal structure would disagree with the SAXS data, which appears unreasonable from looking at the fit to the data.

We revised the sentences on page 6 to emphasize the significant deviations of the theoretical SAXS curves calculated using the 1DG3 crystal structure from the experimental SAXS data. The mismatch is clearly visible in the weighted residuals as well as in the large -value of 9.3 that is an indicator for the mismatch between theoretical curve and experimental data.

Changes to the manuscript, page 6:

The sentences on page 6 have been modified to make it clearer for the reader to see that the hGBP1 solution structure as seen by SAXS is not in agreement with the crystal structure:

“A Kratky-plot of the SAXS data (Figure 2A, middle) visualizes that the non-farnesylated hGBP1 crystal structure 1DG3 disagrees with its structure in solution, which is clearly visible in the weighted residuals in the scattering vector range between 0.05 and 0.2 Å^-1^ showing a significant deviation of the theoretical SAXS curve of 1DG3 from the experimental SAXS data as well as in the large χr2 value of 9.3 that highlights the mismatch between theoretical 1DG3 SAXS curve and experimental data. *Ab initio* modeling of the SAXS data recorded for native non-farnesylated hGBP1 revealed a shape with an additional kink between the LG and the middle domain (Figure 2A, right, Figure 2 —figure supplement 1B), which does not agree with the straight crystal structure of non-farnesylated (PDB-ID: 1DG3) and farnesylated hGBP1 (PDB-ID: 6K1Z) (Figure 2A).”

3. On page 7 the authors argue that observed deviations of distances from the crystal structure and reduced spread, measured by DEER, could originate from a denser packing of the spin labels. How can a packing of spin labels in a solution structure be denser than in that of a crystal, which is optimally packed? The DEER/SAXS result appears to be in conflict with the conformational spread detected by smFRET. Won't a conformationally heterogeneous solution-structural ensemble always represent a more loose structure than that of a crystal? The discrepancies are also mentioned in the paragraph summary, in lines 211-212 on page 10, but appear not to be reconciled.

Although the solution structure of a protein might be (not necessarily!) less dense than its crystal structure, it can be also envisioned that a more flexible overall structure might cause label contacts with side chains in the vicinity of the spin label that point away from the label side chain in the rigid crystal structure. In extreme cases, crystal contacts can cause e.g. loop regions to be displaced compared to the structure in solution, and such loops might restrict the motional freedom of the label side chain in solution.

Changes to the manuscript, page 8:

“This can for example be the case if contacts between the molecules in the crystal reorient parts of the structure that are in contact with the label(s) in the solution ‘structure’.”

4. Wild type hGBP1 contains nine native Cys residues. The authors designed their Cys mutants for FRET and DEER spectroscopy on the background of a construct where all nine native Cys residues were replaced through mutagenesis to prevent their interference with site-specific modification. Replacement of such a high number of native Cys can potentially cause structural/energetic perturbations. Although the authors state in supporting note 1 that these mutations would only weakly affect hGBP1's function, some of the single and double Cys mutants (labeled/unlabeled), shown in Supporting Figure 4A, exhibit significant deviations/reductions of activity. What is the activity of the wild-type protein that contains all native Cys? It would be nice to see the wild-type activity also in the plots in Figure 4A for comparison. Can the authors exclude that their modifications induce conformational heterogeneity detected in smFRET experiments (the M1/M2 conformational ensemble)?

The deviations from X-ray structures were detected already for native hGBP1 by SAXS. We complement (integrate) the FRET / DEER information with SAXS (information not used to integrative modeling). Moreover, the hGBP1 activity of the cys variants is not below the base level of native hGBP1. We modified the manuscript as follows to highlight these facts. Thus, the Cys deletions necessary for the FRET/DEER experiments likely have only minor implications for the native hGBP1 function.

Changes to the manuscript, page 6:

“We performed DEER, FRET and SAXS experiments to probe short distances, long distances, and molecular shapes, respectively. For the DEER and FRET experiments, we used engineered non-farnesylated hGBP1 cysteine variants (Figure 1A) labeled with MTSSL (R1) as spin label and with Alexa488-Alexa647 as FRET pair (Förster radius *R_0_* = 52 Å), respectively. […] *Ab initio* modeling of the SAXS data recorded for native non-farnesylated hGBP1 revealed a shape with an additional kink between the LG and the middle domain (Figure 2A, right, Figure 2 —figure supplement 1B), which does not agree with the straight crystal structure of non-farnesylated (PDB-ID: 1DG3) and farnesylated hGBP1 (PDB-ID: 6K1Z) (Figure 2A). […] DEER and FRET experiments on engineered non-farnesylated hGBP1 cysteine variants probed distances between specific labeling sites (Figure 1A) – exemplified for the dual cysteine variant Q344C/A496C (Figure 2B, C). “

and page 18:

“For SAXS, pairs of structures are compared by computed scattering curves (Figure 4D, right). This comparison demonstrates that the integrative structures capture the essential features of the experiments and that the data recorded on non-farnesylated hGBP1 cysteine variants for FRET/DEER is consistent with SAXS data recorded on native non-farnesylated hGBP1.”

and Figure 2 —figure supplement 3:

We edited and added the wt (activity) in our GTPase activity assays as a horizontal dashed line for reference.

5. As a result of smFRET correlation spectroscopy, the authors find that dynamics in the middle and effector domain are an order of magnitude faster (~14 µs) than in the LG GTPase domain (~140 µs), highlighted in Figure 3E. This is an interesting finding. Since the secondary structure is the same throughout all hGBP1 domains one may have expected similarity in kinetics. The detected differences may report on functional motions. smFRET correlation spectroscopy covers the ns-µs time scale, which is indeed the relevant scale of motion of protein secondary and tertiary structure. NSE spectroscopy nicely complements the faster time scale of up to 200 ns. This faster time scale is also covered by MD simulations. Can we learn more details about the onset of the µs conformational changes seen in experiments from the computed atomistic MD data?

It is very interesting to integrate kinetic information quantitatively, Particularly, because the FRET experiments report on different aspects of the conformational transition. Unfortunately integrating kinetic information is not straight forward as even for a simple two-state system with a single transition path fluctuations of the FRET efficiency along the transition between may arise. Such a fluctuation and / or a more complex transition with transiently populated intermediates could explain the FRET pair dependent relaxation times and the complex experimental observables.

We did not interpret the onset of conformational transitions using MD simulation in more detail because, (i) the starting structure(s) (the X-ray structure) do not describe either of the solution structures, (ii) all structures in the MD simulations are too far away from the solution structure to be discussed in detail (Figure 4 —figure supplement 1C). Thus, we used a conservative interpretation of the MD data. In our analysis we capture the directionality of the motion by principal component analysis (PCA, Figure 3F) without discussing the results in detail or relating the results to the experimental findings. Our findings are consistent with previous MD simulations (Barz, Loschwitz et al. 2019) we observe that hGBP1 kinks in the simulation (PC 1) and motions of helix a12 (PC 4). As the MD simulations produced structures that are not close to the solution structure we used our integrative approach (Docking + NMSim) to model the solution structure. Unfortunately, our integrative model reports reliably on domain orientations and does not attain a resolution / accuracy required for further MD simulations (starting from the solution structure).

Changes to the manuscript, page 15:

“Consistent with previous computational observations (Barz, Loschwitz et al. 2019), a principal component analysis revealed kinking motions of the middle domain and helix α12/13 around a pivot point as most dominant motions in the MD simulations (Figure 3F).

6. Collective motions of protein domains are often slower than ms, in general. There may thus be functionally relevant motions of hGBP1 that were missed out. Can the authors comment on such potential motions and how they could be detected?

The collective motions of the studied hGBP1 are faster than milliseconds and slower than nanoseconds. Obviously we did not stress sufficiently how we come to the conclusion that in hGBP1 there are no slower motions. In case of slow dynamics on the millisecond to second time-scale we would observe broadened smFRET histograms. In our smFRET experiment the integration time is in the order of milliseconds. Thus, heterogeneous FRET efficiencies are visible by broad distributions static on the integration time-scale. For hGBP1 we did not observe broadened smFRET histograms. Thus, conformational dynamics must be faster than the integration time (milliseconds). To clarify these technicalities we modified the main text as follows.

Changes to the manuscript, page 13:

“In MFD-diagrams, “static FRET-lines” serve as a reference to detect fast conformational dynamics (Kalinin, Valeri et al. 2010). In case of sub-millisecond hGBP1 dynamics we expect to observe multimodal distributions (Barth, Opanasyuk et al. 2022, Opanasyuk, Barth et al. 2022). However, analogous to NMR relaxation dispersion experiments, we find a peak shift from the static FRET line towards longer ⟨τ_*D*(*A*)_⟩*_F_* is evidence for conformational dynamics faster than the observation time (~ms) (Kalinin, Valeri et al. 2010, Sisamakis, Valeri et al. 2010).“

7. The authors applied GDP-AlFx to generate a nucleotide-bound hGBP1 construct that remained monomeric under solution conditions of smFRET spectroscopy. It would be interesting to see how dimerization influences the measured hGBP1 dynamics. This could be studied by smFRET through the application of an excess of unlabeled hGBP1 to fluorescently modified, GDP-AlFx-bound hGBP1 construct. Have the authors considered such experiments?

It is an interesting question to be studied in a separate publication with a distinct focus. At low hGBP1 concentrations we do not find an effect on FRET. At higher hGBP1 concentrations the FRET efficiency changes (see point 9). Particularly because the structure of the hGBP1 dimer / oligomer is not very well studied. The dimerization induces large conformational changes. Thus, the dimerized hGBP1 will need to be studied and modeled by more measurements.

8. Binding of GMP is known to stabilize the compact resting state of hGBP1 (reviewed in Kutsch & Coers, FEBS J. 2021, 288, 5826-5849). Have the authors considered experiments using GMP as a ligand? Such experiments would be interesting because one would expect that binding of GMP stalls the dynamics of hGBP1 and thus the M1/M2 motions. Experiments involving the binding of GMP could provide insights into the functional relevance of the detected M1/M2 motions.

The review of Kutsch & Coers, FEBS J. 2021 presents reasonable hypotheses of the authors and working models of the field that to the best of our knowledge are not supported by direct biophysical evidence. Our GTP data show no visible effect on FRET (see point 9).

9. Model (iii), "dimer induced flexibility", in Figure 1B seems to be ruled out by the fact that the binding of GTP triggers the opening of GBP1 (reviewed in Kutsch & Coers, FEBS J. 2021, 288, 5826-5849).

We disagree, because derivatives of GTP without additional unlabeled hGBP that allow for dimerization have no effect (wee Figure 3E). We added these additional control experiments (Figure 3 —figure supplement 5).

Changes to the manuscript, page 14:

“For comparison, the affinity of hGBP1 for mant-GDP is ~3.5 μM and much higher for GDP-AlFx (Praefcke, Geyer et al. 1999). MFD control experiments performed on hydrolysable GTP agree with the non-hydrolysable GDP-AlF_x_ (Figure 3 —figure supplement 5B). Hence, in the sm-measurements GDP-AlF_x_ was bound to the LG domain while the non-farnesylated hGBP1 (20 pM) was still monomeric.”

Added figure for control experiment & referenced control experiments in main text

10. The last sentence of the abstract aims to highlight the conclusions of the work, which is deciphering intrinsic flexibility and conformational heterogeneity of hGBP1, but also mentions the GTP-triggered association of GTPase domains, assembly-dependent GTP hydrolysis, and functional design principles that control hGBP1's reversible oligomerization reported by others.

We agree. Thus, we separate the results of this and previous work.

Changes to the manuscript, page 2:

*Old:* “Our results show that an intrinsic flexibility, a GTP-triggered association of the GTPase-domains, and the assembly-dependent GTP-hydrolysis are functional design principles that control hGBP1’s reversible oligomerization.”

New: “Our results on hGBP1’s conformational heterogeneity and dynamics (intrinsic flexibility) deepen our molecular understanding relevant for its reversible oligomerization, GTP-triggered association of the GTPase-domains and assembly-dependent GTP-hydrolysis.”

Reviewer #2 (Recommendations for the authors):In their work, Peulen et al. investigate the conformational flexibility of the GTPase hGBP1, a dynamin-like protein responsible for the membrane disruption of intracellular parasites. The work was motivated by the apparent disagreement of the X-ray structure of the monomer with the previous FRET- and DEER experiments that suggested at least two conformations of the protein in its GTP-induced dimeric state. This disagreement indicates substantial flexibility, either in the monomeric state, which might or might not be induced by ligand binding or in the dimer. To understand the origin of this flexibility, the authors combined an impressive number of techniques, SAXS, NSE (neutron-spin echo spectroscopy), EPR, and FRET, in an integrative approach to unravel the structures and interconversion timescales of conformers in the apo- and holo-form of the protein. The authors found that at least two conformers (M1 and M2) interconvert at timescales from a few up to hundreds of microseconds. Integrative modeling combining the experimental results with rigid-body docking and structural refinement of a coarse-grained model of hGB1, following the previously published protocol of the authors (Dimura et al. 2020 Nature Comms, 11, 5394), was used to determine low-resolution models of M1 and M2. These models give rise to two potential dimers M1:M2 and M1:M1. The structural differences in M1 and M2 suggested by this integrative approach are substantial and indicate two different binding sites of the α12/13 helices on the ligand-binding (LG) domain.In some parts, the article is written in a rather technical style, which might complicate an understanding by readers unfamiliar with the techniques and approaches (structural biologists for instance). In addition, I have a number of technical comments that the authors should address.

We are aware of the problem that the materials & method section and the result section is difficult to read for non-experts. Thus, deliberately separated the results from the Discussion section. The Results section provides technical details for SAXS, NSE, DEER, and smFRET experts while the Discussion section summarizes the key biological findings for non-experts.

1. Figure 1B of the manuscript gives the impression that the authors investigate flexibility in the apo- and holo-form of the monomer but also in the dimer. I was therefore wondering why the authors excluded experiments in the presence of an excess of unlabeled monomer to investigate flexibility in the dimer. Does the formation of higher oligomers in the presence of GTP and high protein concentrations prevent such experiments?

See Reviewer #1 Q7.

2. Why are M1 and M2 not seen in the experimental DEER distance distributions, e.g., in the Q344C/A496C variant? Do the conformations re-equilibrate during the fast cooling period required in EPR? In fact, all EPR distributions are unimodal (p. 7, and Supplementary Figure 2), which is difficult to understand given the evidence for the existence of M1 and M2 from TCSPC and MFD histograms.

Indeed, most likely re-equilibration during sample freezing (the experiments are carried out at 50K sample temp.) prevents observation of the M1-M2 equilibrium. To clarify this, we modified the summarizing sentences at the end of the respective paragraph (pg. 10, l. 211-3) accordingly:

Changes in the manuscript, page 10:

“To describe the FRET data at least two states are necessary, which are not detected in the DEER experiments most likely due to re-equilibration of the two conformations during sample freezing.”

3. I was a bit puzzled by the eTCSPC analysis. The authors find bi-exponential lifetime decays for the donor in the absence of an acceptor, which is explained by the environment of the dyes after labeling. This indicates the presence of conformers with different degrees of dynamic donor quenching and hence different radiative rates of the donor dye. Yet, the combination of eq. 9 and 10 in the Supporting Information does not account for the propagation of these differences in the transfer rate. In fact, eq. 10 indicates that the same radiative rate (k0) was used to determine distance distributions. Although this approach is convenient as it allows εD to be obtained by the ratio of donor decay in the presence of the acceptor and in the absence of the acceptor, I am not yet convinced the approach is justified. IffD0 = ∑ xD(i) exp[-kD(i) t]is the donor decay in the absence of the acceptor (sum goes over i components), I'd expected that the donor decay in the presence of acceptor isfDA = ∑{xD exp[-kD(i) t] ∫exp[-kT(i,r) t]dr} (A)with kT(i,r)=kD(i) (R0/r)^6 being the component-dependent transfer rate and r is the donor-acceptor distance. Instead, the authors used eq. 9 and 10, which givefDA = {∑xD exp[-kD(i) t] } ∫exp[-kT(r) t]dr (B)with kT(r)=kD (R0/r)^6 with kD = k0 = 1⁄τ0.Clearly, A and B are two very different scenarios. In the latter, εD can simply be determined via fDA / fD0, but not in the former. Hence, if expression A is more adequate, a MEM-analysis of fDA / fD0 might provide false results.

The analysis methodology is an approximation that was previously found to be accurate for labels tethered to proteins (Peulen 2017). The approximation assumes that quenching by the local environment is uncorrelated to quenching by FRET. As highlighted by Figure 7 in Peulen 2017 this is clearly an approximation. Nevertheless, this approximation is accurate. However, the methodology is an approximation for another reason then pointed out by the reviewer, as described in the following equation (A)

fDA=∑i{xD(i)exp⁡[−kD(i)t]∫p(RDA,i)exp⁡[−kT(i,RDA)tdRDA]}*.* (eq.A)

Here, the quantum yield in kT(i) is accounted for in R0(i) and in kD(i) (historic definition of R0). In detail, kT(i)=kD(i)(R0(i)/RDA)6=kFΦD(i)(R0(i)/RDA)6 (where kF is the radiative rate constant) and R0(i)=(ΦD(i)κ2)1/69ln⁡(10)128π5NAJn4=(ΦD(i))1/6R0r. Thus, kT is independent of donor quenching.

The factorization of fDA into a FRET component ϵD(t) and a non-FRET component is an approximation, because p(RDA,i) depends on (i). In a worst case, the dye is located at an far edge of its sterical allowed space (from the point of view of the acceptor) and quenched. Nevertheless, even for such a worst case the homogeneous approximation results only in an error of ~5-6% in the distance (Peulen 2017, Figure 10). In calibrated dye simulations on a large set of proteins, we found that the here used homogeneous approximation recovers distances between quenched dyes flexibly coupled to proteins with an accuracy of ~5% (Peulen. 2017, Figure 12). We thank the reviewer for highlighting the pitfalls in such analysis. To raise the awareness of readers we changed the main text as follows:

Changes in the manuscript, page 8:

“For mono-exponential fDD(D0)(t) the position (time) and the height (amplitude) of steps in ϵD(t) correspond to DA distances and species fractions, respectively. […]

and page 9:

“The dyes are only weakly quenched to an extent that is expected for their local environment validating the used model of a mobile dye. (Appendix 1 – Table 2). In this case the ϵD(t) approximation showed to be accurate (Peulen, Opanasyuk et al. 2017).”

4. Related to point 3, the sub-population-specific TCSPC decays in the single-molecule FRET experiments seem to indicate single-exponential decays for donor-only molecules, at least by visual inspection (see Supporting Figure 3B). How is this reconciled with the double-exponential decays found for singly donor labeled hGBP1 in the ensemble TCSPC experiments?

We thank you for highlighting the necessity of a quantitative analysis. Lifetime of the dye in its unquenched state is ~ 4 ns while the quenched species have an average lifetime of approx 1.4 ns. For such mixtures as visual inspection can be misleading. The fraction of the quenched dyes is on average smaller than 0.15. Such effects can only be visualized by inspecting the weighted residuals and statistical tests, e.g. chi2 or Durbin-Watson test. We discussed the used approach and details on statistical tests and a Bayesian analysis of the posterior model density were presented in greater detail in previous publications (Vöpel, Hengstenberg et al. 2014, Peulen, Opanasyuk et al. 2017, Sanabria, Rodnin et al. 2020). To stress the importance of such statistical tests we changed the main text as follows.

Changes to manuscript, page 9:

“The distances of the states of all 12 data sets (Figure 1) were recovered by a joint/global analysis of all measured datasets. In this analysis, we consider distance uncertainty estimates, statistical uncertainties, potential systematic errors of the references, uncertainties of the orientation factor determined by the anisotropy of donor samples, and uncertainties of the AVs due to the differences of the donor and acceptor linker length (Appendix 2). We find at room temperature a relative population of 0.61 for M_1_ and 0.39 for M_2_ (Appendix 1 – Table 1). A qualitative inspection of fluorescence decay curves can be misleading. Thus, models (see Methods) were selected based on χ2 and Durbin-Watson tests and posterior model parameters densities were sampled in a Bayesian software framework (ChiSurf) as previously described (Vöpel, Hengstenberg et al. 2014, Peulen, Opanasyuk et al. 2017, Sanabria, Rodnin et al. 2020).

5. Have the authors checked whether static quenching of the acceptor (or donor) affects the ratiometric FRET efficiencies? Direct excitation of the acceptor in a (anti)-bunching experiment can reveal the significance of static quenching.

We calibrated the intensity histograms using both TCSPC (dynamic quenching) color FCS (static quenching) of single labeled donor and acceptor samples. We calibrated the intensity-based smFRET histograms by measuring single Cys variants solely labeled by donor and acceptor dyes. Moreover, we combined color FCS by TCSPC to disentangle the excitation power dependent triplet and of Alexa488 and the and excitation power dependent cis-trans isomerization of cyanine Al647 dye to calibrate the fluorescence quantum yield from other dark states. We estimated the fraction aT of Alexa647 in the trans state (or other dark state) and the fluorescence quantum yield of Alexa488 that considers power dependent dark states Moreover, we calibrated our setup using DNA rulers with known FRET efficiencies to determine the relative detection efficiency ratios of the donor and acceptor dye. The used procedures were established in the group two decades ago and are becoming common practice in the field [FRET challenge]. As, the presented results were the correction parameters were solely assembled in the Figure legend of the smFRET experiments and are not highlighted elsewhere.

As (i) our results largely rely on TCSPC and fFCS for determining distances and relaxation time and (ii) such calibrations are established in the field, we did not provide details. Note, TCSPC and fFCS are largely independent of static quenching. For these reasons the calibration of the smFRET was not highlighted and discussed in detail. Nevertheless, we thank you for raising the issue of calibrating intensities for quantitative fluorescence intensities. We agree that calibrations are essential for work as presented here. Thus, in addition to the calibration parameters shown as inset in the supplemental figures we present the calibration parameters more prominently in a separated table (Appendix 1 – Table 3) and change the main text as follows. All raw data including the necessary calibration measurements are included in the Zenodo upload accompanying this manuscript.

Changes to manuscript, pages 12/13:

“To cover sub-µs to ms dynamics, we performed MFD smFRET experiments on freely diffusing molecules. We determine for every molecule the average fluorescence lifetime of the donor, ⟨*τ*_D(A)_⟩*_F_*, and the FRET efficiency, *E*, to create MFD-diagrams that visualize heterogeneities among the molecules. MFD diagrams correlate calibrated intensity-based observables to the fluorescence lifetimes for revealing conformational heterogeneity. Absolute FRET efficiencies require calibrations. We calibrate our instrument by DNA reference samples as previously described (Hellenkamp, Schmid et al. 2018) and account for sample-specific dark-states using fluorescence decay and FCS measurement of single-labeled samples (Appendix 1 – Table 3). In MFD-diagrams, “static FRET-lines” serve as a reference to detect fast conformational dynamics (Kalinin, Valeri et al. 2010). In case of sub-millisecond hGBP1 dynamics we expect to observe multimodal distributions (Barth, Opanasyuk et al. 2022, Opanasyuk, Barth et al. 2022).”

and page 34:

“Moreover, the background, the detection efficiency-ratio of the “green” and “red” detectors, and the spectral cross-talk were considered to determine the FRET efficiency, *E*, of every burst (Sisamakis, Valeri et al. 2010). Absolute FRET efficiencies require calibrated instruments (Hellenkamp, Schmid et al. 2018) and considerations of the excitation power and FRET-dependent photophysics (Widengren, Schweinberger et al. 2001).”

6. Supporting Figure 4: Panels (C) and (D) are switched in the caption.7. It is known that D2O (used in the EPR experiment) can affect protein stabilities and flexibilities (Makhadatze et al. 1995 Nat Struct Biol, 2, 852 / Cioni & Strambini (2002) Biophys J, 82, 3246 / Pica & Graziano (2017) Biopolymers, 109 and many more). Did the authors find differences in the populations and or dynamics of the state between the two solvents? Controls with MFD should be straightforward.

It is indeed well known that D2O stabilizes/rigidifies the "native state" of a protein (the quintessence of the references mentioned by the reviewer). Nevertheless, as the conformational equilibrium is not seen in the DEER experiments most likely already due to the experimental conditions (frozen state), we decided to use samples with deuterated buffer to extend the accessible distance range and maximize the data quality and consequently the reliability of the obtained distance constraints for modeling of M1.

Reviewer #3 (Recommendations for the authors):In their work, Peulen et al. applied an integrative modeling approach to scrutinize the conformational dynamics of the human guanylate binding protein (hGBP1), which is an essential part of the cell's intrinsic immune response to intracellular parasites. The assortment of biophysical methods used is striking – the fact that the authors have been able to plan and assign experiments/simulations, collect and process data, and integrate the diverse data into a meta-analysis workflow to derive molecular candidates for hGBP1 conformers is exemplary and not self-evident. This has allowed deriving important conclusions about the mechanism of hGBP1 self-assembly. In particular, the authors propose that hGBP1 undergoes GTP-independent conformational transitions on the µs-timescale between at least two conformational states, which is essential for the slower, GTP-dependent hGBP1 oligomerization. It is highly unlikely that these conclusions could ever have been made without the cross-disciplinary scheme presented by the authors. On the other hand, it is also crucial to note that the study has to incorporate several caveats that will be important when placing the proposed oligomerization pathway (Figure 6 of the paper) in a broader biological context and that several clarifications will need to be provided to delineate the conclusions based on measurements/calculations actually done by the authors from speculations based on physics intuition and/or other people's work.As mentioned in the public review, I generally consider the work done by the authors timely and solid and would highly recommend it for publication in eLife, provided that a number of questions are addressed to improve the scientific rigor.1. Page 12: "The measure Deff(q) agrees well with the theoretical calculations, accounting for rigid body diffusion alone".Can the authors provide a more objective way to assess the data-theory agreement in Figure 3A? The agreement doesn't really look good around q = 0.5/nm. It is not clear how large "a significant additional contribution of internal protein dynamics" should be to be visible as a clear deviation from the rigid-body diffusion.

Figure 3A should be assessed together with Figure 3 —figure supplement 1A shows only the effective diffusion coefficients as obtained from the initial slopes of the NSE spectra (larger errors, very sensitive to the first points). The long time diffusion coefficients with t > 25 ns as shown in Supp. Figure 5B are well described by the long time model without internal protein dynamics. Initial slopes (extracted using a cumulant fit) show in general a larger noise. The important observation is that the transition of the effective diffusion coefficients from the initial slope to the one obtained from the long time limit t > 25 ns can be fully explained by the asymmetric shape of the hGBP1 protein performing rigid-body diffusion (translational and rotational diffusion) without any need to consider internal protein dynamics.

Changes to the manuscript, page 12:

We added the following sentences to the manuscript on page 12 to provide an objective statement on the contribution of internal protein dynamics as detectable by NSE:

“The same result was obtained by directly optimizing the parameters of an analytical model describing rigid protein-diffusion (Materials and methods, eq. 9) to the NSE spectra (Figure 3 —figure supplement 1B) by ΔDeff∼u2/τ with MSD u2 and relaxation time τ of internal protein dynamics. Hence, a reasonably large u2-value could in principle be compensated by a long relaxation time τ of internal dynamics leading to small ΔDeff-values. To test this scenario and to assess potential contributions of internal protein dynamics to rigid-body diffusion we consider a full analytical model as described in the SI and examine the intermediate scattering functions I(q,t) (Figure 3 —figure supplement 1A). The additional contribution at short times due to internal dynamics can be estimated by a Debye-Waller argument (Biehl, Hoffmann et al. 2008): Internal protein dynamics with a MSD of u2 leading to a change in the I(q,t) within the observed errors (errors ~ 0.007 < 0.01) can be estimated by −u2q23=ln⁡(1−err.). If we consider the deviation at Q=0.5 nm−1 then we obtain a value of u=0.25 nm. Compared to the size of the hGBP1 protein internal motions with such small amplitudes are not observable by NSE. A significant additional contribution of internal protein dynamics to the measured effective diffusion coefficients cannot be identified. Hence, the overall internal protein dynamics may only result in negligible amplitude, i.e., minor overall shape changes, within the observation time up to 200 ns.”

2. Page 13: fFCS experiments.What does "treated as a single dataset" means for the 48 fFCS curves? The average fFCS curve? A multivariate fit to 48-dimensional data? Please clarify in the main text.

Thank you for this editorial remark. Indeed we performed a multivariate optimization of the model parameters. The fluorescence field uses for such analysis the terminology “global-analysis”. In a global analysis “local” and “global” model parameters are optimized simultaneously to all datasets. We combine optimization with sampling and try to minimize the use of classical “fitting”, as it can be quite dangerous (local optima) we optimize and sample the model parameters.

See response to remark 3.

3. It is not clear why a 3-state model with varying amplitudes was used to recover the three correlation times shown in Figure 3C, D. The authors provide no justification for why 3 states are actually sufficient or insufficient. One way to address this could be to set up a Bayesian inference optimization and use the Bayesian-Schwarz criterion to discriminate between models with different parameter numbers.

Thank you for raising this concern. Generally, choosing a model is not trivial. The BIC (or the AIC) can indeed be a useful tool to discriminate models. Here, the 3 relaxation time model was the simplest model that could jointly describe the fFCS curves of all variants (Figure S3). As the model choice does not impact the general statements of the manuscript, we did not discuss the motivation leading to the 3 relaxation time model to improve the readability. Sufficient (exhaustive) parameter sampling and optimization particular in global analysis is challenging. Thus, we performed our analysis that led to the 3-relaxation time model in two steps. First, we analyzed the fFCS data of the FRET pairs individually. As visible in Figure 3C at least two relaxation times are necessary to describe the fFCS curves of each FRET pair. The relaxation times vary across the FRET variants (Figure S3). This makes a comparison of different FRET variants difficult (evident by the variability of the average correlation time). Describing the fFCS curves of all variants individually by 2 correlation times and 1 amplitude (each) requires overall 12 * 3 = 36 free model parameters. A global description of all FRET variants using our global analysis with 3 relaxation times requires less model parameters (27 = 12 * 2 + 3) and allows to compare kinetics across FRET variants. Thus, we performed the described global analysis where the relaxation times were global parameters. As we described previously, we used in this manuscript conservative F-tests to compare models (Vöpel, Hengstenberg et al. 2014, Peulen, Opanasyuk et al. 2017, Sanabria, Rodnin et al. 2020).

Changes to manuscript, pages 13/14:

**“**To quantify the dynamics, we performed filtered FCS (fFCS) and jointly analyzed all species cross-correlation functions (*sCCF*) and the species autocorrelation functions (*sACF*) by a single model (Figure 3 —figure supplement 4) to determine characteristic times (Appendix 1 – Table 6). For a pure two state system (M_1_ ⇌ M_2_), we expected to find a single characteristic time. However, at least two relaxation times with corresponding amplitude were required to describe individual fFCS datasets (36 relevant free parameters). Thus, there are more than two (kinetic) states. To compare the relaxation times across FRET variants, and to reduce the number of free parameters we performed a global analysis of the fFCS data (Materials and methods, Equations. 17-19). In global analysis local and global parameters are simultaneously optimized (Beechem, Gratton et al. 2002). Global parameters are varied parameters shared across datasets. Local parameters are varied parameters of a single dataset. In our analysis, relaxation times were global parameters (3 relaxation times, shared across FRET variants), corresponding amplitudes were local parameters (2 amplitudes per FRET pair) of FRET variants. Model parameters and uncertainties were determined by optimizing and sampling over local and global parameters using the sum of all weighted squared deviations computed for all 48 model and experimental fFCS curves as objective function. Figure 3 —figure supplement 4 display the experimental fFCS and model fFCS curves computed for the global analysis result (Appendix 1 – Table 6). This analysis recovered three correlation times (2, 23 and 297 µs) with significantly varying amplitudes (Figure 3D, Appendix 1 – Table 6) and average relaxation times varied approximately (gray bars in Figure 3D). In most cases, the shortest component has the highest amplitude. This is consistent with the MFD-diagrams because we detected shifted/dynamic unimodal peaks. We mapped the average correlation times color coded to the FRET network (Figure 3E). This highlights that the fast dynamics is associated to α12/13 and the middle domain while the slow dynamics is predominantly linked to the LG domain.”

3. Why was the api-holo comparison done only for the kinetic measurements? Can the authors rule out any effects of GTP binding on the equilibrium quantities measured in Figure 2? The statement that the M1-M2 transition is GTP-independent is only complete and valid if both kinetic and equilibrium parameters of these systems are demonstrated to remain unchanged upon GTP binding.

We found the average FRET efficiency, a measure for the equilibrium concentrations of the states, does not change upon GTP and GDP-AlFx binding (see Reviewer #1, remark 9).

Changes to manuscript

We performed experiments and added Figure 3 —figure supplement 5 to illustrate that the GTP bound state has unchanged FRET efficiencies.

5. Why did the authors use a GTP-analog for their measurements? At picomolar concentrations of hGBP1 were monomeric (page 13, lines 297-298). Therefore, it wouldn't dimerize even in the presence of GTP at that concentration.

smFRET measurements are performed over the time-scale of hours. GTPases like hGBP1 slowly hydrolyze GTP producing GMP and GDP that can also be bound. We wanted to avoid such heterogeneity during the measurement and thus used non-hydrolysable GTP analogs.

Changes to manuscript, page 14:

“Referring to the sketch in Figure 1B, we hypothesize that the states M_1_ and M_2_ and the transition among them are of functional relevance (pathway i). Therefore, we studied the effect on the dynamics exerted by the ligand GDP-AlF_x_ as a non-hydrolysable substrate that mimics the holo-state hGBP1:L.”6. The authors don't mention which version of aMD was used: the one that lowers the kinetic barriers or the one that raises the energy valleys? The use of aMD requires an explanation of how it might affect conclusions about kinetics (the authors do perform autocorrelation analysis).

We used the original aMD formulation implemented in AMBER that raises the valleys. We do not use or discuss the aMD kinetics. The aMD simulations were only performed to enhance the conformational sampling. The correlation analysis was performed on conventional MD simulations. We clarify this as follows.

Changes to manuscript, page 15:

“The apo (PDB-ID: 1DG3) and a GTP bound holo-form of non-farnesylated hGBP1 were simulated in three replicas by conventional MD simulations for 2 µs each (Figure 4 —figure supplement 1A). Additionally, accelerated molecular dynamics (aMD) simulations, which samples free-energy landscapes of a small protein approximately 2000-fold more efficiently (Pierce, Salomon-Ferrer et al. 2012), were performed in two replicas of 200 ns each to enhance the conformational sampling. An autocorrelation analysis of the of the RMSD determined for the conventional MD simulations *vs.* the average structure of the MD simulations reveals fast correlation times. The average correlation time in the presence and the absence of GTP were 11 ns and 17 ns, respectively (Figure 4 —figure supplement 1B).”

7. Potential GTP-dependent effects on the simulated hGBP1 are not sufficiently discussed. The RMSD distributions look different for apo vs. holo (Supp. Figure 6A). The average correlation times in the presence and absence of GTP differ as well, but the author seems to not take this result into consideration, nor do they comment on why this may or may not be statistically insignificant. Overall, the MD data contains much more valuable information than what the authors present. For example, one could assess the inter-domain correlations within hGBP1 as a function of the nucleotide state or compare the simulated data to smFRET measurements (of course, for the main conformer M1 only). Just to throw in a few ideas.

The MD simulations do not sample the conformational transitions seen in the experiments. The slowest relaxation time is in the order of ~300 microseconds. Thus, differences of the MD simulations in the short time-scale are not supported by experimental evidence and were thus not discussed. We did not interpret the MD simulations in more detail because the experiments showed that (i) the starting structure (the X-ray structure(s)) disagrees with the experimental structures/shapes and (ii) the conformation transitions (~us) are slow (compared to the time-scale of the MD simulations). This is experimental evidence on the slow transition between conformers cautions on a more detailed interpretation of the MD simulations.

8. Page 18: "indicative for a frustrated potential energy landscape". Do the authors mean a ragged landscape with several substrates and multiple kinetic barriers?

Changes to manuscript, page 20:

“The distribution of dynamics over such a wide range is indicative of a frustrated/rugged potential energy landscape with several substates and multiple kinetic barriers.”

9. As far as I can tell, the dimerization kinetics/thermodynamics were not assessed in the study because of the low hGBP1 concentrations necessary to keep it monomeric. It is not clear where the data comes from that supports the middle part of Figure 6 (D1 vs D2 states).Along the same lines, the claim that "ligand binding leads to dimerization because of the increased affinity between the LG domain" (page 20) doesn't seem to be supported by data at all.

In the manuscript the results and the Discussion section are separated. The remarks refer to the Discussion section that summarizes knowledge in the field and puts the results into context. We better delineate our findings from the knowledge in the field and highlight better separate results from interpretation.

Changes to manuscript, page 21:

“We previously showed that non-farnesylated hGBP1 forms dimers via the LG domains (in a head-to-head manner) *and* via helix α13 (Vöpel, Hengstenberg et al. 2014) in the presence of a GTP analog. This finding is inconsistent with non-farnesylated nucleotide free (PDB-ID: 1DG3), nucleotide bound (PDB-ID: 1F5N) and farnesylated nucleotide free (PDB-ID: 6K1Z) full-length crystal structures. In dimers formed by two hGBP1s in a 1DG3, 1F5N, or 6K1Z conformation the helices α13 are on opposite sides and thus could not be associated (Figure 5). Similar findings were recently published for hGBP5 (Cui, Braun et al. 2021), showing that the middle domain undergoes a drastic movement after GTP binding, forming a closed dimer. However, in a dimer formed of two distinct conformers (M_1_:M_2_), the helices α13 are located on the same side of their LG domains. Thus, GTP binding likely leads to dimerization because of the increased affinity between the LG domain. In the formed dimer the low affinity between the α12/13 helices and the middle domain suffices to induce opening like a pocket knife which is the prerequisite for oligomerisation.

and page 23:

[…] Previous data revealed two hGBP1 dimer conformations. In the major populated D_2_ conformation two a13 helices dimerize while in the minor D_1_ conformation helix a13 are separated (Kravets, Degrandi et al. 2012, Vöpel, Hengstenberg et al. 2014, Kravets, Degrandi et al. , Shydlovskyi, Zienert et al. 2017). Our new findings in this work lead to a common model which describes the reaction pathway of hGBP1 from a monomer to the formation of mesoscale droplets in vitro and living cells (Figure 6). We found that M_1_ is the prevailing conformation in solution. Thus, even though hGBP1 is flexible it likely first dimerizes via the LG domain to form a stable D_2_ dimer. All structural requirements for this multi-step conformational rearrangement for positioning the two interaction sites and defining the molecular polarity are already predefined in the monomeric hGBP1 molecule.”

10. Also, what do the authors actually mean by "intrinsic flexibility"? When the widths of the M1/M2 potential wells are changed? When the transition rate between M1 and M2 is increased? When the equilibrium populations of M1 and M2 are changed? Or all of these at the same time?

Changes to manuscript page 2:

“Our results on hGBP1’s conformational heterogeneity and dynamics (intrinsic flexibility) deepen our molecular understanding relevant for its reversible oligomerization, GTP-triggered association of the GTPase-domains and assembly-dependent GTP-hydrolysis.

11. Page 21: "Upon addition of GTP the LG domain can bind to another promoter whilst the conformational dynamics appear to remain unchanged" and "As the M1:M2 dimer has a higher stability, …". Again, not clear whether the authors are referring to their own data, which I, unfortunately, wasn't able to find.

Thank you for highlighting this issue. The referred sentence in the Discussion section summarizes the general knowledge of the field and fails to sufficiently delineate the findings (unchanged dynamics upon nucleotide binding) from general knowledge GTP induced dimerization.

Changes to manuscript page 23:

**“**In the absence of substrate and other GBP molecules, hGBP1 adopts at least two distinct conformational states. Upon addition of GTP, the LG domain can bind to another protomer, whilst the conformational dynamics appear to remain unchanged (Vöpel, Hengstenberg et al. 2014) which agrees with our current findings.**”**

12. Overall, there are many places in the conclusions section where the actual findings by the authors should be clearly delineated from other people's work and/or physics intuition. This is very puzzling.

The referred points in the discussion were edited. We separated the results from the Discussion section to avoid confusions of the reader.

Reviewer #4 (Recommendations for the authors):Peulen et al. provide a well-drafted manuscript describing dynamic conformers of the dynamin-related GBP1 GTPase. Using integrative modelling with DEER, FRET, and SAXS data, the authors identify at least two conformational states M1 and M2 in nucleotide-free GBP1 distinguished by a rotation around a pivot point between the GTPase and middle domain. The structural data are used to interpret the GTP-induced dimerization pathway, suggesting that intrinsic conformational flexibility between the GTPase and GTPase effector domain (GED) is a prerequisite for forming a bridged GBP1 dimer.The manuscript provides and integrates a vast amount of structural and biophysical data that appear overall convincing. The chosen approach is of general interest since there is only a limited example combining biophysical approaches in such a systematic fashion to identify dynamic conformers of multi-domain molecular assemblies. Unfortunately, the authors appear to have used only non-farnesylated GBP1 for their studies while native GBP1 is C-terminally farnesylated. Consequently, it is not yet clear whether the observed conformational flexibility in the C-terminal GED is an intrinsic feature of GBP1 or a consequence of the missing post-translational modification. This and a few other minor issues should still be addressed to support the claim of the biological relevance of this study.

In our previous SAXS studies on post-translational modified hGBP1 we identified specific interactions and oligomerized hGBP1 (Lorenz, Ince et al. 2020). In farnesylated hGBP1 the fraction of oligomerized hGBP1 was lower leading to the speculation that hGBP1 farnesylation acts as a safety mechanism preventing structural rearrangements and uninduced dimerization (SASBDB: SASDEG8, SASDEH8). In our previous studies the existing data (only SAXS) did not allow for detailed integrative modeling and SAXS data need to be corroborated by smFRET and other experiments to resolve conformational heterogeneity. Studying hGBP1 including its oligomerization (dynamics) is beyond the scope of this paper. Thus, as the known monomer structures are incompatible with dimer formation, we performed the measurements on non farnesylated hGBP1 to better study the structural prerequisites for dimerization and ask the question: How can a rigid protein form dimers? To address this question, we deliberately release the farnesylation “safety latch” to study nanomechanics for flexible derivatives and find hidden motional possibilities to understand hGBP1’s construction principles for nanomechanics. To clarify our motivation we edited the introduction and the discussion.

Changes to manuscript, page 5:

“In the dominant dimer, the two C-terminal α13 helices associate (Vöpel, Hengstenberg et al. 2014). This is in line with live-cell experiments that highlight the relevance of helix α13 for the immune response (Tietzel, El-Haibi et al. 2009, Li, Jiang et al. 2017, Piro, Hernandez et al. 2017). Previously, we identified monomeric and dimeric forms of farnesylated hGBP1 by SEC-SAXS and ultracentrifugation (Figure 1 —figure supplement 1). These experiments lead to the hypothesis that specific intramolecular interactions stabilize the GTPase and act as a safety mechanism preventing hGBP1 dimerization (Lorenz, Ince et al. 2020). Here, to unravel the conformational changes necessary for the formation of a fully bridged dimer (b-hGBP1:L)_2_ (Figure 1B), we study non-farnesylated hGBP1, where nucleotide ligands L (GTP) are bound and the effector domains and the LG domains are both associated (Figure 1B). The association of two α13 helices in a dimer requires large-scale structural rearrangements that cannot be explained by known X-ray structures (Ghosh, Praefcke et al. 2006). On the pathway to a bridged dimer, there are at least two intermediates – the ligand complex hGBP1:L and the flexible dimer (f-hGBP1:L)_2_ (Figure 1B).”

and page 21:

*“*To understand the functional relevance of M_1_ and M_2_, various observations and existing experimental information on farnesylated hGBP1 must be considered. We previously speculated that the farnesyl anchor acts as a “safety latch” that attaches α12/13 to the LG domain. Nevertheless, we identified monomeric as well as dimeric forms of farnesylated and non-farnesylated hGBP1 by SEC-SAXS that both require large structural rearrangements (Lorenz, Ince et al. 2020). Thus, the dimerization, as the first step in oligomerization of hGBP1, is an important feature that demands flexibility of the structure as deduced from major structural rearrangements described so far (Vöpel, Hengstenberg et al. 2014, Ince, Kutsch et al. 2017, Shydlovskyi, Zienert et al. 2017). […]

and page 24:

The rule that functionally relevant conformational equilibria may be pre-defined by protein design also applies to other steps in protein function. In future, when considering additional quantitative live-cell, single-molecule and kinetic studies on farnesylated hGBP1, such integrative approaches may provide a molecular picture of complex biological processes like intracellular immune response.*”*

Figure 2, Figure 6, supporting Figure 2-4, farnesylationIn the cell, GBP1 is constitutively C-terminally farnesylated and the farnesyl moiety is known to bind into a pocket of the helical domains. If I understand Figure 6 correctly, the authors have used only non-farnesylated protein to characterize conformational flexibility in GBP1 (please mention the used construct explicitly in the Methods). Thus, it cannot be excluded that the observed flexibility in the C-terminal GED is related to the missing farnesyl moiety and may therefore not be an intrinsic property of the farnesylated protein. To exclude this scenario, farnesylated protein should be used to repeat at least a few of the DEER or FRET measurements in which two conformational states can clearly be discerned for the non-farnesylated protein (for example, Q344C/A496C and A496C/V540C). The two boxes 'non-farnesylated' and 'farnesylated' in Figure 6 are confusing since the relevant protein species in the cell is obviously farnesylated throughout the assembly.

The suggestion of reviewer 4 is very good, but the suggested experiments will take a lot of time and effort. Currently, we are working on it. Therefore, studies with farnesylated protein are beyond the scope of this publication. To understand the effect of farnesylation, non-farnesylated protein must be characterized in detail at first, which is the focus of this work.

Figure 3D, GDP-AlFxAlFx is known to bind to GTPases when a stable transition state is achieved. For small GTPases, GDP-AlFx binding was demonstrated in the presence of GTPase activating proteins (GAP), while I am not aware of any study demonstrating GDP-AlFx-binding in the absence of a GAP. Accordingly, one would assume that GBP1 binds GDP-AlFx only in the context of a catalytic dimer, e.g. likely not at a protein concentration of 20 pM, as shown in Figure 3D. In this case, the measurements would represent the GDP-bound form, which would also explain the missing effect of nucleotide on protein dynamics. The authors should repeat the measurements for the selected three mutants in the presence of the relevant substrate GTP-Mg^2+^. At 20 pM protein concentration, GTP hydrolysis should be slow enough to allow FRET assays in the constant presence of GTP.

Some small GTPases (e.g. RAS) require GAP for binding of GDP-AlFx, as an arginine of the GAP molecules stabilizes AlFx in the bound state. hGBP1 differs from RAS, as the arginine that stabilizes AlFx (i.e., R48) is part of the GTPase domain. Thus, a GAP is not necessary for the binding of GDP-AlFx. A binding of AlF4- to monomeric hGBP1 cannot be excluded by the absence of an external GAP. In the measurements the F- ion concentration was 10 mM and the Al3+ concentration 300uM. The acquisition time in a FRET experiment is in the order of 1-2 hours. Over the course of a measurement, even at low GTP concentrations we expected heterogeneity related to nucleotides. Thus, we performed experiments with a non-hydrolysable analog.